**Title:** Elephant megacarcasses increase local nutrient pools in African savanna soils and plants
Courtney G. Reed[1*], Michelle L. Budny[2], Johan T. du Toit[3,4], Ryan Helcoski[3], Joshua P.
Schimel[1], Izak P. J. Smit[5,6], Tercia Strydom[5], Aimee Tallian[3,7], Dave I. Thompson[8,9], Helga van
Coller[8,10], Nathan P. Lemoine[2,11], Deron E. Burkepile[1,8*]
[1]Department of Ecology, Evolution, and Marine Biology, University of California, Santa
Barbara, CA, USA
[2]Department of Biological Sciences, Marquette University, Milwaukee, WI, USA
[3]Department of Wildland Resources, Utah State University, Logan, UT, USA
[4]Institute of Zoology, Zoological Society of London, London, England, UK
[5]Scientific Services, South African National Parks, Skukuza, South Africa
[6]Sustainability Research Unit, Nelson Mandela University, George, South Africa
[7]Norwegian Institute for Nature Research, Høgskoleringen 9 Trondheim, 7485 Norway
[8]South African Environmental Observation Network (SAEON), Ndlovu Node, Phalaborwa,
South Africa
[9]Unit for Environmental Sciences and Management, Potchefstroom Campus, North West
University, Potchefstroom, South Africa
[10]The Expanded Freshwater and Terrestrial Environmental Observation Network (EFTEON),
Kimberley 8306, South Africa
[11]Department of Zoology, Milwaukee Public Museum, Milwaukee, WI, USA
*Corresponding authors: Courtney Reed, courtneyreed@ucsb.edu, Deron Burkepile,
dburkepile@ucsb.edu

**Abstract**

African elephants (*Loxodonta africana*) are the largest extant terrestrial mammals, with bodies containing enormous quantities of nutrients. Yet we know little about how these nutrients move through the ecosystem after an elephant dies. Here, we investigated the initial effects (1-26 months postmortem) of elephant megacarcasses on savanna soil and plant nutrient pools in Kruger National Park, South Africa. We hypothesized that: (H1) elephant megacarcass decomposition would release nutrients into soil, resulting in higher concentrations of soil nitrogen (N), phosphorus (P), and micronutrients near the center of carcass sites; (H2) carbon (C) inputs to the soil would stimulate microbial activity, resulting in increased soil respiration potential near the center of carcass sites; and (H3) carcass-derived nutrients would be absorbed by plants, resulting in higher foliar nutrient concentrations near the center of carcass sites. To test our hypotheses, we identified 10 elephant carcass sites split evenly between nutrient-poor granitic and nutrient-rich basaltic soils. At each site, we ran transects in the four cardinal directions from the center of the carcass site, collecting soil and grass (*Urochloa trichopus,* formerly *U. mosambicensis*) samples at 0, 2.5, 5, 10, and 15 m. We then analyzed samples for CNP and micronutrient concentrations and quantified soil microbial respiration potential. We found that concentrations of soil nitrate, ammonium, $\delta^{15}$N, phosphate, and sodium were elevated closer to the center of carcass sites (H1). Microbial respiration potentials were positively correlated with soil organic C, and both respiration and organic C decreased with distance from the carcass (H2). Finally, we found evidence that plants were readily absorbing carcass-derived nutrients from the soil, with foliar %N, $\delta^{15}$N, iron, potassium, magnesium, and sodium significantly elevated closer to the center of carcass sites (H3). Together, these results indicate that elephant megacarcasses release ecologically consequential pulses of nutrients into the soil

that influence soil microbial activity and are absorbed by plants into the above-ground nutrient

pools. These localized nutrient pulses may drive spatiotemporal heterogeneity in plant diversity,

herbivore behavior, and ecosystem processes.

**Sect. 1 Introduction**

Living animals affect nutrient flows through ecosystems (Schmitz et al. 2018), but we have only recently acknowledged that the nutrients from animal carcasses could also influence ecosystem processes (Barton et al. 2013; Monk et al. 2024). In marine ecosystems, whale carcasses function as unique hotspots of nutrient cycling, biodiversity, and ecosystem processes (Roman et al. 2014). In terrestrial systems, mass mortality events (e.g., wildebeest, cicadas) create nutrient hotspots (Yang, 2004; Subalusky et al. 2020), while individual small and medium-sized carcasses release pulses of nutrients into the soil (Town, 2000; Barton et al. 2016; Olea et al. 2019). Yet, terrestrial ecosystem ecology lacks knowledge about the role of megacarcasses (carcasses of animals such as elephants and rhinoceros that are >1000 kg at death) as potential drivers of spatiotemporal heterogeneity in nutrient cycling and ecosystem processes. Importantly, these megacarcasses may be functionally different than smaller carcasses due to the extraordinarily high concentration of nutrients and residence time of the decomposing animal (see reviews by Barton et al. 2013; Barton, 2016; Barton & Bump 2019). This question around the role of megacarcasses is particularly relevant given the megaherbivore losses that occurred during the Pleistocene extinctions and that are still occurring today (Ripple et al. 2015). We are only beginning to understand how the 'extinction aftershock' of losing the largest species impacts ecosystems (Owen-Smith, 1989; Flannery, 1990), and no study has yet investigated how the loss of megacarcasses might influence the dynamics of terrestrial ecosystems (Doughty et al. 2013; Doughty et al. 2016).

We can only evaluate the importance of terrestrial megacarcasses for nutrient cycling in ecosystems where megaherbivores still exist, such as African savannas. The African savanna elephant (*Loxodonta africana*) is the largest extant land animal and is known for its key

ecological effects in savannas while alive (e.g., dispersing seeds, creating plant refuges,
preventing woody encroachment) (Skarpe et al. 2004; Asner et al. 2009; Campos-Arceiz &
Blake, 2011; Coverdale et al. 2016; Guy et al. 2021). Yet, the elephant's large body mass may
mean that it also has an outsized impact in these ecosystems even after death. A 4000-kg
elephant megacarcass likely represents ~2000 kg carbon (C), ~300 kg nitrogen (N), and ~125 kg
phosphorus (P) deposited in the savanna landscape (estimated from stoichiometry of elephants
and other mammals in Sterner & Elser, 2002). The N deposition from one elephant megacarcass
(in a 700 $m^2$ impact zone assuming a 15 m disturbance radius) is roughly equivalent to the N
delivered to 10,000 $m^2$ of savanna from ~100 years from atmospheric deposition (Mphepya et al.

2006).

If megacarcasses provide large nutrient pulses, then they likely create hotspots of
important below- and aboveground processes. Belowground, soil respiration and organic matter
decomposition might increase with nutrient inputs from carcasses (Risch et al. 2020).
Concentrations of C, N, P, and potassium (K) are often elevated near carcasses of medium-sized
animals (e.g., bison, moose, kangaroo, vicuña) (Towne, 2000; Bump et al. 2009a; Macdonald et
al. 2014; Risch et al. 2020; Monk et al. 2024), and nutrients such as P and calcium (Ca) continue
leaching from bones even after soft tissues have been consumed or degraded (Coe, 1978; Keenan
& Beeler, 2023). Aboveground, plant growth in African savannas is limited by nutrient
availability, most commonly N and P (Ries & Shugart, 2008; Pellegrini, 2016), and
micronutrients such as sodium (Na) and potassium (K) may co-limit plant productivity as well
(Epron et al. 2012; Chen et al. 2024). Thus, the large influx of nutrients released from
megacarcasses might increase the mobilization of nutrients by plants, potentially increasing
nutrient accessibility for vertebrate and invertebrate herbivores (Yang, 2008; Grant & Scholes,
2006; Anderson et al. 2010; Joern et al. 2012). Indeed, carcasses of smaller vertebrates (e.g.,
salmon, deer) can increase the proportions of nitrogen and $\delta^{15}N$ (an indicator of animal-driven N)
in plants within just a few months postmortem (Hocking & Reynolds, 2012; van Klink et al.

2020).

To assess the effects of megacarcasses on local nutrient pools (Figure 1), we measured

the initial contributions of elephant carcasses (1-26 months postmortem) to soil and plant
nutrients in the Kruger National Park (KNP), South Africa. Further, we examined the effects of
elephant carcasses on the two main soil types in KNP: sandy, relatively nutrient-poor granitic
soils and clayey, relatively nutrient-rich basaltic soils (Venter et al. 2003). At each site, we ran
transects in each cardinal direction from the center of the site where an elephant died, collecting
samples of soil and a palatable grass species (*Urochloa trichopus*) at 0, 2.5, 5, 10, and 15 m. We
then analyzed soil samples for CNP and micronutrient content, quantified soil microbial
respiration potential, and measured %N, $\delta^{15}N$, and macro- and micronutrient content in grass
tissue. We hypothesized that: (H1) elephant megacarcass decomposition would release nutrients
into soil, resulting in higher concentrations of soil N, P, and micronutrients near the center of
carcass sites; (H2) C inputs to the soil would stimulate microbial activity, resulting in increased
soil respiration potential near the center of carcass sites; and (H3) carcass-derived nutrients
would move from soil into plants, resulting in higher foliar nutrient concentrations near the
center of carcass sites. We predicted that enrichment effects from megacarcasses would be
greater on sites with fresher carcasses relative to older carcasses and on nutrient-poor granitic
sites compared to nutrient-rich basaltic sites.

**Sect. 2 Methods**

**2.1 Study system and sample collection**

We performed this research in the southern part of the Kruger National Park (KNP), South Africa (24.996 S, 31.592 E, ~275m elevation). The two dominant soil types in KNP are granitic soils (Inceptisols) and basaltic soils (Versitols or Andisols) (Khomo et al. 2017). The clay-rich basaltic soils have relatively large surface area, enabling them to retain larger quantities of water than granitic soils, which drain water more quickly and therefore are lower in water-soluble nutrients (Buitenweref, Kulmatiski, & Higgins, 2014; Rughöft et al. 2016). The landscape of KNP is a mix of savanna grasslands and broadleaf woodlands, with an overstory dominated by trees from the genus *Combretum* (red bushwillow, *C. apiculatum*; russet bushwillow, *C. hereroense*; leadwood, *C. imberbe*) and trees formerly known as acacias (knobthorn, *Senegalensis nigrescens*; umbrella thorn, *Vachellia tortillis*). The park hosts a full suite of African savanna animals, including ~30,000 elephants (*Loxodonta africana*) (Coetsee & Ferreira, 2023), with a mortality rate of ~2% (~600 elephants per year). The targeted region of KNP has a high density of scavengers and predators, including white-backed vultures (*Gyps africanus*), spotted hyenas (*Crocuta crocuta*), and lions (*Panthera leo*) (Owen-Smith & Mills, 2007).

During the wet season in March 2023, we identified ten elephant carcass sites (1-26 months postmortem), five on relatively nutrient-rich basaltic soil and five on nutrient-poor granitic soil. KNP section rangers provided precise GPS locations of where elephant carcasses had been found. Most elephants died of old age, illness, injury, or, in the case of one young bull, fighting over territory. Carcass sites were recognizable *in situ* by a persistent bonefield, undigested gut contents, and an absence of herbaceous vegetation. At each site, we hammered a rebar post into the center of the megacarcass disturbance and ran 15 m transects out from the

post in each of the four cardinal directions. We collected green leaf material from *U. trichopus,* a
common and abundant palatable grass species, and used an auger to collect soil samples to a
depth of 10 cm at five points along each transect (0.5, 2.5, 5, 10, and 15 m) (Bump, Peterson, &
Vucetich, 2009; Holdo & Mack, 2014; Gray & Bond, 2015; Monk et al. 2024). We treated the
10-15m distances as representative of background concentrations of nutrients based on pilot data
showing that the effect of elephant carcasses on soil nutrient concentrations was undetectable at
this distance away from the carcass site, similar to studies on the carcasses of other large
vertebrates (e.g., Towne, 2000; Bump et al. 2009). We pooled and homogenized the samples to
yield one composite leaf and one composite soil sample per sampling distance from each carcass
site. Soil samples were sieved in a 5-mm metal sieve which was cleaned in between samples
with 70% ethanol. Soil samples were stored in a cooler during fieldwork. On the day they were
collected, we used 5 g of each soil sample for soil respiration measurements (described below).
The rest of each sample was placed in a plastic bag on the day of collection and stored in a -20°C
freezer for up to 1 month; samples were stored in coolers with ice blocks during the transition
from the freezer at the field site to the lab for analysis. We chose to freeze samples rather than
storing at room temperature based on literature demonstrating that the impacts of freezing on soil
nitrate and ammonium concentrations are fairly minimal, except in specific cases of high soil
acidity or peaty soils that were not present at our field site (Esala, 1995; Turner & Romero, 2009;
Sollen-Norrlin & Rintoul-Hynes, 2024). Leaf samples were stored in paper bags at room
temperature until dried for analyses (see below).

**2.3 Hypothesis testing**

165 We tested our first hypothesis that elephant carcass decomposition would release nutrients into

166 the soil by performing soil nutrient analyses. We sent 250 g of each soil sample to Eco-Analytica

167 laboratory at the North-West University in Potchefstroom, South Africa for measurements of soil

168 concentrations of ammonium $[NH_4]^+$, nitrate $[NO_3]^-$, phosphate $[PO_4]^{3-}$, and plant-available P.

169 Samples were air-dried and sieved through < 2mm mesh prior to chemical analysis. Plant-

170 available P was extracted from 4 g of soil and 30 ml extraction fluid (1:7.5 ratio) using an acid–

171 fluoride solution (P Bray-1) (FAO, 2021), measured colorimetrically using a Systea

172 EasyChem200 analyser, and expressed as mg/kg. The detection limit was 0.5 mg/kg, and plant

173 available P measurements <0.5 mg/kg were replaced with half the detection limit (0.25 mg/kg)

174 (Croghan & Egeghy, 2003). Water-soluble nitrate and phosphate anions were extracted from

175 volume on volume 100 mg soil and 200 ml deionized water (Sonneveld & van den Ende, 1981),

176 analyzed by ion chromatography on a Metrohm 930 Compact Flex System, and measured as

177 mg/L. Ammonium (also 1:2 water extract) was analyzed colorimetrically using a Systea

178 EasyChem200 analyzer and measured as mg/L. Detection limits for soil ions were 0.01 mg/L,

179 and soil ion concentrations measured as <0.01 mg/L were replaced with half the detection limit

180 (0.005 mg/L). To convert the nitrate, ammonium, and phosphate units from mg/L to mg/kg, we

181 multiplied by 2, based on the 1:2 soil to water extraction ratio.

182  To determine whether soil micronutrients were distinct and elevated at the center of

183 carcass sites relative to soil further from the center, we measured concentrations of sodium (Na),

184 magnesium (Mg), iron (Fe), calcium (Ca), potassium (K), and phosphorus (P). Air-dried and

185 sieved (>2 mm) soil samples, weighed to 0.2 g, were microwaved in 9 ml 65% nitric acid

186 ($HNO_3$) and 3 ml 32% hydrochloric acid (HCl) according to EPA 3051b in a Milestone, Ethos

187 microwave digester with UP, Maxi 44 rotor. A period of 20 minutes allowed the system to reach

1800 MW at a temperature of 200 °C which was maintained for 15 minutes. After cooling, the
samples were brought up to a final volume of 50 ml and analyzed on an Agilent 7500 CE ICP-
MS fitted with CRC (Collision Reaction Cell) technology for interference removal. The
instrument is optimized using a solution containing Li, Y, Ce, and Tl (1 ppb) for standard low-
oxide/low interference levels ($\leq 1.5\%$) while maintaining high sensitivity across the mass range.
The instrument was calibrated using ULTRASPEC® certified custom mixed multi-element stock
standard solutions containing all the elements of interest (De Bruyn Spectroscopic Solutions,
South Africa). Calibrations spanned the range of 0 – 30 ppm for the mineral elements Ca, Mg,
Na, and K and 0 – 0.3 ppm for the rest of the trace elements. Elemental concentrations were
expressed as mg/kg.

Finally, to determine whether elevated N levels in soils were derived from the carcass, we

sent 10 g of each sample to the BIOGRIP laboratory within the Central Analytical Facility at
Stellenbosch University for measurements of soil %N and $\delta^{15}N$, obtained using a Vario Isotope
Select Elemental Analyzer connected to a thermal conductivity detector and an Isoprime
precisions isotope ratio mass spectrometer (IRMS). Samples were oven-dried at 60°C for 48
hours and milled to a fine powder using a Retsch MM400 mill (Germany). The powdered
samples were weighed (2 – 60 mg) prior to combustion at 950°C. The gasses were reduced to $N_2$
(undiluted) in the reduction column, which was held at 600°C. A high organic carbon (HOC) soil
standard ($0.52 \pm 0.02$ %N), along with two international reference standards (USGS40 ($\delta^{15}N$ -
4.52% AIR) and USGS41 ($\delta^{15}N$ +47.57% AIR)) were used for calibration. The N elemental
content was expressed relative to atmospheric N as $N_2$ $\delta^{15}N_{AIR}$ (‰). The quantification limit for
$\delta^{15}N$ on the IRMS is 1 nA (nanoAmp), and the quantification limit for %N is 0.06%. The
precision for %N was 0.02% and for $\delta^{15}N$ is ±0.11%, determined using the HOC standard, which
was run multiple times throughout the analysis.
To test our second hypothesis that nutrient inputs to the soil would stimulate microbial
activity, we measured soil organic C, water content, and microbial respiration potential. We sent
10 g of each sample to the BIOGRIP laboratory for measurements of soil organic C using a
Vario TOC Cube (Elementar, Germany). Samples (dried and milled as above) were weighed (10
– 60 mg), acidified using 10% HCl to remove the total inorganic C (carbonates), and dried
overnight at 60°C. All samples were analyzed through combustion at 950°C. The released $CO_2$
was measured by a non-dispersive infrared (NDIR) sensor. A high organic C (7.45 ± 0.14 %C)
soil standard from Elemental Microanalysis Ltd (UK) was included during the analysis. The
quantification limit for %C is 0.14%. The precision for the %C was 0.09% and was determined
using the low organic C (LOC) standard (1.86 ± 0.14 %C), which was run multiple times
throughout the analysis.
To quantify soil respiration and water content, we used an incubation method (Lemoine
et al. 2023) in which 5 g (± 0.2 g) of each sample was placed into a 100 ml clear glass bottle,
sealed, and flushed with $CO_2$-free air. Following flushing, we incubated the bottles for one hour
at 25°C. We then recorded $CO_2$ concentrations using an LI-850 $CO_2$/$H_2O$ infrared gas analyzer.
After soil respiration measurements, we determined sample dry weight by drying each sample at
60°C for 24-48 hours until stable mass was achieved. We subtracted dry weight from starting
weight to obtain soil water content. Finally, we used the dry weights and the Ideal Gas Law to
standardize all respiration measurements to $CO_2$ $\mu g$ $h^{-1}g$ dry $soil^{-1}$.
To test our third hypothesis that carcass-derived nutrients would be incorporated by
plants, we measured foliar nutrient concentrations in *U. trichopus*. Two grams of each dried leaf
sample was sent to the BIOGRIP laboratory for preparation and measurements of %N and $\delta^{15}N$
via stable isotope analysis as described above. A Sorghum flour standard (1.47 ± 0.25 %N) from
Elemental Microanalysis Ltd (UK) was used for calibration, along with two international
reference standards (USGS40 and USGS41). The quantification limit for $\delta^{15}N$ on the IRMS is 1
nA, and the quantification limit for %N is 1.3%. The precision for the %N was 0.02% and for
$\delta^{15}N$ is ±0.08‰. Limits were determined using the sorghum flour standard, which was run
multiple times throughout the analysis. Additionally, we sent 5 g per sample to Cedara
Analytical Services Laboratory to quantify micronutrients in grass tissue (P, Na, Mg, K, Ca, and
Fe) using Inductively Coupled Plasma Optical Emission Spectroscopy (ICP-OES 5800, Agilent,
USA). Samples were dried (110°C overnight) and milled to a fine powder. Subsamples (0.5 g)
were ashed at 450°C for 4 hours, and the ash was re-wet using 2 mL conc. HCl (32%). Samples
were evaporated to dryness then re-suspended in 25 mL 1M HCl before filtering. Lastly, the
filtrate was diluted with de-ionized water in a ratio of 5:20 filtrate to water. To calibrate the ICP-
OES, solutions containing known amounts of each element were measured (10-20 ppm for Na
and C, 200-1500 ppm for Fe, 0.5-3.75% for K, and 0.125-0.5% for P), prepared from 1000 ppm
primary single standards. At three of the ten sites, we did not find sufficient plant material at the
central point for analysis, resulting in a sample size of N = 7 for the center (distance = 0.5m)
measurement for leaf nutrient analyses.

To test whether each response variable for the three hypotheses was significantly

associated with soil type and/or distance from the carcass center, we performed a model selection
procedure. For each response variable, we ran five generalized linear mixed models using the
gamma family (link = log) in the package *lme4* (Bates et al. 2015): (*i*) soil type + distance + soil
type × distance interaction, (*ii*) soil type + distance, (*iii*) soil type, (*iv*) distance, and (v) a null
model indicating no significant difference in slope or intercept after accounting for carcass site.
All models included carcass site as a random effect to account for individual variation. Each
model included 50 observations (10 sites x 5 distances per site). For samples in which the
nutrient level was listed as 0 or undetectable, we accounted for the uncertainty by using half the
detection level as described above. The narrow distribution of ages (1-26 months since death)
with the sample size of N = 10 sites made testing for the effect of age challenging, so we did not
include carcass age in the models. We compared the models for each response variable using
Akaike Information Criterion (AICc). Models with a $\Delta$AICc $\leq$ 2 were considered roughly
equivalent in fit (Burnham and Anderson, 2002).

In addition to these models, for our second hypothesis we regressed soil respiration

potential against soil organic C, expecting that the two would be positively correlated. We ran a
generalized linear mixed model with soil respiration potential as the response variable. The
model included soil organic C + distance + soil type, with carcass site as a random effect. We did
not include an interaction with soil type in this model due to sample size restrictions. Respiration
potential and organic C were both log-transformed to achieve normality.

To determine whether leaf and soil micronutrient composition differed with distance and

soil type, we ran permutational analysis of variance (perMANOVA) in *vegan* (Oksanen et al.
2022). We ran the same model separately for soil and leaf micronutrient composition (soil type +
distance). To determine which micronutrients contributed most to compositional differences
across distances and soil types, we calculated samplewise Bray-Curtis dissimilarity and
performed principal component analysis. We also tested for differences in variance in
micronutrient composition across distances and soil types using "betadisper" in *vegan* (Oksanen
et al. 2022). We ran linear models to test for correlations between leaf and soil concentrations of
each micronutrient. Each model included distance as a covariate and site as a random effect.

Finally, to test the impact of carcass age on key soil metrics, we ran exponential decay

functions for soil ammonium, nitrate, phosphate, and respiration verses carcass age for samples
from the center of the carcass site (0.5m sampling location). We also performed a t-test to verify
that there was no difference in mean carcass age across soil types.

All statistical analyses were performed in R version 4.2.1 (R Core Team, 2022).


**Sect. 3 Results**
**3.1 Hypothesis 1: Effects of megacarcasses on soil nutrient pools**
We found partial support for our first hypothesis that soil N and P concentrations would be
higher closer to the center of carcass sites (Table S1). Soil %N (Figure 2A) was overall greater in
basaltic soils, and it decreased with distance from the carcass site on granitic soils. Soil nitrate
(Figure 2B) decreased with distance from the carcass site but did not differ between soil types.
Ammonium (Figure 2C) also decreased with distance, but only in granitic soils. $\delta^{15}N$ (Figure 2D)
was greater in granitic soils and decreased with distance in both soil types, indicating that the
proportion of animal-sourced N in the soil was greater near the center of the carcass site. Soil
phosphate, plant available P, and mineral P (Figure 2E-G) all exhibited significant soil × distance
interactions. Phosphate (Figure 2E) was highly elevated at the center of carcass sites and
decreased steeply with distance, but only in granitic soils. Plant-available P (Figure 2F)
decreased with distance in both soil types, but the effect was strongest in granitic soils. Finally,
mineral P (Figure 2G) was greater in basaltic soils, and there was a small decrease with distance
in granitic soils but not in basaltic soils.
Contrary to our first hypothesis, soil micronutrient composition did not differ
significantly with distance from the carcass center; nor did most individual micronutrients (Table
S1). The perMANOVA results showed that soil micronutrient composition did not differ
significantly with distance ($R^2 = 0.00$, $F_{4,44} = 0.0$, $P = 1.000$) (Figure S2A), but it did differ
significantly with soil type ($R^2 = 0.71$, $F_{1,44} = 108.8$, $P = 0.001$) (Figure S2B). There was no
significant difference in variance with distance ($F_{4,45} = 0.0$, $P = 0.996$) or soil type ($F_{1,48} = 2.6$, $P$
$= 0.115$). Principal components analysis showed that dimension 1 explained 53.6% of the
variation between soil types and was driven primarily by differences in Mg, Ca, and Fe.
Dimension 2 explained 25.9% of variation and was driven primarily by differences in K. Soil Na
(Figure S3A) was marginally greater in granitic soils and decreased with distance from the
carcass, with the effect greater in granitic soils. Soil K (Figure S3B) was greater in basaltic soils
and decreased marginally with distance. Soil Fe, Mg, and Ca (Figure S3C-E) were greater in
basaltic soils, with minimal effects of distance.

**3.2 Hypothesis 2: Effects of megacarcasses on soil carbon and respiration**
Consistent with our second hypothesis, soil respiration potential was marginally positively
correlated with soil organic carbon concentration and decreased significantly with distance but
did not differ with soil type (Figure 3). We found no evidence for differences in soil water
content (Figure S4A) or soil pH (Figure S4B) with distance or soil type. In both cases, the null
ranked among the set of top models (Table S1).

**3.3 Hypothesis 3: Effects of megacarcasses on plant nutrient pools**
Consistent with our third hypothesis, we found elevated foliar nutrient concentrations in *U.*
*trichopus* at elephant carcass sites. Leaf %N (Figure 4A) and $\delta^{15}N$ (Figure 4B) both decreased
with distance from the carcass center. $\delta^{15}N$ exhibited a significant soil × distance interaction in
which it was overall greater in basaltic soils, but the difference between the two soil types was
greater closer to the carcass site. Foliar P was greater in basaltic soils and decreased only
marginally with distance in the granite soils. Finally, the foliar N:P ratio was greater in granitic
soils and decreased with distance in the basaltic soils.

Leaf micronutrient composition did not differ significantly with distance ($R^2 = 0.13$, $F_{4,40}$

$= 1.9$, $P = 0.062$; Figure S5A) but did differ with soil type ($R^2 = 0.17$, $F_{1,40} = 9.7$, $P = 0.001$;
Figure S5B). There was no significant difference in variance with distance ($F_{4,41} = 0.5$, $P =$
$0.713$) or soil type ($F_{1,44} = 1.9$, $P = 0.173$). Dimension 1 explained 42.8% of the variance across
soil types and was primarily driven by Mg, Na, and P. Dimension 2 explained 26.6% of the
variance and was driven mainly by K, Ca, and Fe. Foliar Na (Figure S6A) and Mg (Figure S6B)
were both greater in basaltic soils and decreased with distance from the carcass center. Foliar K
(Figure S6C) and Fe (Figure S6D) both decreased with distance as well but did not differ with
soil type. The null model was in the top set for foliar Ca, indicating no significant relationship
between foliar Ca concentrations and soil type or distance from the carcass center. Individual
micronutrients (K, Ca, Mg, Fe) were not correlated between leaf and soil samples, with the
exception of Na (Table S3).

**3.4 Effects of carcass age on soil ions and respiration potential**
Soil ammonium ($\alpha = 0.018$, $P < 0.001$), phosphate ($\alpha = 0.023$, $P < 0.001$), and respiration
potential ($\alpha = 0.058$, $P < 0.001$) all decreased significantly with carcass age (Figure 5A-C). The
exponential decay model for nitrate failed to converge due to an outlier with extremely high soil
nitrate (1454 mg/kg) at 258 days postmortem (Figure 5D). We ran a t-test to test for a difference
in mean carcass age between soil types and found no significant difference between the two
groups ($P = 0.294$).

**Sect. 4 Discussion**
Here, we show that elephant megacarcasses influence soil and foliar nutrients during at least the
first two years following mortality. Consistent with our hypotheses, soil nitrate (Figure 2B),
ammonium (Figure 2C), $\delta^{15}N$ (Figure 2D), and P (Figure 2E-F) concentrations were all elevated
at the center of carcass sites and decreased with distance from the center. Soil %N, nitrate,
ammonium, and plant-available P concentrations at the 15m point were consistent with those
found in other studies of soil nutrient content in Kruger (Aranibar et al. 2003; Rughöft et al.
2016), confirming that the 15m point serves as an effective baseline control in this experiment.
Microbial respiration potential was also elevated towards the center of carcass sites and was
correlated with the abundance of organic C (Figure 3). Finally, %N (Figure 4A) and $\delta^{15}N$ in a
common grass (Figure 4B) were both elevated closer to the centers of carcass sites compared to
grass farther from carcasses. Together, these results indicate that carcass-derived nutrients move
into soil and subsequently get absorbed by plants over relatively short time scales, cycling
essential nutrients such as N from carrion into the soil and then back into aboveground nutrient
pools.

The initial influx of ammonium from elephant carcasses may have time-dependent

impacts on plant abundance at elephant carcass sites. The mean ammonium level at the center of
carcass sites (34.8 mg/kg) was higher than the level generally considered toxic to plants (Britto
& Kronzucker, 2002). Yet, we found living grass—typically *U. trichopus*—in the center of the
carcass site at seven out of ten of our sites (ammonium range 10-172 mg/kg) and at the 2.5m
distance for all sites (ammonium range 0-72 mg/kg). The three sites without vegetation in the
center had the highest ammonium levels (70-144 mg/kg), suggesting that *U. trichopus* has a
higher degree of ammonium tolerance than some sympatric grass species but may still be limited
by the high ammonium levels at the centers of these three relatively fresh carcass sites. However,
the recentness of the disturbance from the carcass likely also plays a role in determining plant
abundance near the center of the carcass. Because of the elephant carcass site age distribution,
(mean 350 days postmortem; range 24-811 days), this study may not have captured the full
impact of ammonium release from carcasses during the early stages of decomposition. Soil
ammonium spiked early and decreased rapidly (Figure 5A), and future research on carcasses
within the first few weeks postmortem would enhance our understanding of these early nutrient
dynamics.

Soil nitrate (Figure 2B) and soil respiration potential (Figure 3) were also elevated near

the center of carcass sites, indicating higher activity rates for nitrifying bacteria and heterotrophic
microbes (Prosser, 2011). These results are consistent with other work on carrion, where
microbial activity tends to be greater in soils near carcasses as compared to surrounding soil
(Bump et al. 2009b). However, carcass effects on soil microbial respiration exhibit a high degree
of intra-system variation (Risch et al. 2020), and the potentially short window during which
increased respiration occurs may make capturing these variations challenging. For example, soil
respiration potential at the center of the three youngest carcass sites was on average 2x higher
than the seven older sites (18.43 and 9.62 µg $CO_2$ per hour per gram of dry soil, respectively;
Figure 5D). Thus, the impact of increased organic C on soil microbial processes may be
relatively short lived and only last a matter of months (Keenan et al. 2018; Keenan, Schaeffer, &
DeBruyn, 2019). These trends are consistent with soil ammonium and phosphate, both of which
are highest at the youngest carcass sites (<200 days postmortem; Figure 5A-B). Soil microbial
respiration rate is also highly elevated early on, but it decreases at a faster rate over time than soil
ions (Figure 5C). Thus, soil dynamics during the first several months after death may play a
crucial role in determining the long-term impacts of megacarcasses on savannas and therefore
provides a promising avenue for future research.
Elevated soil phosphate (Figure 2E) and plant-available P (Figure 2F) at the center of
carcass sites were also consistent with expectations from the literature (Bump et al. 2009a;
Parmenter & MacMahon, 2009). However, elevated P levels in soil did not translate to elevated
P in grass leaves (Figure 4C), which could suggest a lag between trends in soil and plants that is
longer for P than for N. This lag could occur because phosphate easily forms chemical bonds
with other soil ions (e.g., iron and aluminum in acidic soils and calcium in basic soils). Nitrate
does not form these bonds and therefore has greater water solubility and mobility in soils and
may be more readily taken up by plants (Wiersum, 1962; Arai & Sparks, 2007). However, it is
also possible that P limitation in Kruger is not as strong as it is in some other African savanna
systems (Pellegrini, 2016). The foliar N:P ratios measured in this experiment were higher closer
to the center of the carcass site (median 9.38 at 0 m and 4.83 at 15 m), indicating that N
limitation may be relatively stronger further from the carcass site, and P limitation may be
relatively stronger closer to the center (Figure 4D, Table S2). These relatively high foliar N:P
ratios at the center of carcass sites are similar to those found in N fertilization studies in Kruger
(Craine et al. 2008), further supporting the idea that the influx of N from megacarcasses may
shift the soil from relatively more N limited to more P limited.
The elevated plant-available P at the center of carcass sites likely came primarily from
phosphate released from decomposing tissue (Yong et al. 2019). Bone decomposition, which is
also likely a major source of P from animal carcasses (Subalusky et al. 2020), occurs over long
time scales (Coe, 1978; Subalusky et al. 2020) and therefore should result in the slow release of
P and a gradual decrease in the N:P ratio (Parmenter & MacMahon, 2009; Quaggiotto et al.
2019). Indeed, initial inorganic N influxes to the Mara River in Kenya from mass wildebeest die-
offs are 10-fold greater than concurrent increases in P, which instead releases slowly over about
seven years of bone decomposition (Subalusky et al. 2017). Research following megacarcasses
over longer timeframes postmortem is needed to clarify when P from enriched soil is absorbed
by plants and at what stage megacarcass bones begin contributing to soil P dynamics. It is also
possible that bone dispersal by scavengers may result in less P leaching from bones close to
where the elephant died and more P being distributed across the landscape at distances far from
the carcass site.
The contributions of megacarcasses to soil nutrient pools were strongly associated with
soil type. Our results confirmed the previously-established trend that basaltic soils are overall
more cation rich than granitic soils, with greater concentrations of P, K, Fe, Mg, and Ca (Figure
2G; Figure S3B-E; Gertenbach, 1983; Craine, Morrow, & Stock, 2008; Wigley et al. 2014).
However, soil ammonium, $\delta^{15}N$, and phosphate were all higher in the granitic soils towards the
center of carcass sites, decreasing steeply to be similar to basaltic soils about 10 m from the
carcass center (Figure 2C-E). These results indicate that the impact of organic matter from
megacarcasses may be stronger in relatively nutrient-poor and sandy granitic soil compared with
nutrient-rich and clayey basaltic soil. We were surprised that grass on basaltic soil did not
consistently exhibit greater nutrient concentrations. One potential explanation is that grass may
primarily be limited by macronutrients like N and P on both soil types (Craine et al. 2008;
Holdo, 2013) rather than by micronutrients. Thus, even with increased micronutrient availability
their actual uptake may not differ substantially. Studies on ungulate carcasses (e.g., muskoxen,
moose, zebra) have shown increased foliar N at carcass sites (Danell et al. 2002; Bump et al.
2009b; Turner et al. 2014), but to date there is little research on the flow of micronutrients from
carrion to plants and none on the pipeline from megacarcasses to plants. Moreover, it remains to
be seen whether increases in foliar N and other nutrients affect herbivory rates at carcass sites
and how long such effects may last.

The magnitude of nitrogen inputs from megacarcasses, as well as the substantial size and

duration of their impact zones, means their impacts on ecosystem processes may be functionally
distinct from smaller carrion. Soil nitrate concentrations at elephant carcass sites are orders of
magnitude higher than at carcass sites of smaller carrion (e.g., rabbits, white-tailed deer,
kangaroo) (Quaggiato et al. 2019; Bump et al. 2009; Barton et al. 2016). Even for large ungulates
such as moose, total soil inorganic nitrogen (ammonium + nitrate) at carcass sites is a mean 300
mg/kg (Bump, Peterson, & Vucetich, 2009), substantially lower than the mean total soil
inorganic nitrogen at elephant carcass sites (2.5m distance; 473 mg/kg). Termite mounds,
another long-lasting source of savanna nutrient heterogeneity, have mean soil nitrate
concentrations (197 mg/kg) lower than elephant carcass sites, but maximum nitrate
concentrations that are on par with them (974 mg/kg) (Seymour et al. 2014), again indicating that
elephant carcasses are one of the strongest known individual contributors of soil nitrogen in
African savanna ecosystems, which may have important implications for savanna ecology.
Indeed, there is evidence that carcass size strongly impacts scavenger food web structure
(Moleón et al. 2015; Morris et al. 2023). Moreover, the attraction of animals to carcasses via
scavenging, predation, or mourning (Goldenberg & Wittemyer, 2020) could have positive
feedbacks on nutrient cycling (Bump, Peterson, & Vucetich, 2009; Monk et al. 2024), which
may be magnified by carcass size. Thus, the impacts of megacarcasses on savanna ecosystem
processes may be dissimilar to the effects of small carrion and more similar to other more
persistent contributors to savanna ecosystem processes, such as termite mounds (Davies et al.
2016), cattle bomas (Augustine, 2003), and even mass animal mortality events (Subalusky et al.

2017, 2020).


**Sect. 5 Conclusions**
This study is an important first step in understanding the ecological legacies of megacarcasses on
savanna ecosystem processes. During the first two years postmortem, elephant carcasses released
pulses of nitrogen and phosphate, which influence savanna primary productivity. These nutrients
stimulated soil microbial activity and enriched foliar N, and the effects were strongest in
nutrient-poor soil, with potential long-term impacts on savanna nutrient heterogeneity. These
carcass-derived nutrient hotspots represent a previously unstudied function of megaherbivores on
savannas—one that we need to better understand in order to comprehend the full impacts of
megaherbivore population declines in modern ecosystems.

**Data and Code Availability:** Data and computer code are archived on Dryad Digital Repository
(https://doi.org/10.5061/dryad.wpzgmsbwm).

**Author Contributions:** Deron E. Burkepile, Nathan P. Lemoine, Izak P. J. Smit, Tercia
Strydom, Aimee Tallian, Johan T. du Toit, Dave I. Thompson, and Joshua P. Schimel conceived
the study. Michelle L. Budny, Johan T. du Toit, Nathan P. Lemoine, Joshua P. Schimel, Izak P.
J. Smit, Tercia Strydom, Aimee Tallian, Dave I. Thompson, Helga van Coller, and Deron E.
Burkepile collected samples. Courtney G. Reed, Nathan P. Lemoine, Dave I. Thompson, and
Deron E. Burkepile analyzed the data. Courtney G. Reed drafted the manuscript, and all authors
contributed to editing.

**Competing Interests:** The authors declare that they have no conflict of interest.

**Acknowledgments:** Funding for this research was provided by the National Science Foundation
(#s 2128092, 2128093, and 2128094) and the University of California Santa Barbara Academic
Senate. All research was completed under permits from South African National Parks (SS554).
We thank the field assistants of SANParks for guiding and protection in the field, as well as the
section rangers and Sandra Snelling for GPS locations and ages of carcasses. We thank Mr.
Lucky Sithole (Cedara Analytical Services Laboratory), Ms. Terina Vermeulen (Eco-Analytica
Laboratory), and Dr. Janine Colling (BIOGRIP laboratory) for extensive laboratory support.
Select data used in this research paper were generated using equipment in the DSI funded
BIOGRIP soil and water node at Stellenbosch University.

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

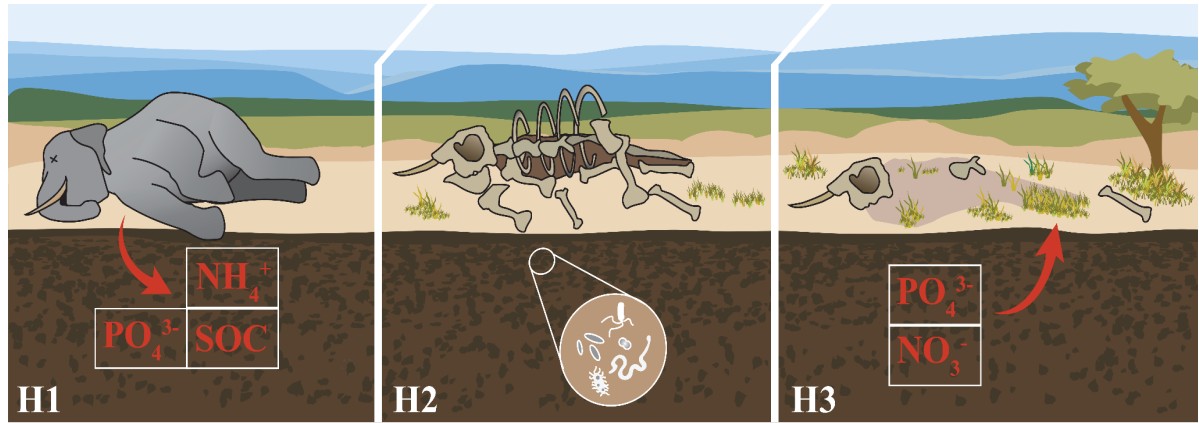

**Figure 1.** Hypothesized impacts of elephant megacarcasses on soil and plant nutrients. First (H1), we hypothesized that elephant carcasses would release pulses of nutrients into the soil, resulting in higher concentrations of soil ions such as nitrogen (ammonium, $[NH_4]^+$), phosphorus (phosphate, $[PO_4]^{3-}$), and soil organic C. Second (H2), we hypothesized that C inputs from the carcass would result in increased soil microbial respiration potential. Third (H3), we hypothesized that plants would take up nutrients from the carcass soil, resulting in plants with distinct nutrient profiles and increased concentrations of key limiting nutrients such as N and P. Image credit: Kirsten Boeh.

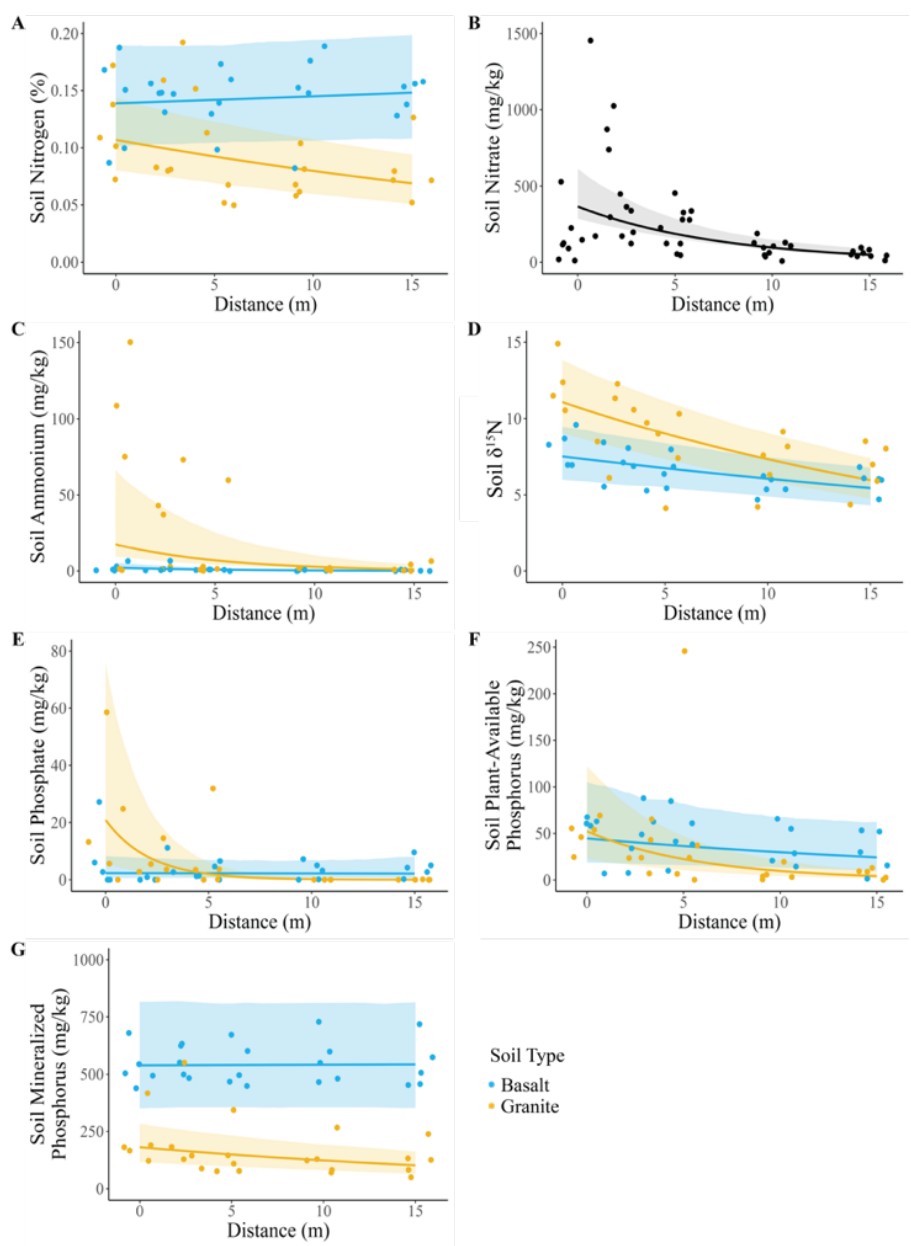

770

**Figure 2.** Effects of elephant carcasses on soil N and P concentrations in granitic and basaltic

soils. (A) Soil N (%) was greater in basaltic soils, and in granitic soils it decreased with distance

from the carcass site. (B) Soil nitrate nitrogen decreased with distance but did not differ between

soil types. (C) Soil ammonium nitrogen and (D) $\delta^{15}N$ were both greater in granitic soils and

decreased with distance from the carcass. (E) Soil phosphate, (F) plant-available P, and (G)

mineralized P decreased with distance in granitic soils but not basaltic soils. Points represent

individual measurements from soil samples taken at 0, 2.5, 5, 10, and 15m and are offset to be
visible when they would otherwise overlap. Lines show predictions calculated from the top
generalized linear mixed model, which may include soil type, distance, and soil type by distance
interaction as covariates (Table S1). Only significant relationships are shown on plots. Shading
indicates the 95% confidence interval.

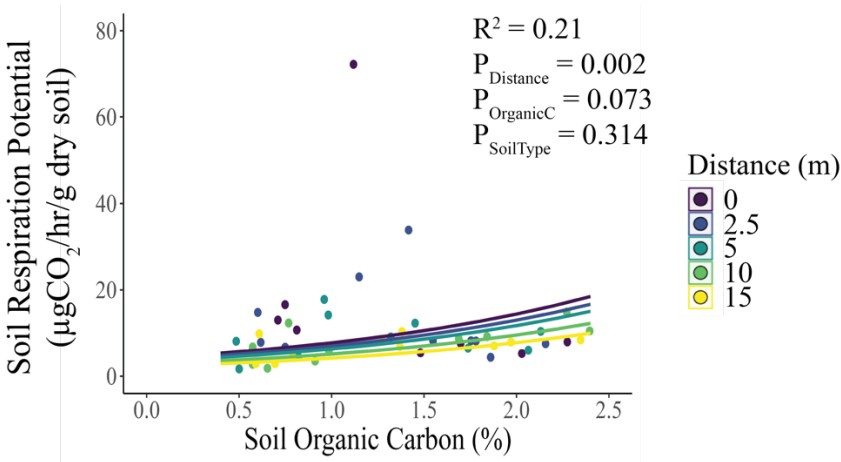


**Figure 3.** Effects of elephant carcasses on soil respiration potential. We regressed soil respiration
against soil organic carbon, with distance and soil type as covariates. Soil respiration potential
was marginally positively correlated with soil organic C (%) and decreased significantly with
distance from the carcass. Points represent individual measurements taken from soil samples at
0, 2.5, 5, 10, and 15m and are offset to be visible when they would otherwise overlap. Lines
represent model predictions. Only significant relationships are shown.

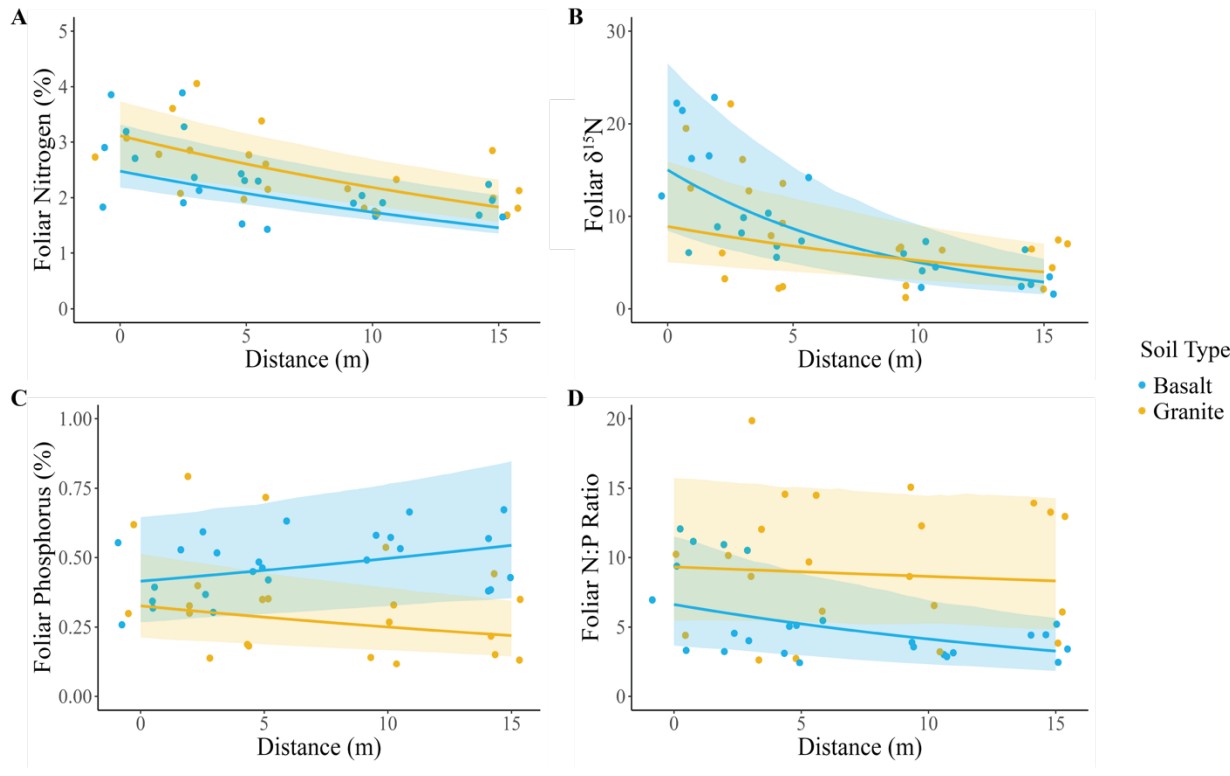


**Figure 4.** Effects of elephant carcasses on foliar N and P concentrations in granitic and basaltic

soils. (A) Foliar %N was higher in granitic soils and decreased with distance from the carcass

center. (B) Foliar $\delta^{15}N$ decreased with distance from the carcass center and exhibited a

significant interaction in which $\delta^{15}N$ decreased more rapidly with distance in basaltic soils. (C)

Foliar P was greater in basaltic soils and decreased with distance in granitic soils. (D) Foliar N:P

ratio was greater in granitic soils and decreased with distance from the carcass center for both

soil types. Points represent individual measurements from soil samples taken at 0, 2.5, 5, 10, and

15m and are offset to be visible when they would otherwise overlap. Lines show predictions

calculated from the top generalized linear mixed model, which may include soil type, distance,

and soil type by distance interaction as covariates (Table S2). Only significant relationships are

shown on plots. Shading indicates the 95% confidence interval. Three of the ten sites had bare

ground at the 0 m distance, resulting in a sample size of 7 sites for that distance and 10 for the

other distances.

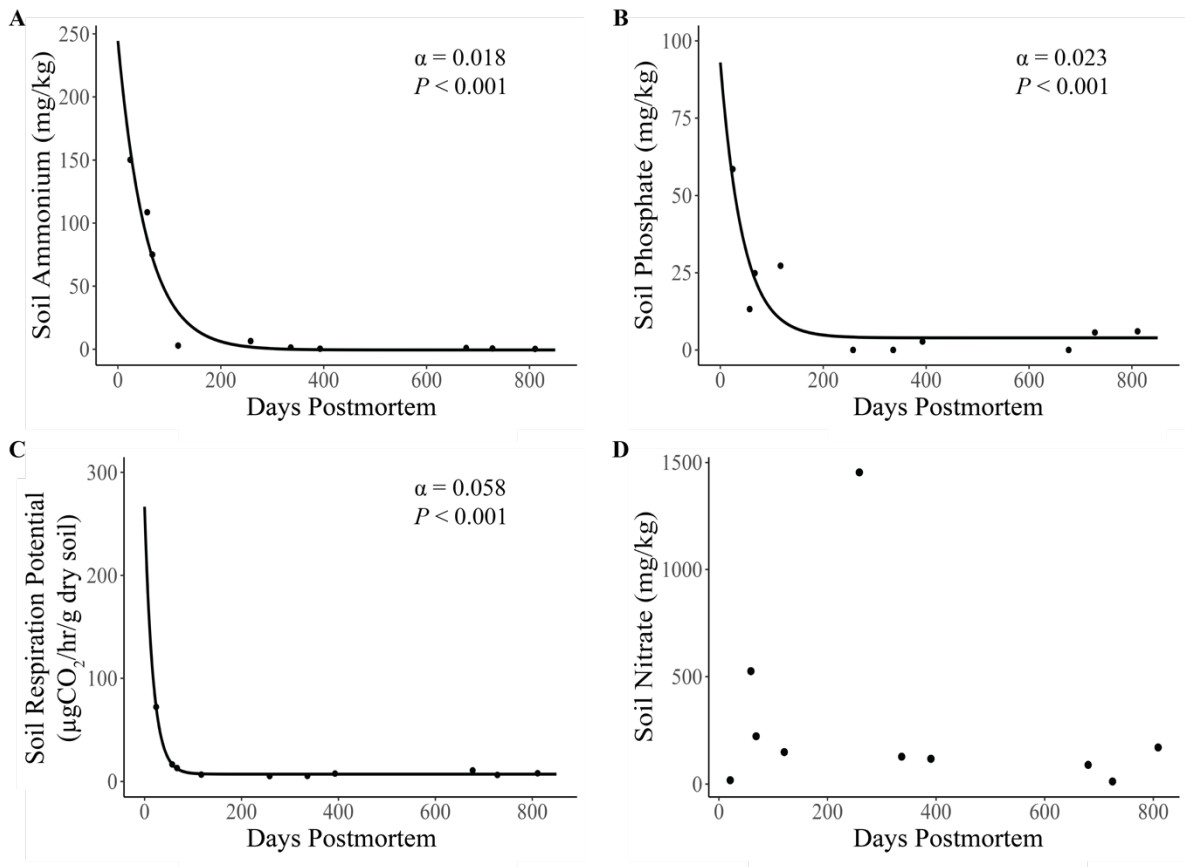

803

**Figure 5.** Relationship between carcass age and key soil metrics (soil ion concentrations and respiration potential). Lines represent predictions from exponential decay models, with α equal to the rate of decay. (A) Soil ammonium, (B) phosphate, and (C) respiration potential all decreased significantly with carcass age. The model for (D) soil nitrate failed to converge. Points represent individual measurements taken at the center of the carcass site (distance = 0.5m). Only significant relationships are shown on plots.