# Peer review of "Title: Elephant megacarcasses increase local nutrient pools in African savanna soils and plants"

_EGUsphere, 2024_

## Author Comment (AC2)

**Response to Feedback from Reviewer 1**

**Summary**

Animals impact elemental cycling in many direct and indirect ways. Evidence from several biomes demonstrates that even after death, animal carcasses can change the biogeochemistry of ecosystems and these impacts can be long lasting. Most studies of carcass impacts on ecosystems, however, are done on small to medium (1kg to 200kg) sized animals. In this contribution, the authors investigate the effects of elephant megacarcasses on the biogeochemistry of soils and plants. The authors report significant effects of elephant carcasses on components of soil and plant elemental cycling and they discuss how these effects may be important components of spatiotemporal heterogeneity in ecosystems.

**General comments**

1) Overall, I found the writing good. The authors have crafted a nice narrative that makes a compelling case that megacarcasses can be important parts of ecosystems and therefore we need to learn more about the impacts of these carcasses on ecosystems.

AUTHORS' RESPONSE: Thank you! We appreciate your kind words and thoughtful review.

2) I have a few questions about the analysis. The effective sample size is 10. Obviously, it is hard to find carcasses (I would have great difficulty in finding 10 fresh moose carcasses in my system!) but the authors are trying to squeeze a lot of information out of very few data points. I have the following specific questions about the analysis:

i) While I like the transect approach, the design may have been stronger if the authors had random transects (ie, transects with no known carcass) like Risch et al. work. This would strengthen inference.

AUTHORS' RESPONSE: Thanks for the suggestion, and we definitely appreciate the value of control/random sites. In fact, our original plan was to use random transects as controls (Risch et al. 2020), but during a pilot experiment we realized that high landscape heterogeneity (differences in hill slope, vegetation, water drainage, proximity to termite mounds, etc.), all of which have implications for nutrient distribution across the landscape (Venter et al. 2003; Holdo & McDowell, 2004), made the random transects challenging for interpretation as controls. Instead, we looked at our pilot data to see whether there was a consistent size of the impact site and found that soil nutrients were elevated until about 5-8m away from the center of the carcass site. Past this 5-8m radius, soil nutrients dropped to consistently lower levels, indicative of background concentrations. Thus, we designed the sampling scheme of 0.5m, 2.5m, 5m, 10m, and 15m distances away from the carcass site to capture both the impact of the elephant carcass and the background ("control") concentration of soil nutrients (at the 10m and 15m distance). There was never a significant difference in nutrient concentrations between the 10 and 15m distances, suggesting our sampling scheme successfully captured the transition from the influence of the elephant carcass through to the background level of nutrients in the matrix soils.

We have updated the methods section of the manuscript as follows: "Based on pilot data, we treat the 10-15m distances as controls, sine the high degree of landscape heterogeneity in the system (e.g., differences in hill slope, vegetation, water drainage, proximity to termite mounds) made random transects difficult for interpretation."

Venter FJ, Scholes RJ, Eckhardt HC. Abiotic template and its associated vegetation pattern. In: JT Du Toit, KH Rogers, HC Biggs, eds. The Kruger experience: ecology and management of savanna heterogeneity. Washington, DC, USA: Island Press, 83–129, 2003.

Holdo, R. M. & McDowell, L. R. Termite mounds as nutrient-rich food patches for elephants. Biotropica, 36, 231-239, https://doi.org/10.1111/j.1744-7429.2004.tb00314.x, 2004.

ii) lines 180-183. The author's approach to checking for normality of response data does not seem sound to me. The assumption of normality (for linear models) is normality in light of the model, i.e., investigating the normality of residuals is a more common approach to this. Either way, it is often better to avoid transforming the data and generalized linear models do allow for a lot of flexibility to fit different error distributions. For example, the gamma family in glm is very flexible and can handle log-normal data sets. Did the authors try different families of error distributions before transforming their data?

AUTHORS' RESPONSE: Thanks so much for this suggestion. We have revised our analysis and implemented the gamma family (link = log) in for all of our models now instead of log-transforming and have updated the text in the methods accordingly. We have updated all results, and new versions of all tables and figures (including supplemental) are appended at the end of this document for reference. Even with this change in the structure of the analyses, the major patterns across the different analyses did not change. In fact, these changes actually strengthen the major patterns in the results showing the importance of elephant carcasses in savanna nutrient dynamics.

iii) lines 187-189. How many data points did the authors have per estimated parameter in the most complex model here?

AUTHORS' RESPONSE: For each of these models, we had 50 observations total (10 sites x 5 samples per site). In our most complicated model, that averages to ~17 observations per parameter, which is above the recommended 10 observations per parameter (Burnham & Anderson, 2002).

iv) line 194. This is fine but I think Burham & Anderson would say that any model within deltaAIC of 2 of the null model should not be considered to be supported. In several cases, the authors interpret top models that are ranked above the null but within deltaAIC of 2 of the null as supported (e.g., lines 217-218, 218-221).

AUTHORS' RESPONSE: With the updated model structure (see response to 2.iii), there are now only three response variables (soil water, soil pH, and foliar calcium) for which the null and

another model fall within a ΔAICc value of 2 (Table S1). In all three cases, we will interpret this as the results not supporting a relationship between the response variable and soil type, distance from the carcass, or soil type by distance interaction.

v) what $R^2$ are the authors reporting? In the captions of Tables S1 and S2 (thank you for providing full AIC and coefficient tables), the authors state "$R^2$ is the proportion of variance explained by a model". This is unclear. These are mixed models, and the most common approach is to report the marginal $R^2$ and conditional $R^2$. Is the $R^2$ in these tables one of those or another pseudo $R^2$? This is critical for many reasons but most importantly, given the small sample size and large number of mixed-models, I would expect at least one of the models to not converge. There are many indicators when a mixed-model does not converge and one of the best is when the marginal $R^2$ = conditional $R^2$. Without having both of these pieces of information, the reader is unable to adequately assess the fit of the models. Other indicators of models not converging are coefficients estimates or errors that are very large or very small (i.e., 0 – see next comment).

AUTHORS' RESPONSE: We have updated supplemental tables (see below) to include both marginal and conditional $R^2$ values. The only place where we had issues with model non-convergence was soil phosphate; two of the five models failed to converge and are indicated in Table S1.

vi) I am confused by the magnitude of Table S2 sodium and iron coefficients and/or the scale of reported on the y-axis of Figure 5 for these. The iron coefficients in Table S2 seem small relative to the Figure 5c? Or am I misreading things?

AUTHORS' RESPONSE: The original models for soil and leaf micronutrients used log-transformed data, which meant that the coefficients and standard errors were in log units as well. When plotting, we back-transformed the data to make the axis scales easier to interpret, which is why the values in the table and the figures were different. The updated models (see response to 2.iii) still use a log link, so the model outputs in the updated tables are still in log units as well. Again in this version, we exponentiated when calculating the prediction lines so that we could plot them with the raw data, which we believe is visually more intuitive than figures with axes on the log scale. We have updated the table captions for clarification as follows: "Coefficients (± standard error) are shown for each predictor and model and are in log units."

3) the reporting of results could be improved. I recommend, the authors report: top ranked models (AIC + measure of independent fit like $R^2$). Then report effect size or relationships (coefficients). I found key statistics to be missing throughout. Statements like "Phosphate concentrations were greater in granitic soils…" would be more informative if they included the coefficient + error in parenthesis. Coefficients can be reported for the top-ranked model or from model averaged results when there are several competing models.

AUTHORS' RESPONSE: We appreciate this feedback and have updated the tables accordingly, including AIC values, marginal and conditional $R^2$, and coefficients + standard error (see below). In the text of the results section, we will include the coefficient + standard error from the topranked model in parentheses. We agree that this will make the interpretation of the results much more straightforward for the reader.

4) in section 3.2 I think the reader may be more interested in coefficients and confidence intervals around those relationships than p-values that are currently reported.

AUTHORS' RESPONSE: We will replace the p-values in this section with coefficients + standard error and agree this will help greatly with interpretation.

**Specific comments:**

5) I found the use of three different terms that mean similar things (nutrient flows, ecosystem processes, nutrient availability) in the introductory sentence confusing. I recommend the authors replace "nutrient availability" with "ecosystem processes" or "nutrient flows". Surely living animals (not just carcasses) influence nutrient availability (which is just a part of a continual nutrient cycle).

AUTHORS' RESPONSE: We will edit that line for consistency in phrasing: "Living animals affect nutrient flows through ecosystems (Schmitz et al. 2018), but we have only recently acknowledged that animal carcasses could also influence ecosystem processes."

6) line 83. I believe there is no "e" at the end of the citation Risch et al.

AUTHORS' RESPONSE: We have corrected the citation. Thanks for catching it!

7) lines 96-111. How do these elephants die? As someone with no experience with megacarcasses, I would appreciate some insight on the causes of death. Most large herbivore deaths in my empirical systems are from predation which I assume is not the case for elephants.

AUTHORS' RESPONSE: We received GPS coordinates for carcasses from KNP rangers, who also keep record of the cause of death for each elephant. The reviewer is right that predation tends not to be a major issue for elephants, and none that we know of died from it. Most of the elephants in our dataset died of natural causes such as old age, illness, injury, or, in the case of one young bull, a territorial dispute that ended in his death.

We have updated the methods section as follows: "Most elephants died of old age, illness, injury, or, in the case of one young bull, territorial fighting."

8) really excellent job with clear hypotheses and nice work carrying forward these hypotheses throughout the ms – really makes the job easier for the reader.

AUTHORS' RESPONSE: Thank you!

9) lines 132-133 Why 10cm deep core? Is that mineral soil only?

AUTHORS' RESPONSE: We used a 10cm core to ensure that we captured the soil surface horizon. It is a commonly used depth and is more conservative than shallower sampling. Prior work on the soil impacts of carcasses uses this depth (Bump, Peterson, & Vucetich, 2009; Monk et al. 2024). Moreover, previous work in the same system has shown that soil auger sampling depths of 7.5-10cm are sufficient for detecting differences in N, C, and soil micronutrients (Gray & Bond 2015, Holdo & Mack 2014). We will update the text in the methods to include these references.

Bump, J. K., Peterson, R. O., & Vucetich, J. A. Wolves modulate soil nutrient heterogeneity and foliar nitrogen by configuring the distribution of ungulate carcasses. Ecology, 90, 3159–3167, 2009.

Monk, J. D., Donadio, E., Smith, J. A., Perrig, P. L., Middleton, A. D., & Schmitz, O. J. Predation and Biophysical Context Control Long-Term Carcass Nutrient Inputs in an Andean Ecosystem. Ecosystems, 27, 346–359, 2024.

Gray, E. F. & Bond, W. 2015. Soil nutrients in an African forest/savanna mosaic: Drivers or driven? South African Journal of Botany, 101, 66-72. https://doi.org/10.1016/j.sajb.2015.06.003

Holdo, R. M. & Mack, M. C. 2014. Functional attributes of savanna soils: contrasting effects of tree canopies and herbivores on bulk density, nutrients and moisture dynamics. Journal of Ecology, 102, 1171-1182. https://doi.org/10.1111/1365-2745.12290

10) the discussion is well done – concise and touches on all hypotheses.

AUTHORS' RESPONSE: Thank you!

11) Figure 1 is an outstanding visual!

AUTHORS' RESPONSE: Thank you!

12) in figures 2-5 I recommend the authors consider reminding the reader of the sampling resolution because the jitter of points makes it impossible to see what distances were measured below 5m.

AUTHORS' RESPONSE: We have updated the captions in figures 2-5 to include sampling resolution as follows: "Points represent individual measurements taken at 0, 2.5, 5, 10, and 15m and are offset to be visible when they would otherwise overlap."

[Figure]

**Figure 1.** Hypothesized impacts of elephant megacarcasses on soil and plant nutrients. First (H1), we hypothesized that elephant carcasses would release pulses of nutrients into the soil, resulting in higher concentrations of soil nutrients such as nitrogen (ammonium, $[NH_4]^+$), phosphorus (phosphate, $[PO_4]^{3-}$), and soil organic C. Second (H2), we hypothesized that C inputs from the carcass would result in increased soil microbial respiration potential. Third (H3), we hypothesized that plants would take up nutrients from the carcass soil, resulting in plants with distinct nutrient profiles and increased concentrations of key limiting nutrients such as N and P. Image credit: Kirsten Boeh.

[Figure]

**Figure 2.** Soil N and P responses to elephant carcasses. (A) Soil N (%) was greater in basaltic soils, and in granitic soils it decreased with distance from the carcass site. (B) Soil nitrate nitrogen decreased with distance but did not differ with soil type. (C) Soil ammonium nitrogen and (D) $\delta^{15}N$ were both greater in granitic soils and decreased with distance from the carcass. (E) Soil phosphate, (F) plant-available P, and (G) mineralized P decreased with distance in granitic soils but not basaltic soils. Points represent individual measurements taken at 0, 2.5, 5, 10, and 15m and are offset to be visible when they would otherwise overlap. Lines show predictions calculated from the top model. Shading indicates the 95% confidence interval.

[Figure]

**Figure 3.** Soil respiration potential was marginally positively correlated with soil organic C (%) and decreased significantly with distance from the carcass. Points represent individual measurements taken at 0, 2.5, 5, 10, and 15m and are offset to be visible when they would otherwise overlap. Lines represent model predictions.

[Figure]

**Figure 4.** Foliar N and P responses to elephant carcasses. (A) Foliar %N and (B) $\delta^{15}$N both decreased with distance from the carcass center. (C) Foliar P was greater in basaltic soils and decreased with distance in granitic soils. (D) Foliar N:P ratio was greater in granitic soils and decreased with distance from the carcass center. Points represent individual measurements taken at 0, 2.5, 5, 10, and 15m and are offset to be visible when they would otherwise overlap. Lines show predictions calculated from the top model. Shading indicates the 95% confidence interval. Three of the ten sites had bare ground at the 0 m distance, resulting in a sample size of 7 sites for that distance and 10 for the other distances.

[Figure]

Relationship between carcass age and key metrics (soil ion concentrations and respiration potential). (A) Soil ammonium, (B) soil nitrate, (C) soil phosphate, and (D) soil respiration potential are all higher at fresher carcass sites. Point respresent values at the center of the carcass site (distance = 0-0.5m).

**Figure 5.** Relationship between carcass age and key soil metrics (soil ion concentrations and respiration potential). (A) Soil ammonium, (B) nitrate, (C) phosphate, and (D) respiration potential are all higher at fresher carcass sites. Points represent individual measurements taken at the center of the carcass site (distance = 0-0.5m).

**Table S1.** Generalized linear mixed model results for soil variables. The same five models were run for each response variable, including a null model, and each included site as a random effect to account for repeat measurements. AICc is Akaike's Information Criterion, and ΔAICc is the difference between a given model and the best fit model for that response variable. Cum.Wt stand for cumulative weight; it gives the sum of Akaike's weights and indicates the likelihood that the models up to that point are the best in the set. Models with a ΔAICc value of 2 are considered roughly equivalent in fit and are italicized. Marginal $R^2$ is the proportion of variance explained by both fixed and random effects in a model, and conditional $R^2$ is the proportion of variance explained by fixed effects. Coefficients (± standard error) are shown for each predictor and model and are in log units. Rows are organized in blocks by response variable. Within blocks, models are listed in order of increasing ΔAICc.

| Model | Model Fit | | | | | Coefficients ± *SE* | | |
|---|---|---|---|---|---|---|---|---|
| | AICc | ΔAICc | Cum.Wt | Mar. $R^2$ | Con. $R^2$ | Soil | Distance | Soil × Distance |
| **Nitrogen (%)** | | | | | | | | |
| *Soil × Distance* | *-227.32* | *0.00* | *0.99* | *0.54* | *0.74* | *-0.26 ± 0.22* | *0.00 ± 0.01* | *-0.03 ± 0.01* |
| Soil + Distance | -216.13 | 11.20 | 1.00 | 0.46 | 0.67 | -0.48 ± 0.21 | -0.01 ± 0.00 | |
| Distance | -214.95 | 12.37 | 1.00 | 0.04 | 0.52 | | -0.01 ± 0.00 | |
| Soil | -212.36 | 14.97 | 1.00 | 0.40 | 0.62 | -0.47 ± 0.21 | | |
| Null | -211.23 | 16.09 | 1.00 | | | | | |
| **δ15N** | | | | | | | | |
| *Soil × Distance* | *180.87* | *0.00* | *0.77* | *0.55* | *0.70* | *0.39 ± 0.16* | *-0.02 ± 0.01* | *-0.02 ± 0.01* |
| Soil + Distance | 184.66 | 3.79 | 0.88 | 0.50 | 0.66 | 0.26 ± 0.15 | -0.03 ± 0.00 | |
| Distance | 184.67 | 3.79 | 1.00 | 0.34 | 0.60 | | -0.03 ± 0.00 | |
| Soil | 219.35 | 38.47 | 1.00 | 0.20 | 0.34 | 0.28 ± 0.14 | | |
| Null | 219.96 | 39.09 | 1.00 | | | | | |
| **Nitrate (mg/kg)** | | | | | | | | |
| *Distance* | *624.84* | *0.00* | *0.70* | *0.48* | *0.52* | | *-0.14 ± 0.02* | |

| | | | | | | | | |
|---|---|---|---|---|---|---|---|---|
| Soil + Distance | 627.06 | 2.23 | 0.93 | 0.48 | 0.52 | -0.14 ± 0.27 | -0.14 ± 0.02 | |
| Soil × Distance | 629.51 | 4.67 | 1.00 | 0.48 | 0.52 | -0.24 ± 0.39 | -0.14 ± 0.03 | 0.02 ± 0.04 |
| Null | 649.77 | 24.93 | 1.00 | | | | | |
| Soil | 651.82 | 26.99 | 1.00 | 0.01 | 0.04 | -0.18 ± 0.31 | | |
| **Ammonium (mg/kg)** | | | | | | | | |
| *Soil + Distance* | *219.52* | *0.00* | *0.65* | *0.58* | *0.77* | *2.49 ± 0.66* | *-0.18 ± 0.03* | |
| *Soil × Distance* | *220.94* | *1.43* | *0.97* | *0.60* | *0.77* | *2.91 ± 0.73* | *-0.15 ± 0.04* | *-0.07 ± 0.06* |
| Distance | 225.87 | 6.35 | 1.00 | 0.21 | 0.77 | | -0.18 ± 0.02 | |
| Soil | 244.57 | 25.05 | 1.00 | 0.34 | 0.70 | 2.51 ± 0.76 | | |
| Null | 249.38 | 29.86 | 1.00 | | | | | |
| **Phosphate (mg/kg)** | | | | | | | | |
| *Soil × Distance* | *167.99* | *0.00* | *0.98* | *0.52* | *0.79* | *2.20 ± 0.96* | *0.00 ± 0.05* | *-0.46 ± 0.08* |
| Soil + Distance | 178.68 | 10.69 | 1.00 | 0.18 | 0.18 | -0.38 ± 0.70 | -0.14 ± 0.06 | |
| Null | 180.65 | 12.66 | 1.00 | | | | | |
| Soil | Model did not converge | | | | | | | |
| Distance | Model did not converge | | | | | | | |
| **Plant Available Phosphorus (mg/kg)** | | | | | | | | |
| *Soil × Distance* | *447.18* | *0.00* | *0.94* | *0.34* | *0.63* | *0.16 ± 0.62* | *-0.04 ± 0.03* | *-0.13 ± 0.04* |
| Distance | 453.68 | 6.50 | 0.98 | 0.20 | 0.55 | | -0.10 ± 0.02 | |
| Soil + Distance | 454.80 | 7.62 | 1.00 | 0.26 | 0.55 | -0.66 ± 0.55 | -0.11 ± 0.02 | |
| Null | 467.35 | 20.17 | 1.00 | | | | | |
| Soil | 469.19 | 22.01 | 1.00 | 0.03 | 0.30 | -0.35 ± 0.47 | | |
| **Mineral Phosphorus (mg/kg)** | | | | | | | | |
| *Soil × Distance* | *537.77* | *0.00* | *1.00* | *0.86* | *0.95* | *-1.09 ± 0.32* | *0.00 ± 0.00* | *-0.04 ± 0.01* |
| Soil + Distance | 560.48 | 22.71 | 1.00 | 0.82 | 0.92 | -1.35 ± 0.31 | -0.02 ± 0.00 | |
| Distance | 566.38 | 28.61 | 1.00 | 0.04 | 0.76 | | -0.02 ± 0.00 | |
| Soil | 573.55 | 35.78 | 1.00 | 0.78 | 0.89 | -1.33 ± 0.31 | | |
| Null | 579.62 | 41.85 | 1.00 | | | | | |
| **Sodium (mg/kg)** | | | | | | | | |
| *Soil × Distance* | *438.56* | *0.00* | *0.73* | *0.29* | *0.59* | *0.22 ± 0.35* | *-0.03 ± 0.01* | *-0.04 ± 0.02* |
| Distance | 441.09 | 2.53 | 0.94 | 0.22 | 0.54 | | -0.05 ± 0.00 | |
| Soil + Distance | 443.53 | 4.97 | 1.00 | 0.22 | 0.54 | -0.06 ± 0.35 | -0.05 ± 0.01 | |
| Null | 464.02 | 25.45 | 1.00 | | | | | |

| | | | | | | | | |
|---|---|---|---|---|---|---|---|---|
| Soil | 466.38 | 27.82 | 1.00 | 0.00 | 0.34 | 0.00 ± 0.00 | | |
| **Potassium (mg/kg)** | | | | | | | | |
| *Soil × Distance* | *676.07* | *0.00* | *0.94* | *0.29* | *0.81* | *-0.23 ± 0.42* | *0.01 ± 0.00* | *-0.02 ± 0.01* |
| Null | 682.93 | 6.86 | 0.97 | | | | | |
| Soil | 684.55 | 8.48 | 0.99 | 0.25 | 0.78 | -0.37 ± 0.41 | | |
| Distance | 685.17 | 9.10 | 1.00 | 0.00 | 0.72 | | 0.00 ± 0.00 | |
| Soil + Distance | 686.89 | 10.82 | 1.00 | 0.26 | 0.78 | -0.37 ± 0.41 | 0.00 ± 0.00 | |
| **Calcium (mg/kg)** | | | | | | | | |
| *Soil* | *749.09* | *0.00* | *0.60* | *0.82* | *0.94* | *-1.45 ± 0.41* | | |
| *Soil + Distance* | *751.01* | *1.92* | *0.83* | *0.82* | *0.94* | *-1.45 ± 0.01* | *0.00 ± 0.00* | |
| Soil × Distance | 753.00 | 3.91 | 0.91 | 0.82 | 0.94 | -1.42 ± 0.41 | 0.00 ± 0.01 | -0.01 ± 0.01 |
| Null | 753.55 | 4.46 | 0.97 | | | | | |
| Distance | 755.37 | 6.27 | 1.00 | 0.00 | 0.81 | | 0.00 ± 0.00 | |
| **Iron (mg/kg)** | | | | | | | | |
| *Soil* | *914.44* | *0.00* | *0.67* | *0.88* | *0.96* | *-1.22 ± 0.28* | | |
| Soil + Distance | 916.83 | 2.39 | 0.87 | 0.88 | 0.96 | -1.22 ± 0.28 | 0.00 ± 0.00 | |
| Soil × Distance | 918.54 | 4.10 | 0.95 | 0.88 | 0.96 | -1.19 ± 0.28 | 0.00 ± 0.00 | 0.00 ± 0.01 |
| Null | 920.27 | 5.83 | 0.99 | | | | | |
| Distance | 922.55 | 8.11 | 1.00 | 0.00 | 0.82 | | 0.00 ± 0.00 | |
| **Magnesium (mg/kg)** | | | | | | | | |
| *Soil* | *700.88* | *0.00* | *0.63* | *0.87* | *0.96* | *-1.53 ± 0.37* | | |
| Soil + Distance | 703.33 | 2.45 | 0.81 | 0.87 | 0.96 | -1.53 ± 0.37 | 0.00 ± 0.00 | |
| Soil × Distance | 703.97 | 3.09 | 0.95 | 0.88 | 0.96 | -1.48 ± 0.37 | 0.00 ± 0.00 | -0.01 ± 0.01 |
| Null | 706.40 | 5.52 | 0.99 | | | | | |
| Distance | 708.75 | 7.87 | 1.00 | 0.00 | 0.84 | | 0.00 ± 0.00 | |
| **Water (mmol/mol)** | | | | | | | | |
| *Null* | *111.87* | *0.00* | *0.32* | | | | | |
| *Distance* | *112.09* | *0.22* | *0.61* | *0.03* | *0.38* | | *0.02 ± 0.01* | |
| *Soil* | *112.92* | *1.05* | *0.80* | *0.12* | *0.40* | *0.45 ± 0.38* | | |
| *Soil + Distance* | *113.27* | *1.40* | *0.96* | *0.14* | *0.42* | *0.45 ± 0.38* | *0.02 ± 0.01* | |
| Soil × Distance | 115.86 | 3.99 | 1.00 | 0.14 | 0.42 | 0.44 ± 0.42 | 0.02 ± 0.02 | 0.00 ± 0.03 |
| **pH** | | | | | | | | |
| *Soil × Distance* | *55.04* | *0.00* | *0.37* | *0.07* | *0.44* | *0.05 ± 0.07* | *0.00 ± 0.00* | *-0.01 ± 0.00* |
| *Null* | *55.26* | *0.22* | *0.71* | | | | | |

| Distance | 56.94 | 1.90 | 0.86 | 0.01 | 0.38 | | 0.00 ± 0.00 | |
| Soil | 57.63 | 2.59 | 0.96 | 0.00 | 0.37 | 0.00 ± 0.07 | | |
| Soil + Distance | 59.41 | 4.37 | 1.00 | 0.01 | 0.38 | 0.00 ± 0.00 | 0.00 ± 0.00 | |

**Table S2.** Generalized linear mixed model results for leaf variables. The same five models were run for each response variable, including a null model, and each included site as a random effect to account for repeat measurements. AICc is Akaike's Information Criterion, and ΔAICc is the difference between a given model and the best fit model for that response variable. Cum.Wt stand for cumulative weight; it gives the sum of Akaike's weights and indicates the likelihood that the models up to that point are the best in the set. Models with a ΔAICc value of 2 are considered roughly equivalent in fit and are italicized. Marginal $R^2$ is the proportion of variance explained by both fixed and random effects in a model, and conditional $R^2$ is the proportion of variance explained by fixed effects. Coefficients (± standard error) are shown for each predictor and model and are in log units. Rows are organized in blocks by response variable. Within blocks, models are listed in order of increasing ΔAICc.

| Model | Model Fit | | | | | Coefficients ± *SE* | | |
|---|---|---|---|---|---|---|---|---|
| | AICc | ΔAICc | Cum.Wt | Mar. $R^2$ | Con. $R^2$ | Soil | Distance | Soil × Distance |
| **Nitrogen (%)** | | | | | | | | |
| *Distance* | *56.12* | *0.00* | *0.64* | *0.40* | *0.60* | | *-0.03 ± 0.00* | |
| Soil + Distance | 57.79 | 1.67 | 0.92 | 0.43 | 0.61 | 0.13 ± 0.14 | -0.03 ± 0.00 | |
| Soil × Distance | 60.33 | 4.20 | 1.00 | 0.43 | 0.61 | 0.15 ± 0.15 | -0.03 ± 0.01 | 0.00 ± 0.01 |
| Null | 89.78 | 33.66 | 1.00 | | | | | |
| Soil | 91.66 | 35.53 | 1.00 | 0.03 | 0.21 | 0.10 ± 0.13 | | |
| **δ15N** | | | | | | | | |
| *Soil × Distance* | *229.95* | *0.00* | *0.95* | *0.51* | *0.77* | *-0.52 ± 0.43* | *-0.11 ± 0.01* | *0.06 ± 0.02* |
| Distance | 236.55 | 6.60 | 0.99 | 0.44 | 0.70 | | -0.08 ± 0.01 | |
| Soil + Distance | 238.97 | 9.02 | 1.00 | 0.45 | 0.70 | -0.12 ± 0.40 | -0.08 ± 0.01 | |
| Null | 282.45 | 52.50 | 1.00 | | | | | |
| Soil | 284.30 | 54.34 | 1.00 | 0.04 | 0.36 | -0.30 ± 0.41 | | |
| **Phosphorus (%)** | | | | | | | | |
| *Soil × Distance* | -87.04 | 0.00 | 0.99 | 0.47 | 0.75 | -0.24 ± 0.31 | 0.02 ± 0.01 | -0.04 ± 0.01 |
| Soil | -76.10 | 10.94 | 1.00 | 0.38 | 0.68 | -0.55 ± 0.31 | | |
| Null | -75.98 | 11.06 | 1.00 | | | | | |

| | | | | | | | | |
|---|---|---|---|---|---|---|---|---|
| Soil + Distance | -73.69 | 13.34 | 1.00 | 0.38 | 0.68 | -0.55 ± 0.31 | 0.00 ± 0.01 | |
| Distance | -73.68 | 13.36 | 1.00 | 0.00 | 0.56 | | 0.00 ± 0.01 | |
| **N:P Ratio** | | | | | | | | |
| *Soil × Distance* | 209.64 | 0.00 | 0.86 | 0.41 | 0.71 | 0.34 ± 0.38 | -0.05 ± 0.01 | 0.04 ± 0.01 |
| Distance | 214.60 | 4.96 | 0.94 | 0.09 | 0.59 | | -0.03 ± 0.01 | |
| Soil + Distance | 214.85 | 5.21 | 1.00 | 0.36 | 0.67 | 0.62 ± 0.01 | -0.03 ± 0.00 | |
| Null | 225.74 | 16.10 | 1.00 | | | | | |
| Soil | 226.21 | 16.57 | 1.00 | 0.23 | 0.57 | 0.55 ± 0.37 | | |
| **Sodium (mg/kg)** | | | | | | | | |
| *Soil + Distance* | *839.97* | *0.00* | *0.60* | *0.62* | *0.78* | *-0.99 ± 0.32* | *-0.03 ± 0.01* | |
| *Soil × Distance* | *841.56* | *1.59* | *0.88* | *0.62* | *0.79* | *-0.88 ± 0.34* | *-0.03 ± 0.01* | *-0.02 ± 0.01* |
| Distance | 843.18 | 3.21 | 1.00 | 0.09 | 0.64 | | -0.03 ± 0.01 | |
| Soil | 852.98 | 13.02 | 1.00 | 0.53 | 0.71 | -1.00 ± 0.32 | | |
| Null | 856.49 | 16.52 | 1.00 | | | | | |
| **Magnesium (mg/kg)** | | | | | | | | |
| *Soil × Distance* | *722.20* | *0.00* | *0.99* | *0.45* | *0.80* | *-0.20 ± 0.28* | *0.00 ± 0.00* | *-0.02 ± 0.01* |
| Distance | 731.74 | 9.54 | 0.99 | 0.07 | 0.66 | | -0.01 ± 0.00 | |
| Soil + Distance | 732.78 | 10.58 | 1.00 | 0.39 | 0.76 | -0.36 ± 0.28 | -0.01 ± 0.00 | |
| Null | 743.56 | 21.36 | 1.00 | | | | | |
| Soil | 744.46 | 22.26 | 1.00 | 0.31 | 0.69 | -0.37 ± 0.28 | | |
| **Potassium (mg/kg)** | | | | | | | | |
| *Distance* | *936.99* | *0.00* | *0.73* | *0.20* | *0.57* | | *-0.03 ± 0.00* | |
| Soil + Distance | 939.50 | 2.51 | 0.94 | 0.20 | 0.57 | 0.02 ± 0.25 | -0.03 ± 0.00 | |
| Soil × Distance | 941.96 | 4.97 | 1.00 | 0.20 | 0.57 | 0.05 ± 0.26 | -0.02 ± 0.01 | 0.00 ± 0.01 |
| Null | 956.55 | 19.57 | 1.00 | | | | | |
| Soil | 958.95 | 21.96 | 1.00 | 0.00 | 0.38 | 0.00 ± 0.24 | | |
| **Calcium (mg/kg)** | | | | | | | | |
| *Null* | *799.64* | *0.00* | *0.42* | | | | | |
| *Distance* | *800.68* | *1.04* | *0.67* | *0.01* | *0.50* | | *0.00 ± 0.00* | |
| *Soil* | *801.22* | *1.58* | *0.86* | *0.14* | *0.53* | *-0.20 ± 0.21* | | |
| Soil + Distance | 802.36 | 2.72 | 0.96 | 0.14 | 0.54 | -0.20 ± 0.21 | 0.00 ± 0.00 | |
| Soil × Distance | 804.45 | 4.81 | 1.00 | 0.15 | 0.54 | -0.16 ± 0.22 | 0.01 ± 0.01 | -0.01 ± 0.01 |
| **Iron (mg/kg)** | | | | | | | | |
| *Distance* | *591.87* | *0.00* | *0.69* | *0.21* | *0.57* | | *-0.08 ± 0.01* | |

| | | | | | | | |
|---|---|---|---|---|---|---|---|
| Soil + Distance | 594.14 | 2.27 | 0.92 | 0.23 | 0.58 | -0.26 ± 0.50 | -0.08 ± 0.01 | |
| Soil × Distance | 596.15 | 4.27 | 1.00 | 0.23 | 0.59 | -0.09 ± 0.39 | -0.07 ± 0.00 | -0.02 ± 0.02 |
| Null | 616.95 | 25.08 | 1.00 | | | | | |
| Soil | 619.06 | 27.19 | 1.00 | 0.02 | 0.48 | -0.31 ± 0.00 | | |

**Table S3.** Generalized linear mixed model results testing for correlations between leaf and soil micronutrients. The same model was run for each of five micronutrients (Na, K, Ca, Mg, and Fe) with leaf micronutrient concentration as the response variable, soil micronutrient + distance as the main effects, and site as a random effect. Marginal $R^2$ is the proportion of variance explained by both fixed and random effects in a model, and conditional $R^2$ is the proportion of variance explained by fixed effects. Coefficients (± standard error) are shown for each predictor and model.

| Leaf Micronutrient | Mar. $R^2$ | Con. $R^2$ | Soil Micronutrient Coefficient ± *SE* | Distance Coefficient ± *SE* |
|---|---|---|---|---|
| Sodium | 0.08 | 0.82 | 11.56 ± 11.67 | -146.47 ± 43.04 |
| Potassium | 0.29 | 0.73 | 0.00 ± 0.00 | -0.06 ± 0.01 |
| Calcium | 0.12 | 0.58 | 0.00 ± 0.00 | 0.00 ± 0.00 |
| Magnesium | 0.17 | 0.79 | 0.00 ± 0.00 | 0.00 ± 0.00 |
| Iron | 0.11 | 0.32 | 0.00 ± 0.01 | -52.85 ± 20.57 |

**Revised Supplemental Figures**

[Figure]

**Figure S1.** Representative photos of two elephant carcass sites of different ages and soil types. (A) The first site is 67 days post-death and is on granitic soil. (B) The second site is 811 days post-death and is on basaltic soil. In both images, there is a visible impact zone with reduced vegetation coverage. At the first site, elephant bones have all been dispersed, though some are still present at the second site. Photos taken by Deron Burkepile at time of sample collection in March 2023.

[Figure]

**Figure S2.** (A) Soil micronutrient composition did not differ significantly with distance from the carcass but (B) was distinct in different soil types.

[Figure]

**Figure S3.** Effects of elephant carcasses on soil micronutrients. (A) Soil sodium decreased significantly with distance from the carcass. (B) Potassium decreased with distance but only in granitic soils. (C) Iron, (D) magnesium, and (E) calcium were greater in basaltic soils. Distance appeared in the top model for calcium, but the effect size was minimal. Points represent individual measurements taken at 0, 2.5, 5, 10, and 15m and are offset to be visible when they would otherwise overlap. Lines show predictions calculated from the top model. Shading indicates the 95% confidence interval.

[Figure]

**Figure S4.** Neither (A) soil water nor (B) soil pH differed with distance or soil type. Points represent individual measurements taken at 0, 2.5, 5, 10, and 15m and are offset to be visible when they would otherwise overlap.

[Figure]

**Figure S5.** (A) Foliar micronutrient composition did not differ significantly with distance from the carcass but (B) was distinct in different soil types.

[Figure]

**Figure S6**. Effects of elephant carcasses on grass foliar micronutrients. (A) Foliar Na and (B) Mg were greatest in basaltic soil and decreased significantly with distance. (C) Foliar K and (D) Fe decreased with distance but did not differ with soil type. (E) Foliar Ca did not differ with distance or soil type. Points represent individual measurements taken at 0, 2.5, 5, 10, and 15m and are offset to be visible when they would otherwise overlap. Lines show predictions calculated from the top model. Shading indicates the 95% confidence interval.

---

## Author Comment (AC3)

**Response to Feedback from Reviewer 2**

Summary:

Reed and coauthors present a well-written study on the impacts of megacarcasses (elephants) to soil biogeochemistry after up to 2 years of decomposition. The authors examined 10 carcass hotspots with 5 carcasses each on two different soil types. They quantified soil major and trace element chemistry as well as plants associated with the hotspots to determine if carcasses influenced soil N and P chemistry and if those elements were subsequently enriched in vegetation. The current version of this manuscript does not adequately describe the methods in enough detail to make the work reproduceable. Additionally, the handling of the data for statistical analyses is strange and non-standard. The discussion needs to be re-written to better emphasize the importance of the work (as framed in the introduction). I think this work has potential to be an important contribution, but there needs to be some major revisions.

AUTHORS' RESPONSE: Thanks for all of your feedback. We have made substantial changes in response to your suggestions, including rewriting the methods section to include more details on the lab analyses. We have also updated our statistical analyses to use the gamma family of generalized linear mixed models, which allows us to run non-normally distributed data without the log transformation. These changes add to the methodological clarity and statistical robustness of this research, and they actually strengthen the major patterns in the results showing the importance of elephant carcasses in savanna nutrient dynamics. Finally, we have updated the text to ensure that the functional distinctiveness of megacarcass relative to smaller carcasses carries through from the introduction to the discussion. All major updates are appended to the end of this document, including methods section 2.3, supplemental tables, and main and supplemental figures.

General comments:

- The importance of this study in adding to our knowledge about nutrient transfer at carrion hotspots is not emphasized clearly in the discussion. The introduction frames how megacarcasses may be "functionally different than smaller carcasses" but never returns to this aspect in the discussion, which is really where this work could add to our knowledge. Adding more to the discussion would help address this issue and would make the impact of the work clearer.

  AUTHORS' RESPONSE: Thanks for the suggestion. We will update the discussion to more clearly link our results to the functional differences between megacarcasses and smaller carrion that we bring up in the introduction. The magnitude of nutrient inputs from megacarcasses, as well as the substantial size and duration of their impact zones, may have important impacts on ecosystem processes that are not seen at smaller carcass sites. For example, we will discuss the potential for positive feedbacks via increased herbivory (Bump, Peterson, & Vucetich, 2009) and predation (Monk et al. 2024) as well as the potential impacts on savanna nutrient heterogeneity (Barton et al. 2013).

• Parts of the results belong in the discussion, and I've tried to highlight those below in specific comments.

AUTHORS' RESPONSE: Thanks for pointing this out. We have edited and/or removed those sentences from the results section and focused on them in the discussion, as described in response to specific comments below.

• The methods need significantly more specific details, highlighted in specific comments.

AUTHORS' RESPONSE: We have substantially updated the methods section in the manuscript (section 2.3) to include specific details on the lab analysis methods. The full updated text for this section is appended to the end of this document (changes are in blue font), and we address specific comments below as well.

• Additionally, there were no control soil or plant samples examined here. Please describe in the methods why there were no controls.

AUTHORS' RESPONSE: The reviewer raises an important point here, and we definitely appreciate the value of control/random sites that other studies of carrion have used. In fact, our original plan was to use random transects as controls (Risch et al. 2020), but during a pilot experiment we realized that high landscape heterogeneity (differences in hill slope, vegetation, water drainage, proximity to termite mounds, etc.), all of which have implications for nutrient distribution across the landscape (Venter et al. 2003; Holdo & McDowell, 2004), made the random transects challenging for interpretation as controls. Instead, we looked at our pilot data to see whether there was a consistent size of the impact site and found that soil nutrients were elevated until about 5-8m away from the center of the carcass site. Past this 5-8m radius, soil nutrients dropped to consistently lower levels, indicative of background concentrations. Thus, we designed the sampling scheme of 0.5m, 2.5m, 5m, 10m, and 15m distances away from the carcass site to capture both the impact of the elephant carcass and the background ("control") concentration of soil nutrients (at the 10m and 15m distance). There was never a significant difference in nutrient concentrations between the 10 and 15m distances, suggesting our sampling scheme successfully captured the transition from the influence of the elephant carcass through to the background level of nutrients in the matrix soils.

We have updated the methods section of the manuscript as follows: "Based on pilot data, we treat the 10-15m distances as controls, sine the high degree of landscape heterogeneity in the system (e.g., differences in hill slope, vegetation, water drainage, proximity to termite mounds) made random transects difficult for interpretation."

Venter FJ, Scholes RJ, Eckhardt HC. Abiotic template and its associated vegetation pattern. In: JT Du Toit, KH Rogers, HC Biggs, eds. The Kruger experience: ecology and management of savanna heterogeneity. Washington, DC, USA: Island Press, 83–129, 2003.

Holdo, R. M. & McDowell, L. R. Termite mounds as nutrient-rich food patches for elephants. Biotropica, 36, 231-239, https://doi.org/10.1111/j.1744-7429.2004.tb00314.x, 2004.

The handling of the data for statistical analyses is non-standard and not clearly justified. If data were non-normally distributed (it seems like some datasets were and some were not), why not just use a non-parametric statistical test rather than log-transforming the data? It is a bit strange to log-transform some data but not all. The approach of adding 0.001 to zero values is also not correct (described below in specific comments).

AUTHORS' RESPONSE: We have updated our model selection procedure to use the gamma family with a log link for our generalized linear mixed models rather than transforming the data beforehand (see RC1, 2.ii).

We have re-run the analyses using 0.005 mg/L as the replacement value for any zeros in the soil ion concentration data, as that is half of the detection limit (0.10 mg/L). The tables with these updated results are appended at the end of this document, but this update did not result in any changes to statistical significance or model performance in the results. We have updated the methods as follows:

"Plant available P, the proportion of water-soluble P in soil that is available for uptake by plants, was extracted from 4 g of soil and 30 ml extraction fluid (1:7.5 ratio) using an acid–fluoride solution (P Bray-1), measured colorimetrically using a Systea EasyChem200 analyser, and expressed as mg/kg. The detection limit was 0.5 mg/kg, and plant available P measurements <0.5 mg/kg were replaced with half the detection limit (0.25 mg/kg)…………… Detection limits for soil ions were 0.01 mg/L, and soil ion concentrations measured as <0.01 mg/L were replaced with half the detection limit (0.005 mg/L)."

- The presentation of elemental data for soil and plant composition is non-standardized throughout. Some data (i.e., iron) are presented as mg/kg (is this soil dry weight?), while others are presented as % (Ca% of what?) in the same figure (figure 5 for example). Other data are presented as mg/L (figure 2). Part of this confusion is from the missing details in the methods that clearly explain how these data were generated. In several of the figures there is a statement about back-transformed data, which is also confusing.

AUTHORS' RESPONSE: We appreciate the attention to detail here from the reviewer. We have updated the manuscript so that soil ions, soil anions, and foliar micronutrients are all in mg/kg. Soil and foliar nitrogen are given as percentages, as this is the standard unit of measurement for the instrument used (IRMS) (methodological details appended below). We appreciate you pointing this out and agree that including these methodological details aids greatly in interpretation.

With regards to the comment on back-transformed data, we originally were log-transforming the nutrient data prior to analysis. For aid in visual interpretation of the results, we had displayed the data in its original units. Now that we have updated the

model structure and are no longer log-transforming before analysis, we have removed mentions of back-transformation from the manuscript.

- I can appreciate that finding carcasses that have decomposed for the same amount of time is challenging, but 1 month to 26 months is a huge range of time (at least from what we know from not megacarcasses). The biogeochemical processes occurring at a carcass decaying after 1 month postmortem is very different than a carcass that has been decaying for 26 months (from smaller carcasses). It would be useful to see some of the data, particularly ammonium, plotted as a function of postmortem interval (months) even if that is not a variable that could be included in statistical analyses because of the small sample size. It would also be helpful to see if the postmortem interval for the 10 carcasses is evenly distributed between the two soil types or if one has more fresh carcasses and the other has older carcasses, that could help with interpretation of the results.

AUTHORS' RESPONSE: Thanks for the suggestion! We made a figure showing soil ions (ammonium, nitrate, and phosphate) and respiration potential plotted against carcass age. In these four cases, it is clear that these soil metrics are higher at fresher carcasses. In fact, the trends are so compelling that we have added this figure to the main text (Figure 5). This figure suggests the pattern of elevated soil nutrients that we found may be even stronger when considering younger carcasses given how quickly the nutrients decline with age.

We ran a t-test to test for a difference in mean carcass age across soil types and found no significant difference between the two groups ($P = 0.294$). We will add this finding to the results section.

- I think it may be useful to add some photos to supplemental information (or even the main text) showing what the carcasses/sites looked like (maybe representative images from a fresher carcass and one that is older).

AUTHORS' RESPONSE: Thanks for the suggestion! We have added a supplemental figure (now Figure S1) that shows two carcass sites – one is fresh and on basaltic soil, and the other is older and on granitic soil. The new figure is appended to the end of this document.

Specific comments:

- Lines 99-100: There should be more details provided on the soil type and what makes the granitic soils "nutrient poor" compared to soils developed from a basalt protolith. Because soil type becomes an important part of this study, the details of the soil types need to be expanded in the introduction.

AUTHORS' RESPONSE: Thanks for the suggestion. Kruger National Park has two primary soil types – a clay-rich soil derived from basalt ("basaltic") and a sandy soil

derived from granite ("granitic"). Basalitic soils have clay particles with relatively large surface area, thereby enabling them to retain larger quantities of water than granitic soils, which drain water more quickly and therefore are lower in water-soluble nutrients (Buitenweref, Kulmatiski, & Higgins, 2014). We agree that these distinctions are important for understanding the impacts of carcass-derived nutrients on different soil types and have updated the methods section as follows: "The two dominant soil types in KNP are granitic soils (inceptisols) and basaltic soils (versitols or andisols) (Khomo et al. 2017). The clay-rich basaltic soils have relatively large surface area, enabling them to retain larger quantities of water than granitic soils, which drain water more quickly and therefore are lower in water-soluble nutrients (Buitenweref, Kulmatiski, & Higgins, 2014)."

Khomo, L., Trumbore, S., Bern, C. R., & Chadwick, O. A. Timescales of carbon turnover in soils with mixed crystalline mineralogies. SOIL, 3, 17-30, https://doi.org/10.5194/soil-3-17-2017, 2017.

Buitenwerf, R., Kulmatiski, A. & Higgins, S. I. Soil water retention curves for the major soil types of the Kruger National Park. Koedoe, 56, a1228, http://dx.doi.org/10.4102/koedoe.v56i1.1228, 2014.

- Line 132: Include a citation or discuss why soil samples were collected to a depth of 10 cm rather than the upper 5 cm. For decomposition studies, typically the upper 5 cm is examined, not the upper 10 cm.

AUTHORS' RESPONSE: We used a 10cm core to ensure that we captured the soil surface horizon. It is a commonly used depth and is more conservative than shallower sampling. Prior work on the soil impacts of carcasses uses this depth (Bump, Peterson, & Vucetich, 2009; Monk et al. 2024). Moreover, previous work in the same system has shown that soil auger sampling depths of 7.5-10cm are sufficient for detecting differences in N, C, and soil micronutrients (Gray & Bond 2015, Holdo & Mack 2014). We will update the text in the methods to include these references.

Bump, J. K., Peterson, R. O., & Vucetich, J. A. Wolves modulate soil nutrient heterogeneity and foliar nitrogen by configuring the distribution of ungulate carcasses. Ecology, 90, 3159–3167, 2009.

Monk, J. D., Donadio, E., Smith, J. A., Perrig, P. L., Middleton, A. D., & Schmitz, O. J. Predation and Biophysical Context Control Long-Term Carcass Nutrient Inputs in an Andean Ecosystem. Ecosystems, 27, 346–359, 2024.

Gray, E. F. & Bond, W. 2015. Soil nutrients in an African forest/savanna mosaic: Drivers or driven? South African Journal of Botany, 101, 66-72. https://doi.org/10.1016/j.sajb.2015.06.003

Holdo, R. M. & Mack, M. C. 2014. Functional attributes of savanna soils: contrasting effects of tree canopies and herbivores on bulk density, nutrients and moisture dynamics. Journal of Ecology, 102, 1171-1182. https://doi.org/10.1111/1365-2745.12290

- Line 145: More details are needed beyond "measurements of soil ion concentrations". What instrumentation was used? What specific extraction protocol was followed? I'm assuming deionized water was used (1:2 soil to deionized water?), but those details are not provided. How long were samples mixed (shaking platform?), what speed, etc.

AUTHORS' RESPONSE: We have updated the relevant portion of the methods as follows: "We sent 250 g of each soil sample to Eco-Analytica laboratory at the North-West University in Potchefstroom, South Africa for measurements of soil macro-element concentrations of ammonium $[NH_4]^+$, nitrate $[NO_3]^-$, phosphate $[PO_4]^{3-}$, and plant-available P. Samples were air-dried and sieved through < 2mm mesh prior to chemical analysis. Plant available P, the proportion of water-soluble P in soil that is available for uptake by plants, was extracted from 4 g of soil and 30 ml extraction fluid (1:7.5 ratio) using an acid–fluoride solution (P Bray-1), measured colorimetrically using a Systea EasyChem200 analyser, and expressed as mg/kg. The detection limit was 0.5 mg/kg, and plant available P measurements <0.5 mg/kg were replaced with half the detection limit (0.25 mg/kg). Water-soluble nitrate and phosphate anions were extracted from volume on volume 100 ml soil and 200 ml deionized water, analyzed by ion chromatography on a Metrohm 930 Compact Flex System, and expressed as mg/L. Ammonium (also 1:2 water extract) was analyzed colorimetrically using a Systea EasyChem200 analyzer and expressed as mg/L. Detection limits for soil ions were 0.01 mg/L, and soil ion concentrations measured as <0.01 mg/L were replaced with half the detection limit (0.005 mg/L). To convert the nitrate, ammonium, and phosphate units from mg/L to mg/kg, we multiplied by 2, based on the 1:2 soil to water extraction ratio.

- Line 148: "mass spectrometry"—elaborate on what this means with respect to instrumentation used to analyze cations. Here and throughout the methods, please also include what standards were used for the different analysis types.

AUTHORS' RESPONSE: We have updated the relevant section of the methods as follows: "To determine whether soil micronutrients were distinct and elevated at the center of carcass sites relative to soil further from the center, concentrations of sodium (Na), magnesium (Mg), iron (Fe), calcium (Ca), potassium (K), and phosphorus (P) cations were measured using microwave-assisted digestion. Air-dried and sieved (>2 mm) soil samples, weighed to 0.2 g, were microwaved in 9 ml 65% nitric acid ($HNO_3$) and 3 ml 32% hydrochloric acid (HCl) according to EPA 3051b in a Milestone, Ethos microwave digester with UP, Maxi 44 rotor. A period of 20 minutes allowed the system to reach 1800 MW at a temperature of 200 °C which was maintained for 15 minutes. After cooling, the samples were brought up to a final volume of 50 ml and analyzed on an Agilent 7500 CE ICP-MS fitted with CRC (Collision Reaction Cell) technology for interference removal. The instrument is optimized using a solution containing Li, Y, Ce, and Tl (1 ppb) for standard low-oxide/low interference levels (≤ 1.5%) while maintaining high sensitivity across the

mass range. The instrument was calibrated using ULTRASPEC® certified custom mixed multi-element stock standard solutions containing all the elements of interest (De Bruyn Spectroscopic Solutions, South Africa). Calibrations spanned the range of 0 – 30 ppm for the mineral elements Ca, Mg, Na, and K and 0 – 0.3 ppm for the rest of the trace elements. Elemental concentrations were expressed as mg/kg."

- Lines 146-150: Clarify if these analyses were conducted on the water extracts.

AUTHORS' RESPONSE: We have clarified as follows: "Water-soluble nitrate and phosphate anions were extracted from volume on volume 100 ml soil and 200 ml deionized water, analyzed by ion chromatography on a Metrohm 930 Compact Flex System, and expressed as mg/L. Ammonium (also 1:2 water extract) was analyzed colorimetrically using a Systea EasyChem200 analyzer and expressed as mg/L. Detection limits for soil ions were 0.01 mg/L, and soil ion concentrations measured as <0.01 mg/L were replaced with half the detection limit (0.005 mg/L). To convert the nitrate, ammonium, and phosphate units from mg/L to mg/kg, we multiplied by 2, based on the 1:2 soil to water extraction ratio.."

- Line 152: Were stable isotope analyses conduct on oven-dried soil? 10 g is an exceptionally large amount of soil—how much was actually analyzed with EA-IRMS? Were samples powdered prior to combustion?

AUTHORS' RESPONSE: We have clarified as follows: "Samples were oven-dried at 60°C for 48 hours and milled to a fine powder using a Retsch MM400 mill (Germany). The powdered samples were weighed off (2 – 60 mg) prior to combustion at 950°C."

- Line 154 (and throughout with respect to stable nitrogen isotope results): The authors refer to "$^{15}N$" measurements, but surely this should be presented as the ratio of 15/14N and in delta notation? In the methods here there also needs to be more description of the standard, the materials used for linearity, and the analytical precision of the instrument.

AUTHORS' RESPONSE: We have changed the notation throughout the manuscript to $\delta^{15}N$. We have updated the methods to include information on standards and precision as follows: "A high organic carbon (HOC) soil standard (0.52 ± 0.02 %N), along with two international reference standards (USGS40 ($\delta^{15}N$ -4.52% AIR) and USGS41 ($\delta^{15}N$ +47.57% AIR)) were used for calibration. The N elemental content was expressed relative to atmospheric N as $N_2$ $\delta^{15}N_{AIR}$ (‰). The quantification limit for $\delta^{15}N$ on the IRMS is 1 nA (nanoAmp), and the quantification limit for %N is 0.06%. The precision for %N was 0.02% and for $\delta^{15}N$ is ±0.11%, determined using the HOC standard, which was run multiple times throughout the analysis."

- Line 175: More details on the ICP-MS are needed, including standards, detection limits, etc. Additionally, were these samples digested in nitric acid? Water? How long were they microwaved?

AUTHORS' RESPONSE: We have updated the relevant section of the methods as follows: "To determine whether soil micronutrients were distinct and elevated at the center of carcass sites relative to soil further from the center, concentrations of sodium (Na), magnesium (Mg), iron (Fe), calcium (Ca), potassium (K), and phosphorus (P) cations were measured using microwave-assisted digestion. Air-dried and sieved (>2 mm) soil samples, weighed to 0.2 g, were microwaved in 9 ml 65% nitric acid ($HNO_3$) and 3 ml 32% hydrochloric acid (HCl) according to EPA 3051b in a Milestone, Ethos microwave digester with UP, Maxi 44 rotor. A period of 20 minutes allowed the system to reach 1800 MW at a temperature of 200 °C which was maintained for 15 minutes. After cooling, the samples were brought up to a final volume of 50 ml and analyzed on an Agilent 7500 CE ICP-MS fitted with CRC (Collision Reaction Cell) technology for interference removal. The instrument is optimized using a solution containing Li, Y, Ce, and Tl (1 ppb) for standard low-oxide/low interference levels ($\leq 1.5\%$) while maintaining high sensitivity across the mass range. The instrument was calibrated using ULTRASPEC® certified custom mixed multi-element stock standard solutions containing all the elements of interest (De Bruyn Spectroscopic Solutions, South Africa). Calibrations spanned the range of $0 – 30$ ppm for the mineral elements Ca, Mg, Na, and K and $0 – 0.3$ ppm for the rest of the trace elements. Elemental concentrations were expressed as mg/kg."

- Line 182: Adding some random number to each variable is not a standard way to handle data that are zero in your dataset (or if it is, there is no citation here and I am not familiar with that approach). Typically for geochemical data (like what was generated with ICP-MS), you can replace zero values with ½ the detection limit to remove non-zero data. There are other more technical ways to deal with zero values from a statistical standpoint, but the ½ the detection limit is the easiest and has the longest history of use. Please justify the use of your approach or re-run the analyses following a standard method for handling non-zero data in a geochemical dataset.

AUTHORS' RESPONSE: We have re-run the analyses using 0.005 mg/L as the replacement value for any zeros in the soil ion concentration data, as that is half of the detection limit (0.10 mg/L). The tables with these updated results are appended at the end of this document, but this update did not result in any changes to statistical significance or model performance in the results. We have updated the methods as follows: "Plant available P, the proportion of water-soluble P in soil that is available for uptake by plants, was extracted from 4 g of soil and 30 ml extraction fluid (1:7.5 ratio) using an acid–fluoride solution (P Bray-1), measured colorimetrically using a Systea EasyChem200 analyser, and expressed as mg/kg. The detection limit was 0.5 mg/kg, and plant available P measurements <0.5 mg/kg were replaced with half the detection limit (0.25 mg/kg)…………… Detection limits for soil ions were 0.01 mg/L, and soil ion concentrations measured as <0.01 mg/L were replaced with half the detection limit (0.005 mg/L)."

- Line 255: The part of the sentence that reads "…we found evidence that N from carcasses had moved from soils into plants" does not belong in the results section. This is interpretation and should be moved to the discussion.

AUTHORS' RESPONSE: We have edited this sentence to read: "Consistent with our third hypothesis, we found elevated foliar nutrient concentrations at elephant carcass sites." We include interpretation in the first paragraph of the discussion: "Together, these results indicate that carcass-derived nutrients move into soil and subsequently into plants over relatively short time scales, cycling essential nutrients such as N from carrion into the soil and back into aboveground nutrient pools."

- Lines 256-258: Similar comment as above where the content of this sentence is interpretation and should be moved to the discussion.

AUTHORS' RESPONSE: We have deleted the following clause from that sentence: "….indicating that the high N content in leaves closer to the center of a megacarcass site likely had an animal origin."

- Lines 295-297: I'm not quite sure I understand the logic presented here. First, soil microbial biomass was not measured. The respiration potential (through production of $CO_2$) was measured, but heterotrophic activity (which is how respiration can be interpreted) consumes oxygen. I think the phrasing here needs to be re-worked to not imply that the soil respiration (and the communities producing $CO_2$) are not necessarily the same that are driving nitrification.

AUTHORS' RESPONSE: We have edited this sentence to focus on heterotrophic activity rather than nitrification. It now reads: "Soil nitrate (Figure 2B) and soil respiration potential (Figure 3) were also elevated near the center of carcass sites, indicating higher rates of heterotrophic activity (Prosser, 2011)."

- Lines 304-305: There are prior studies that demonstrate the impact of increased organic C during decomposition on soil microbial processes that should be cited here (see studies by DeBruyn and colleagues)

AUTHORS' RESPONSE: Thanks for the suggestion. We have added two relevant citations to these lines from DeBruyn and colleagues.
1. Keenan, S. W., Schaeffer, S. M., Jin, V. L. & DeBruyn, J. M. Mortality hotspots: Nitrogen cycling in forest soils during vertebrate decomposition. Soil Biol. Biochem., 121, 165-176, https://doi.org/10.1016/j.soilbio.2018.03.005, 2018.
2. Keenan, S. W., Schaeffer, S. M., and DeBruyn, J. M.: Spatial changes in soil stable isotopic composition in response to carrion decomposition, Biogeosciences, 16, 3929–3939, https://doi.org/10.5194/bg-16-3929-2019, 2019.

- Lines 310-311: I'm not sure that this is phrased correctly. Phosphorus (predominantly as phosphate) is considered immobile in soil partly because of low solubility because it is often sorbed with Ca, Fe, Al or organics, and the release of P is tightly controlled by soil (or fluid) pH. N does not face the same sorts of sorption immobilization constraints. I think if you rephrased it to clarify that P and N are held within different reservoirs within soils that make them behave differently (and add some citations), that would help.

AUTHORS' RESPONSE: Thanks for the feedback. We have expanded that section to better explain the differences in how N and P are held in soils and agree that this adds clarity. The lines now read: "This lag could occur because phosphate easily forms chemical bonds with other soil ions (e.g., iron and aluminum in acidic soils and calcium in basic soils). Nitrate does not form these bonds and therefore has greater water solubility and soil mobility (Wiersum, 1962; Arai & Sparks, 2007)."

We have also added a second citation:

Aria, Y. & Sparks, D. L. Phosphate reaction dynamics in soils and soil components: a multiscale approach. Adv. Agron., 94, 135-179, https://doi.org/10.1016/S0065-2113(06)94003-6, 2007.

- Lines 324-328: As mentioned above, because the composition of the two soil types were not included, this part of the discussion is not supported by the results. It's unclear if the authors here are trying to say that the basaltic soils contain more nutrients after being impacted by decomposition or if the native state of the soils (background conditions) are more nutrient rich. I think if the introduction described the background chemistry of the two soil types this would be better supported. Additionally, basalt and granite contain different types of minerals and therefore additional sources of elements like P. I don't know what the specific mineralogy is of these two rock types in KNP, but it might be worth exploring. In particular, the presence of apatite (Ca-P bearing mineral) in the granite might also be contributing to elevated P measured in the granite soils.

AUTHORS' RESPONSE: Thanks for bringing this up. We will update the text to clarify that our results confirm a well-established pattern in the literature – that the background state of basaltic soils is more nutrient-rich than granitic soils. What we find interesting here is the significant interaction between soil type and distance with regards to ammonium and phosphate concentrations (Table S1; Figure 2). Ammonium and phosphate levels are elevated at the center of carcass sites in granitic soils relative to basaltic soils, but the difference between soil types disappears as distance from the carcass site increases. These results suggest that the impact of elephant carcasses on these soil ions is greater in the nutrient-poor granitic soils relative to basaltic soils. We will update the text in the discussion to better explain the importance of this soil type by distance interaction. We have also updated the introduction to include more information on the background differences between the two soil types, as described above.

- Figure 5 (and others): I'm a little bit confused by the figure caption. I think it would be better to present this as selected elements plotted as a function of distance. These are not the results of generalized linear mixed models, but the selection of how to present the data were informed by the models.

AUTHORS' RESPONSE: We have changed the text in the figure captions to clarify that we are plotting all the parameters that were included in the top model(s) for a given

response variable (Tables S1-2), which is standard practice. We hope this change in language is clearer.

**Revised Methods**

**2.1 Study system and sample collection**

We performed this research in the southern part of the Kruger National Park (KNP), South Africa (24.996 S, 31.592 E, ~275m elevation). The two dominant soil types in KNP are granitic soils (inceptisols) and basaltic soils (versitols or andisols) (Khomo et al. 2017). The clay-rich basaltic soils have relatively large surface area, enabling them to retain larger quantities of water than granitic soils, which drain water more quickly and therefore are lower in water-soluble nutrients (Buitenweref, Kulmatiski, & Higgins, 2014; Rughöft et al. 2016). The landscape at KNP is a mix of savanna grasslands and broadleaf woodlands, with an overstory dominated by trees from the genus *Combretum* (red bushwillow, *C. apiculatum*; russet bushwillow, *C. hereroense*; leadwood, *C. imberbe*) and trees formerly known as acacias (knobthorn, *Senegalensis nigrescens*; umbrella thorn, *Vachellia tortillis*). The park hosts a full suite of African savanna animals, including ~30,000 elephants (*Loxodonta africana*) (Coetsee & Ferreira, 2023), with a mortality rate of ~2% (~600 elephants per year). The targeted region of KNP has a high density of scavengers and predators, including white-backed vultures (*Gyps africanus*), spotted hyenas (*Crocuta crocuta*), and lions (*Panthera leo*) (Owen-Smith & Mills, 2007).

During the wet season in March 2023, we identified ten elephant carcass sites (1-26 months post-death), five on relatively nutrient-rich basaltic soil and five on nutrient-poor granitic soil. KNP section rangers provided precise GPS locations of where elephant carcasses had been found. Most elephants died of old age, illness, injury, or, in the case of one young bull, territorial fighting. These sites were recognizable *in situ* by a persistent bonefield, undigested gut contents,

and an absence of herbaceous vegetation. At each site, we hammered a rebar post into the center of the megacarcass disturbance and ran 15 m transects out from the post in each of the four cardinal directions. Based on pilot data, we treat the 10-15m distances as controls, sine the high degree of landscape heterogeneity in the system (e.g., differences in hill slope, vegetation, water drainage, proximity to termite mounds) made random transects difficult for interpretation. 
[revised manuscript text omitted]

[Figure]

Relationship between carcass age and key metrics (soil ion concentrations and respiration potential). (A) Soil ammonium, (B) soil nitrate, (C) soil phosphate, and (D) soil respiration potential are all higher at fresher carcass sites. Point respresent values at the center of the carcass site (distance = 0-0.5m).

**Figure 5.** Relationship between carcass age and key soil metrics (soil ion concentrations and respiration potential). (A) Soil ammonium, (B) nitrate, (C) phosphate, and (D) respiration potential are all higher at fresher carcass sites. Points represent individual measurements taken at the center of the carcass site (distance = 0-0.5m).

**Table S1.** Generalized linear mixed model results for soil variables. The same five models were run for each response variable, including a null model, and each included site as a random effect to account for repeat measurements. AICc is Akaike's Information Criterion, and ΔAICc is the difference between a given model and the best fit model for that response variable. Cum.Wt stand for cumulative weight; it gives the sum of Akaike's weights and indicates the likelihood that the models up to that point are the best in the set. Models with a ΔAICc value of 2 are considered roughly equivalent in fit and are italicized. Marginal $R^2$ is the proportion of variance explained by both fixed and random effects in a model, and conditional $R^2$ is the proportion of variance explained by fixed effects. Coefficients (± standard error) are shown for each predictor and model and are in log units. Rows are organized in blocks by response variable. Within blocks, models are listed in order of increasing ΔAICc.

| Model | Model Fit | | | | | Coefficients ± *SE* | | |
|---|---|---|---|---|---|---|---|---|
| | AICc | ΔAICc | Cum.Wt | Mar. $R^2$ | Con. $R^2$ | Soil | Distance | Soil × Distance |
| **Nitrogen (%)** | | | | | | | | |
| *Soil × Distance* | *-227.32* | *0.00* | *0.99* | *0.54* | *0.74* | *-0.26 ± 0.22* | *0.00 ± 0.01* | *-0.03 ± 0.01* |
| Soil + Distance | -216.13 | 11.20 | 1.00 | 0.46 | 0.67 | -0.48 ± 0.21 | -0.01 ± 0.00 | |
| Distance | -214.95 | 12.37 | 1.00 | 0.04 | 0.52 | | -0.01 ± 0.00 | |
| Soil | -212.36 | 14.97 | 1.00 | 0.40 | 0.62 | -0.47 ± 0.21 | | |
| Null | -211.23 | 16.09 | 1.00 | | | | | |
| **δ15N** | | | | | | | | |
| *Soil × Distance* | *180.87* | *0.00* | *0.77* | *0.55* | *0.70* | *0.39 ± 0.16* | *-0.02 ± 0.01* | *-0.02 ± 0.01* |
| Soil + Distance | 184.66 | 3.79 | 0.88 | 0.50 | 0.66 | 0.26 ± 0.15 | -0.03 ± 0.00 | |
| Distance | 184.67 | 3.79 | 1.00 | 0.34 | 0.60 | | -0.03 ± 0.00 | |
| Soil | 219.35 | 38.47 | 1.00 | 0.20 | 0.34 | 0.28 ± 0.14 | | |
| Null | 219.96 | 39.09 | 1.00 | | | | | |
| **Nitrate (mg/kg)** | | | | | | | | |
| *Distance* | *624.84* | *0.00* | *0.70* | *0.48* | *0.52* | | *-0.14 ± 0.02* | |

| | | | | | | | | |
|---|---|---|---|---|---|---|---|---|
| Soil + Distance | 627.06 | 2.23 | 0.93 | 0.48 | 0.52 | -0.14 ± 0.27 | -0.14 ± 0.02 | |
| Soil × Distance | 629.51 | 4.67 | 1.00 | 0.48 | 0.52 | -0.24 ± 0.39 | -0.14 ± 0.03 | 0.02 ± 0.04 |
| Null | 649.77 | 24.93 | 1.00 | | | | | |
| Soil | 651.82 | 26.99 | 1.00 | 0.01 | 0.04 | -0.18 ± 0.31 | | |
| **Ammonium (mg/kg)** | | | | | | | | |
| *Soil + Distance* | *219.52* | *0.00* | *0.65* | *0.58* | *0.77* | *2.49 ± 0.66* | *-0.18 ± 0.03* | |
| *Soil × Distance* | *220.94* | *1.43* | *0.97* | *0.60* | *0.77* | *2.91 ± 0.73* | *-0.15 ± 0.04* | *-0.07 ± 0.06* |
| Distance | 225.87 | 6.35 | 1.00 | 0.21 | 0.77 | | -0.18 ± 0.02 | |
| Soil | 244.57 | 25.05 | 1.00 | 0.34 | 0.70 | 2.51 ± 0.76 | | |
| Null | 249.38 | 29.86 | 1.00 | | | | | |
| **Phosphate (mg/kg)** | | | | | | | | |
| *Soil × Distance* | *167.99* | *0.00* | *0.98* | *0.52* | *0.79* | *2.20 ± 0.96* | *0.00 ± 0.05* | *-0.46 ± 0.08* |
| Soil + Distance | 178.68 | 10.69 | 1.00 | 0.18 | 0.18 | -0.38 ± 0.70 | -0.14 ± 0.06 | |
| Null | 180.65 | 12.66 | 1.00 | | | | | |
| Soil | Model did not converge | | | | | | | |
| Distance | Model did not converge | | | | | | | |
| **Plant Available Phosphorus (mg/kg)** | | | | | | | | |
| *Soil × Distance* | *447.18* | *0.00* | *0.94* | *0.34* | *0.63* | *0.16 ± 0.62* | *-0.04 ± 0.03* | *-0.13 ± 0.04* |
| Distance | 453.68 | 6.50 | 0.98 | 0.20 | 0.55 | | -0.10 ± 0.02 | |
| Soil + Distance | 454.80 | 7.62 | 1.00 | 0.26 | 0.55 | -0.66 ± 0.55 | -0.11 ± 0.02 | |
| Null | 467.35 | 20.17 | 1.00 | | | | | |
| Soil | 469.19 | 22.01 | 1.00 | 0.03 | 0.30 | -0.35 ± 0.47 | | |
| **Mineral Phosphorus (mg/kg)** | | | | | | | | |
| *Soil × Distance* | *537.77* | *0.00* | *1.00* | *0.86* | *0.95* | *-1.09 ± 0.32* | *0.00 ± 0.00* | *-0.04 ± 0.01* |
| Soil + Distance | 560.48 | 22.71 | 1.00 | 0.82 | 0.92 | -1.35 ± 0.31 | -0.02 ± 0.00 | |
| Distance | 566.38 | 28.61 | 1.00 | 0.04 | 0.76 | | -0.02 ± 0.00 | |
| Soil | 573.55 | 35.78 | 1.00 | 0.78 | 0.89 | -1.33 ± 0.31 | | |
| Null | 579.62 | 41.85 | 1.00 | | | | | |
| **Sodium (mg/kg)** | | | | | | | | |
| *Soil × Distance* | *438.56* | *0.00* | *0.73* | *0.29* | *0.59* | *0.22 ± 0.35* | *-0.03 ± 0.01* | *-0.04 ± 0.02* |
| Distance | 441.09 | 2.53 | 0.94 | 0.22 | 0.54 | | -0.05 ± 0.00 | |
| Soil + Distance | 443.53 | 4.97 | 1.00 | 0.22 | 0.54 | -0.06 ± 0.35 | -0.05 ± 0.01 | |
| Null | 464.02 | 25.45 | 1.00 | | | | | |

| | | | | | | | | |
|---|---|---|---|---|---|---|---|---|
| Soil | 466.38 | 27.82 | 1.00 | 0.00 | 0.34 | 0.00 ± 0.00 | | |
| **Potassium (mg/kg)** | | | | | | | | |
| *Soil × Distance* | *676.07* | *0.00* | *0.94* | *0.29* | *0.81* | *-0.23 ± 0.42* | *0.01 ± 0.00* | *-0.02 ± 0.01* |
| Null | 682.93 | 6.86 | 0.97 | | | | | |
| Soil | 684.55 | 8.48 | 0.99 | 0.25 | 0.78 | -0.37 ± 0.41 | | |
| Distance | 685.17 | 9.10 | 1.00 | 0.00 | 0.72 | | 0.00 ± 0.00 | |
| Soil + Distance | 686.89 | 10.82 | 1.00 | 0.26 | 0.78 | -0.37 ± 0.41 | 0.00 ± 0.00 | |
| **Calcium (mg/kg)** | | | | | | | | |
| *Soil* | *749.09* | *0.00* | *0.60* | *0.82* | *0.94* | *-1.45 ± 0.41* | | |
| *Soil + Distance* | *751.01* | *1.92* | *0.83* | *0.82* | *0.94* | *-1.45 ± 0.01* | *0.00 ± 0.00* | |
| Soil × Distance | 753.00 | 3.91 | 0.91 | 0.82 | 0.94 | -1.42 ± 0.41 | 0.00 ± 0.01 | -0.01 ± 0.01 |
| Null | 753.55 | 4.46 | 0.97 | | | | | |
| Distance | 755.37 | 6.27 | 1.00 | 0.00 | 0.81 | | 0.00 ± 0.00 | |
| **Iron (mg/kg)** | | | | | | | | |
| *Soil* | *914.44* | *0.00* | *0.67* | *0.88* | *0.96* | *-1.22 ± 0.28* | | |
| Soil + Distance | 916.83 | 2.39 | 0.87 | 0.88 | 0.96 | -1.22 ± 0.28 | 0.00 ± 0.00 | |
| Soil × Distance | 918.54 | 4.10 | 0.95 | 0.88 | 0.96 | -1.19 ± 0.28 | 0.00 ± 0.00 | 0.00 ± 0.01 |
| Null | 920.27 | 5.83 | 0.99 | | | | | |
| Distance | 922.55 | 8.11 | 1.00 | 0.00 | 0.82 | | 0.00 ± 0.00 | |
| **Magnesium (mg/kg)** | | | | | | | | |
| *Soil* | *700.88* | *0.00* | *0.63* | *0.87* | *0.96* | *-1.53 ± 0.37* | | |
| Soil + Distance | 703.33 | 2.45 | 0.81 | 0.87 | 0.96 | -1.53 ± 0.37 | 0.00 ± 0.00 | |
| Soil × Distance | 703.97 | 3.09 | 0.95 | 0.88 | 0.96 | -1.48 ± 0.37 | 0.00 ± 0.00 | -0.01 ± 0.01 |
| Null | 706.40 | 5.52 | 0.99 | | | | | |
| Distance | 708.75 | 7.87 | 1.00 | 0.00 | 0.84 | | 0.00 ± 0.00 | |
| **Water (mmol/mol)** | | | | | | | | |
| *Null* | *111.87* | *0.00* | *0.32* | | | | | |
| *Distance* | *112.09* | *0.22* | *0.61* | *0.03* | *0.38* | | *0.02 ± 0.01* | |
| *Soil* | *112.92* | *1.05* | *0.80* | *0.12* | *0.40* | *0.45 ± 0.38* | | |
| *Soil + Distance* | *113.27* | *1.40* | *0.96* | *0.14* | *0.42* | *0.45 ± 0.38* | *0.02 ± 0.01* | |
| Soil × Distance | 115.86 | 3.99 | 1.00 | 0.14 | 0.42 | 0.44 ± 0.42 | 0.02 ± 0.02 | 0.00 ± 0.03 |
| **pH** | | | | | | | | |
| *Soil × Distance* | *55.04* | *0.00* | *0.37* | *0.07* | *0.44* | *0.05 ± 0.07* | *0.00 ± 0.00* | *-0.01 ± 0.00* |
| *Null* | *55.26* | *0.22* | *0.71* | | | | | |

| Distance | 56.94 | 1.90 | 0.86 | 0.01 | 0.38 | | 0.00 ± 0.00 | |
| Soil | 57.63 | 2.59 | 0.96 | 0.00 | 0.37 | 0.00 ± 0.07 | | |
| Soil + Distance | 59.41 | 4.37 | 1.00 | 0.01 | 0.38 | 0.00 ± 0.00 | 0.00 ± 0.00 | |

**Table S2.** Generalized linear mixed model results for leaf variables. The same five models were run for each response variable, including a null model, and each included site as a random effect to account for repeat measurements. AICc is Akaike's Information Criterion, and ΔAICc is the difference between a given model and the best fit model for that response variable. Cum.Wt stand for cumulative weight; it gives the sum of Akaike's weights and indicates the likelihood that the models up to that point are the best in the set. Models with a ΔAICc value of 2 are considered roughly equivalent in fit and are italicized. Marginal $R^2$ is the proportion of variance explained by both fixed and random effects in a model, and conditional $R^2$ is the proportion of variance explained by fixed effects. Coefficients (± standard error) are shown for each predictor and model and are in log units. Rows are organized in blocks by response variable. Within blocks, models are listed in order of increasing ΔAICc.

| Model | Model Fit | | | | | Coefficients ± *SE* | | |
|---|---|---|---|---|---|---|---|---|
| | AICc | ΔAICc | Cum.Wt | Mar. $R^2$ | Con. $R^2$ | Soil | Distance | Soil × Distance |
| **Nitrogen (%)** | | | | | | | | |
| *Distance* | *56.12* | *0.00* | *0.64* | *0.40* | *0.60* | | *-0.03 ± 0.00* | |
| Soil + Distance | 57.79 | 1.67 | 0.92 | 0.43 | 0.61 | 0.13 ± 0.14 | -0.03 ± 0.00 | |
| Soil × Distance | 60.33 | 4.20 | 1.00 | 0.43 | 0.61 | 0.15 ± 0.15 | -0.03 ± 0.01 | 0.00 ± 0.01 |
| Null | 89.78 | 33.66 | 1.00 | | | | | |
| Soil | 91.66 | 35.53 | 1.00 | 0.03 | 0.21 | 0.10 ± 0.13 | | |
| **δ15N** | | | | | | | | |
| *Soil × Distance* | *229.95* | *0.00* | *0.95* | *0.51* | *0.77* | *-0.52 ± 0.43* | *-0.11 ± 0.01* | *0.06 ± 0.02* |
| Distance | 236.55 | 6.60 | 0.99 | 0.44 | 0.70 | | -0.08 ± 0.01 | |
| Soil + Distance | 238.97 | 9.02 | 1.00 | 0.45 | 0.70 | -0.12 ± 0.40 | -0.08 ± 0.01 | |
| Null | 282.45 | 52.50 | 1.00 | | | | | |
| Soil | 284.30 | 54.34 | 1.00 | 0.04 | 0.36 | -0.30 ± 0.41 | | |
| **Phosphorus (%)** | | | | | | | | |
| *Soil × Distance* | *-87.04* | *0.00* | *0.99* | *0.47* | *0.75* | *-0.24 ± 0.31* | *0.02 ± 0.01* | *-0.04 ± 0.01* |
| Soil | -76.10 | 10.94 | 1.00 | 0.38 | 0.68 | -0.55 ± 0.31 | | |
| Null | -75.98 | 11.06 | 1.00 | | | | | |

| | | | | | | | | |
|---|---|---|---|---|---|---|---|---|
| Soil + Distance | -73.69 | 13.34 | 1.00 | 0.38 | 0.68 | -0.55 ± 0.31 | 0.00 ± 0.01 | |
| Distance | -73.68 | 13.36 | 1.00 | 0.00 | 0.56 | | 0.00 ± 0.01 | |
| **N:P Ratio** | | | | | | | | |
| *Soil × Distance* | 209.64 | 0.00 | 0.86 | 0.41 | 0.71 | 0.34 ± 0.38 | -0.05 ± 0.01 | 0.04 ± 0.01 |
| Distance | 214.60 | 4.96 | 0.94 | 0.09 | 0.59 | | -0.03 ± 0.01 | |
| Soil + Distance | 214.85 | 5.21 | 1.00 | 0.36 | 0.67 | 0.62 ± 0.01 | -0.03 ± 0.00 | |
| Null | 225.74 | 16.10 | 1.00 | | | | | |
| Soil | 226.21 | 16.57 | 1.00 | 0.23 | 0.57 | 0.55 ± 0.37 | | |
| **Sodium (mg/kg)** | | | | | | | | |
| *Soil + Distance* | *839.97* | *0.00* | *0.60* | *0.62* | *0.78* | *-0.99 ± 0.32* | *-0.03 ± 0.01* | |
| *Soil × Distance* | *841.56* | *1.59* | *0.88* | *0.62* | *0.79* | *-0.88 ± 0.34* | *-0.03 ± 0.01* | *-0.02 ± 0.01* |
| Distance | 843.18 | 3.21 | 1.00 | 0.09 | 0.64 | | -0.03 ± 0.01 | |
| Soil | 852.98 | 13.02 | 1.00 | 0.53 | 0.71 | -1.00 ± 0.32 | | |
| Null | 856.49 | 16.52 | 1.00 | | | | | |
| **Magnesium (mg/kg)** | | | | | | | | |
| *Soil × Distance* | *722.20* | *0.00* | *0.99* | *0.45* | *0.80* | *-0.20 ± 0.28* | *0.00 ± 0.00* | *-0.02 ± 0.01* |
| Distance | 731.74 | 9.54 | 0.99 | 0.07 | 0.66 | | -0.01 ± 0.00 | |
| Soil + Distance | 732.78 | 10.58 | 1.00 | 0.39 | 0.76 | -0.36 ± 0.28 | -0.01 ± 0.00 | |
| Null | 743.56 | 21.36 | 1.00 | | | | | |
| Soil | 744.46 | 22.26 | 1.00 | 0.31 | 0.69 | -0.37 ± 0.28 | | |
| **Potassium (mg/kg)** | | | | | | | | |
| *Distance* | *936.99* | *0.00* | *0.73* | *0.20* | *0.57* | | *-0.03 ± 0.00* | |
| Soil + Distance | 939.50 | 2.51 | 0.94 | 0.20 | 0.57 | 0.02 ± 0.25 | -0.03 ± 0.00 | |
| Soil × Distance | 941.96 | 4.97 | 1.00 | 0.20 | 0.57 | 0.05 ± 0.26 | -0.02 ± 0.01 | 0.00 ± 0.01 |
| Null | 956.55 | 19.57 | 1.00 | | | | | |
| Soil | 958.95 | 21.96 | 1.00 | 0.00 | 0.38 | 0.00 ± 0.24 | | |
| **Calcium (mg/kg)** | | | | | | | | |
| *Null* | *799.64* | *0.00* | *0.42* | | | | | |
| *Distance* | *800.68* | *1.04* | *0.67* | *0.01* | *0.50* | | *0.00 ± 0.00* | |
| *Soil* | *801.22* | *1.58* | *0.86* | *0.14* | *0.53* | *-0.20 ± 0.21* | | |
| Soil + Distance | 802.36 | 2.72 | 0.96 | 0.14 | 0.54 | -0.20 ± 0.21 | 0.00 ± 0.00 | |
| Soil × Distance | 804.45 | 4.81 | 1.00 | 0.15 | 0.54 | -0.16 ± 0.22 | 0.01 ± 0.01 | -0.01 ± 0.01 |
| **Iron (mg/kg)** | | | | | | | | |
| *Distance* | *591.87* | *0.00* | *0.69* | *0.21* | *0.57* | | *-0.08 ± 0.01* | |

| | | | | | | | | |
|---|---|---|---|---|---|---|---|---|
| Soil + Distance | 594.14 | 2.27 | 0.92 | 0.23 | 0.58 | -0.26 ± 0.50 | -0.08 ± 0.01 | |
| Soil × Distance | 596.15 | 4.27 | 1.00 | 0.23 | 0.59 | -0.09 ± 0.39 | -0.07 ± 0.00 | -0.02 ± 0.02 |
| Null | 616.95 | 25.08 | 1.00 | | | | | |
| Soil | 619.06 | 27.19 | 1.00 | 0.02 | 0.48 | -0.31 ± 0.00 | | |

**Table S3.** Generalized linear mixed model results testing for correlations between leaf and soil micronutrients. The same model was run for each of five micronutrients (Na, K, Ca, Mg, and Fe) with leaf micronutrient concentration as the response variable, soil micronutrient + distance as the main effects, and site as a random effect. Marginal $R^2$ is the proportion of variance explained by both fixed and random effects in a model, and conditional $R^2$ is the proportion of variance explained by fixed effects. Coefficients (± standard error) are shown for each predictor and model.

| Leaf Micronutrient | Mar. $R^2$ | Con. $R^2$ | Soil Micronutrient Coefficient ± *SE* | Distance Coefficient ± *SE* |
|---|---|---|---|---|
| Sodium | 0.08 | 0.82 | 11.56 ± 11.67 | -146.47 ± 43.04 |
| Potassium | 0.29 | 0.73 | 0.00 ± 0.00 | -0.06 ± 0.01 |
| Calcium | 0.12 | 0.58 | 0.00 ± 0.00 | 0.00 ± 0.00 |
| Magnesium | 0.17 | 0.79 | 0.00 ± 0.00 | 0.00 ± 0.00 |
| Iron | 0.11 | 0.32 | 0.00 ± 0.01 | -52.85 ± 20.57 |

[Figure]

**Figure S1.** Representative photos of two elephant carcass sites of different ages and soil types. (A) The first site is 67 days post-death and is on granitic soil. (B) The second site is 811 days post-death and is on basaltic soil. In both images, there is a visible impact zone with reduced vegetation coverage. At the first site, elephant bones have all been dispersed, though some are still present at the second site. Photos taken by Deron Burkepile at time of sample collection in March 2023.

[Figure]

**Figure S2.** (A) Soil micronutrient composition did not differ significantly with distance from the carcass but (B) was distinct in different soil types.

[Figure]

**Figure S3.** Effects of elephant carcasses on soil micronutrients. (A) Soil sodium decreased significantly with distance from the carcass. (B) Potassium decreased with distance but only in granitic soils. (C) Iron, (D) magnesium, and (E) calcium were greater in basaltic soils. Distance appeared in the top model for calcium, but the effect size was minimal. Points represent individual measurements taken at 0, 2.5, 5, 10, and 15m and are offset to be visible when they would otherwise overlap. Lines show predictions calculated from the top model. Shading indicates the 95% confidence interval.

[Figure]

**Figure S4**. Neither (A) soil water nor (B) soil pH differed with distance or soil type. Points represent individual measurements taken at 0, 2.5, 5, 10, and 15m and are offset to be visible when they would otherwise overlap.

[Figure]

**Figure S5.** (A) Foliar micronutrient composition did not differ significantly with distance from the carcass but (B) was distinct in different soil types.

[Figure]

**Figure S6**. Effects of elephant carcasses on grass foliar micronutrients. (A) Foliar Na and (B) Mg were greatest in basaltic soil and decreased significantly with distance. (C) Foliar K and (D) Fe decreased with distance but did not differ with soil type. (E) Foliar Ca did not differ with distance or soil type. Points represent individual measurements taken at 0, 2.5, 5, 10, and 15m and are offset to be visible when they would otherwise overlap. Lines show predictions calculated from the top model. Shading indicates the 95% confidence interval.

---

## Author Comment (AC4)

**Response to Reviewer 3**

This manuscript presents data on the influence of elephant carcasses on nutrient availability in South African savanna soils. It would be a surprise if a decaying elephant did not increase nutrient concentrations in the proximity of the carcass, but there are some interesting differences among nutrients in terms of the distance over which the effects extend. There are parallels with other nutrient hotspots in tropical ecosystems, including glades in African savannas (e.g. Augustine 2003) and leafcutter ant nests in tropical forests (e.g. Hudson et al. 2009) - it would be worth introducing these into the discussion for comparison. I have several questions about methodology and results that should be addressed before this manuscript could be acceptable for publication.

AUTHORS' RESPONSE: Thank you for all of your feedback. We have made substantial changes in response to your suggestions, including rewriting the methods section to include more details on the soil lab analyses. We have updated the introduction to better explain the importance of cations and cation exchange in savanna soils. We have also updated the discussion to more thoroughly address the ecological significance of our results, including comparison with other nutrient hotspots like the cattle bomas (Augustine 2003) and leafcutter ant nests (Hudson et al. 2009) suggested by the reviewer.

The most significant critique throughout this review was concern that freezing the soil samples prior to ion analysis may have resulted in elevated ammonium and nitrate levels. We agree that this is an important concern, but we are reassured by the findings in the literature showing that nitrogen measurements (especially nitrate) are relatively robust to storage method. Indeed, in Turner and Romero (2009) (a paper the reviewer cites), freezing did not impact nitrate concentrations relative to fresh soils except in cases of high soil acidity, which were not present in our study. Other studies show that nitrate is unaffected by freezing treatment for at least seven weeks post collection, well within the time frame of analysis for our samples (Esala, 1995; Sollen-Norrlin & Rintoul-Hynes, 2024). Soil ammonium can increase when frozen, although the increase is often relatively small (<1 mg/kg per week frozen; Esala 1995, Fig. 1). It is true that freezing can have a large impact on ammonium in peaty soils, but freezing has only minimal effects on ammonium measurements in clay soils such as ours (Esala, 1995; Sollen-Norrlin & Rintoul-Hynes, 2024). Further, we have compared the soil % nitrogen, nitrate, and ammonium values from the 15m distance in our study (essentially representing the background levels of nutrients in the soils in Kruger) to those found in other soil analysis research in Kruger, and our values are consistent with that prior research (Aranibar et al. 2003; Rughöft et al. 2016; see table below). Our foliar N:P values were consistent with the Kruger literature as well. We found much higher soil nitrogen values at 0-5m distances from the carcass site than those found in these papers, but that is what we hypothesized would happen with nutrient inputs from elephant carcasses. Moreover, even if the absolute values of nitrogen in our study are elevated due to freezing, which there is little evidence to support, there is no reason that the effects of freezing would differ with distance from an elephant carcass, so we are confident that the overall trends in this manuscript are robust.

All major updates are appended to the end of this document, including methods section 2.3, supplemental tables, and main and supplemental figures.

| Metric | Source | Mean | Range | Method |
|---|---|---|---|---|
| Soil N | Reed et al. | 11.4% | 5 – 16% | Stable isotope analysis |
| | Aranibar et al. 2003 | | ~5 – 23% | Stable isotope analysis |
| Soil Nitrate | Reed et al. | 57.1 mg/kg | 11.1 – 95.7 mg/kg | 1:2 water extract analysis |
| | Rughöft et al. 2016 | 28.9 mg/kg | 0.0 – 121.9 mg/kg | 2:5 water extract analysis |
| Soil Ammonium | Reed et al. | 1.38 mg/kg | 0.01 – 6.5 mg/kg | 1:2 water extract analysis |
| | Rughöft et al. 2016 | 11.3 mg/kg | 0.7 – 33.3 mg/kg | 2:5 water extract analysis |
| Soil Plant-Available P | Reed et al. | 2.20 mg/kg | 0.01 – 9.62 mg/kg | P Bray I |
| | Craine et al. 2008 | 38.52 mg/kg | 3.23 – 85.43 mg/kg | P Bray II |
| Leaf N:P Ratio | Reed et al. | 7.0 | 2.5 – 13.9 | |
| | Craine et al. 2008 | 5.8 | 3.2 – 9.2 | |

*For Reed et al., we used values from the 15m distance, and for Craine et al. 2008, we used values from the control plots, as in both cases these best represent the background levels of soil nutrients.

Aranibar, J. N., Macko, S. A., Anderson, I. C., Potgieter, A. L. F., Sowry, R. & Shugart, H. H. Nutrient cycling responses to fire frequency in the Kruger National Park (South Africa) as indicated by stable isotope analysis. Isotopes Environ. Health Stud., 39, 141-158, 2003.

Rughöft, S., Hermann, M., Lazar, C. S., Cesarz, S., Levick, S. R., Trumbore, S. E. & Küsel, K. Community composition and abundance of bacterial, archaeal and nitrifying populations in savanna soils on contrasting bedrock material in Kruger National Park, South Africa. Frontiers in Microbiology, 7, 2016.

Craine et al. Nutrient concentration ratios and co-limitation in South African grasslands. New Phytologist, 179, 829-836, 2008.

Esala, M. J. Changes in the extractable ammonium- and nitrate-nitrogen contents of soil samples during freezing and thawing. Communications in Soil Science and Plant Analysis, 26, 61-68, 1995.

Sollen-Norrlin, M. & Rintoul-Hynes, N. L. J. Soil sample storage conditions affect measurements of pH, potassium, and nitrogen. Soil Science Society of America Journal, 88, 930-941, 2024.

Turner, B. L. & Romero, T. E. Short-term changes in extractable inorganic nutrients during storage of tropical rainforest soils. Soil Science Society of America Journal, 73, 1972-1979, 2009.

Line 38 – these are not graves. A grave is an excavation for burial.

AUTHORS' RESPONSE: We have changed the word gravesite to carcass site.

Line 77 – what about the amounts of cations in an elephant?

AUTHORS' RESPONSE: This is an interesting question. We estimated the N, P, and C quantities in an elephant based on the body size to macronutrient scaling rules described in Sterner & Elser (2002). Unfortunately, we are not aware of a well-established scaling rule for cations that would allow us to estimate cation concentrations in elephant tissue.

Sterner, R. W. & Elser, J. J. Ecological Stoichiometry: The Biology of Elements from Molecules to the Biosphere. Princeton University Press. 2002.

Line 89 – cations are not micronutrients. Is there direct evidence for widespread (or any) cation limitation of growth in savanna ecosystems?

AUTHORS' RESPONSE: Cation exchange is essential for soil fertility, and there is evidence of cation limitation (particularly $K^+$ and $Ca^{2+}$) on African savannas (Lathwell & Grove, 1986; Agbenin & Yakubu, 2006). We now refer to this idea in the manuscript. Further, we have edited the manuscript to refer to these elements as cations rather than micronutrients for clarity.

Agbenin, J. O. & Yakubu, S. Potassium-calcium and potatssium-magnesium exchange equilibria in an acid savanna soil from northern Nigeria. Geoderma, 136, 542-554, https://doi.org/10.1016/j.geoderma.2006.04.008, 2006.

Lathwell, D. J. & Grove, T. L. Soil-Plant Relationships in the Tropics. Annual Review of Ecology and Systematics, 17, 1-16, https://www.jstor.org/stable/2096986, 1986.

Line 99 – please include classifications for the granitic and basaltic soils in one of the internationally recognized systems. In Soil Taxonomy these are presumably Inceptisols / Alfisols and Vertisols, respectively?

AUTHORS' RESPONSE: Thanks for the suggestion. The granitic soils are inceptisols, while the basaltic soils may be versitols or andisols (Khomo et al. 2017). We agree that these classifications are important for understanding the impacts of carcass-derived nutrients on different soil types and have updated the methods section as follows: "The two dominant soil types in KNP are granitic soils (inceptisols) and basaltic soils (versitols or andisols) (Khomo et al. 2017). The clay-rich basaltic soils have relatively large surface area, enabling them to retain larger quantities of water than granitic soils, which drain water more quickly and therefore are lower in water-soluble nutrients (Buitenweref, Kulmatiski, & Higgins, 2014)."

Khomo, L., Trumbore, S., Bern, C. R., & Chadwick, O. A. Timescales of carbon turnover in soils with mixed crystalline mineralogies. SOIL, 3, 17-30, https://doi.org/10.5194/soil-3-17-2017, 2017.

Buitenwerf, R., Kulmatiski, A. & Higgins, S. I. Soil water retention curves for the major soil types of the Kruger National Park. Koedoe, 56, a1228, http://dx.doi.org/10.4102/koedoe.v56i1.1228, 2014.

Line 138 – freezing soil has implications for subsequent measurements of extractable nutrients (e.g. Turner and Romero 2009). This should be mentioned here. Given the apparently very high values for some measurements (see below) I suspect that pretreatment had a major impact on results.

AUTHORS' RESPONSE: As detailed above, we acknowledge that freezing may have an impact on soil nitrogen measurements, though the literature is mixed on this topic. Generally, it seems that freezing more often impacts concentrations of ammonium but not nitrate (Esala, 1995; Turner & Romero, 2009; Sollen-Norrlin & Rintoul-Hynes, 2024). Indeed, in the Turner and Romero (2009) paper that the reviewer cites, freezing did not impact nitrate concentrations relative to fresh soils except in cases of high soil acidity, which were not present in our study. They write, "Frozen storage of the Pipeline Road and Fort Sherman soils preserved NO3 at similar concentrations to fresh samples (Table 2), but caused a marked decline for the acidic Albrook soil, indicating a possible effect of soil pH on NO3 stability. (Note: NO3 determined after 3 mo of storage in the Albrook soil)" (pg. 1974). The impacts of freezing on ammonium are also relatively minimal (<1 mg/kg per week of freezing according to Esala, 1995). Thus, based on what is known from the literature any impacts of freezing on the concentrations of nutrients in our samples were likely quite minimal. Further, we have compared the soil % nitrogen, nitrate, and ammonium values from the 10 and 15m distances in our study (those most similar to the background soils in Kruger) to those found in other soil analysis research in Kruger, and our values are consistent with that prior research (Aranibar et al. 2003; Rughöft et al. 2016).

| **Metric** | **Source** | **Mean** | **Range** | **Method** |
|---|---|---|---|---|
| Soil %N | Reed et al. | 11.4% | 5 – 16% | Stable isotope analysis |
| | Aranibar et al. 2003 | | ~5 – 23% | Stable isotope analysis |
| Soil Nitrate | Reed et al. | 57.1 mg/kg | 11.1 – 95.7 mg/kg | 1:2 water extract analysis |
| | Rughöft et al. 2016 | 28.9 mg/kg | 0.0 – 121.9 mg/kg | 2:5 water extract analysis |
| Soil Ammonium | Reed et al. | 1.38 mg/kg | 0.01 – 6.5 mg/kg | 1:2 water extract analysis |
| | Rughöft et al. 2016 | 11.3 mg/kg | 0.7 – 33.3 mg/kg | 2:5 water extract analysis |
| Soil Plant-Available P | Reed et al. | 2.20 mg/kg | 0.01 – 9.62 mg/kg | P Bray I |
| | Craine et al. 2008 | 38.52 mg/kg | 3.23 – 85.43 mg/kg | P Bray II |
| Leaf N:P Ratio | Reed et al. | 7.0 | 2.5 – 13.9 | |
| | Craine et al. 2008 | 5.8 | 3.2 – 9.2 | |

*For Reed et al., we used values from the 15m distance, and for Craine et al. 2008, we used values from the control plots, as in both cases these best represent the background levels of soil nutrients.

If freezing the samples did alter the measured soil ammonium concentrations, there is no reason to believe that the effect size would differ with distance from the carcass. Thus, even if absolute values are elevated by freezing, for which there is little evidence in the literature, the overall trends of decreasing ammonium and nitrate with carcass distance should still stand. We have added these citations and explanation to the discussion.

Esala, M. J. Changes in the extractable ammonium- and nitrate-nitrogen contents of soil samples during freezing and thawing. Communications in Soil Science and Plant Analysis, 26, 61-68, https://doi.org/10.1080/00103629509369280, 1995.

Sollen-Norrlin, M. & Rintoul-Hynes, N. L. J. Soil sample storage conditions affect measurements of pH, potassium, and nitrogen. Soil Science Society of America Journal, 88, 930-941, https://doi.org/10.1002/saj2.20653, 2024.

Turner, B. L. & Romero, T. E. Short-term changes in extractable inorganic nutrients during storage of tropical rainforest soils. Soil Science Society of America Journal, 73, 1972-1979, doi:10.2136/sssaj2008.0407, 2009.

Aranibar, J. N., Macko, S. A., Anderson, I. C., Potgieter, A. L. F., Sowry, R. & Shugart, H. H. Nutrient cycling responses to fire frequency in the Kruger National Park (South Africa) as indicated by stable isotope analysis. Isotopes Environ. Health Stud., 39, 141-158, 2003.

Rughöft, S., Hermann, M., Lazar, C. S., Cesarz, S., Levick, S. R., Trumbore, S. E. & Küsel, K. Community composition and abundance of bacterial, archaeal and nitrifying populations in savanna soils on contrasting bedrock material in Kruger National Park, South Africa. Frontiers in Microbiology, 7, 2016.

Craine et al. Nutrient concentration ratios and co-limitation in South African grasslands. New Phytologist, 179, 829-836, 2008.

Line 145/146 – please explain the difference between phosphate and plant-available P. As written, it appears they were both measured in the water extracts. Most plant-available P tests are not conducted in water (e.g. Olsen, Mehlich, Bray, etc).

AUTHORS' RESPONSE: We have updated the methods section as follows to include more details on soil nutrient analyses, including distinguishing between the methods used for phosphate and plant-available P.

"We sent 250 g of each soil sample to Eco-Analytica laboratory at the North-West University in Potchefstroom, South Africa for measurements of soil macro-element concentrations of ammonium $[NH_4]^+$, nitrate $[NO_3]^-$, phosphate $[PO_4]^{3-}$, and plant-available P. Samples were air-dried and sieved through < 2mm mesh prior to chemical analysis. Plant available P, the

proportion of water-soluble P in soil that is available for uptake by plants, was extracted from 4 g of soil and 30 ml extraction fluid (1:7.5 ratio) using an acid–fluoride solution (P Bray-1), measured colorimetrically using a Systea EasyChem200 analyser, and expressed as mg/kg. The limit was 0.5 mg/kg, and plant available P measurements <0.5 mg/kg were replaced with half the detection limit (0.25 mg/kg). Water-soluble nitrate and phosphate anions were extracted from volume on volume 100 ml soil and 200 ml deionized water, analyzed by ion chromatography on a Metrohm 930 Compact Flex System, and expressed as mg/L. Ammonium (also 1:2 water extract) was analyzed colorimetrically using a Systea EasyChem200 analyzer and expressed as mg/L. Detection limits for soil ions were 0.01 mg/L, and soil ion concentrations measured as <0.01 mg/L were replaced with half the detection limit (0.005 mg/L). To convert the nitrate, ammonium, and phosphate units from mg/L to mg/kg, we multiplied by 2, based on the 1:2 soil to water extraction ratio."

Line 158 – was there any inorganic C in the samples? Savanna Vertisols developed in basalt can have considerable carbonate concentrations, albeit often in subsoil. Soil pH values would help indicate this possibility – how did carcasses affect soil pH?

AUTHORS' RESPONSE: We did not directly measure inorganic C, but we did measure soil pH and found no significant difference in soil pH with soil type or distance from the carcass center. We have added this result to the manuscript (Table S1, Figure S4B).

Line 161 – was moisture standardized prior to the incubations?

AUTHORS' RESPONSE: We did not standardize moisture prior to incubations, since the impact of carcasses on soil moisture was one of our questions (Figure S5 in original manuscript, Figure S4 below). However, we did account for soil moisture after the incubations following the methods process described in lines 163-167: "After soil respiration measurements, we determined sample dry weight by drying each sample at 60°C for 24-48 hours until stable mass was achieved. We subtracted dry weight from starting weight to obtain soil water content. Finally, we used the dry weights and the Ideal Gas Law to standardize all respiration measurements to $CO_2$ µg $h^{-1}$g dry $soil^{-1}$." These methods are described in further detail in Lemoine et al. 2024, which is cited earlier in that paragraph.

Lemoine, N. P., Budny, M. L., Rose, E., Lucas, J., & Marshall, C. W. Seasonal soil moisture thresholds inhibit bacterial activity and decomposition during drought in a tallgrass prairie. Oikos, 2024, e10210, https://doi.org/10.1111/oik.10201, 2024.

Line 182 – an alternative is to set values to ½ detection limit.

AUTHORS' RESPONSE: We have re-run the analyses using 0.005 mg/L as the replacement value for any zeros in the soil ion concentration data, as that is half of the detection limit (0.10 mg/L). The tables with these updated results are appended at the end of this document, but this

update did not result in any changes to statistical significance or model performance in the results. We have updated the methods as follows:

"Plant available P, the proportion of water-soluble P in soil that is available for uptake by plants, was extracted from 4 g of soil and 30 ml extraction fluid (1:7.5 ratio) using an acid–fluoride solution (P Bray-1), measured colorimetrically using a Systea EasyChem200 analyser, and expressed as mg/kg. The detection limit was 0.5 mg/kg, and plant available P measurements <0.5 mg/kg were replaced with half the detection limit (0.25 mg/kg)…………… Detection limits for soil ions were 0.01 mg/L, and soil ion concentrations measured as <0.01 mg/L were replaced with half the detection limit (0.005 mg/L)."

Line 188 – I understand that there were insufficient carcasses to allow inclusion of carcass age in models. However, major differences would be expected between carcasses aged 1 month vs 2.5 years. Is there any way to provide an indication of the magnitude of the age effect? How would distance effects look if young carcasses were excluded, for example?

AUTHORS' RESPONSE: Thanks for bringing this up. We made a figure showing soil ions ammonium, nitrate, and phosphate) and respiration potential plotted against carcass age. In these four cases, it is clear that these soil metrics are higher at fresher carcasses. In fact, the trends are so compelling that we have added this figure to the main text (Figure 5). This figure suggests the pattern of elevated soil nutrients that we found may be even stronger when considering younger carcasses given how quickly the nutrients decline with age.

Line 228 – It is perhaps not surprising that P concentrations showed little variation with distance, given that P was measured in water extracts (i.e. the extraction is recovering a relatively small pool of soluble P).

AUTHORS' RESPONSE: This is an interesting point. The results included in this manuscript are soil phosphate (1:2 water extract) and plant-available P (P Bray-I), but we also measured mineral phosphorus (P31) in mg/kg using the same method we did for soil Na, Mg, Fe, Ca, and K—microwave-assisted digestion in a solution of 9 mL 65% nitric acid ($HNO_3$) and 3 mL 32% hydrochloric acid (HCl) (full methodological details appended to the end of this document). We have added this result to the manuscript (Figure 2G; Table S1), thereby increasing the proportion of total soil P covered by our analyses. All three individual measurements of soil P (phosphate, plant-available P, and mineral P) indicate an interaction between distance and soil type in which P decreases with distance from the carcass center, but only in granitic soils.

Line 230 – how is plant-available P defined here?

AUTHORS' RESPONSE: Plant-available P refers to P measured via the the P Bray I method. We have updated the methods as follows: "Plant available P was extracted from 4 g of soil and

30 ml extraction fluid (1:7.5 ratio) using an acid–fluoride solution (P Bray-1), measured colorimetrically using a Systea EasyChem200 analyser, and expressed as mg/kg. The detection limit was 0.5 mg/kg, and plant available P measurements <0.5 mg/kg were replaced with half the detection limit (0.25 mg/kg)."

Line 286 – Soil extractable nutrients should be expressed on the basis of dry soil, not volume.

AUTHORS' RESPONSE: We have updated the manuscript so that here and throughout, soil ion concentrations are given in mg/kg soil rather than mg/L. The updated tables and figures are appended to the end of this document.

Line 286 - these very high extractable nitrogen concentrations are presumably in part a consequence of soils being frozen prior to analysis. Another factor is time between sampling and freezing - or storage prior to freezing. Please provide a statement about sample treatment prior to analysis (time from sampling to freezing, storage conditions during this time, etc, as relevant).

AUTHORS' RESPONSE: We have updated the methods section to include more details on soil storage as follows: "Soil samples were stored in a cooler during fieldwork. On the day they were collected, we used 5 g of each soil sample for soil respiration measurements (described below). The rest of each sample was stored plastic bags in a -20°C freezer until nutrient analyses; they were stored in coolers with ice blocks during the transition from the freezer at the field site to the freezers at the labs." As we discuss above, there were likely negligible, if any, impacts of freezing on nitrate concentrations given the data available in the literature. Similarly, the impacts on ammonium were likely also minor. Further, we have also compared our measured nitrogen concentrations to those from other studies performed in Kruger (% N, Aranibar et al. 2003; ammonium and nitrate, Rughöft et al. 2016; see table below). We used the 15m distance in our dataset for comparison, since that is most representative of the baseline nitrogen concentrations. Our values were similar to those found by other researchers in Kruger, which increases our confidence that the measurements we took are accurate. The nitrogen concentrations that we found at the 0-5m distances are indeed much higher than those found elsewhere, but we attribute this to the presence of the elephant carcass.

| **Metric** | **Source** | **Mean** | **Range** | **Method** |
|---|---|---|---|---|
| Soil %N | Reed et al. | 11.4% | 5 – 16% | Stable isotope analysis |
| | Aranibar et al. 2003 | | ~5 – 23% | Stable isotope analysis |
| Soil Nitrate | Reed et al. | 57.1 mg/kg | 11.1 – 95.7 mg/kg | 1:2 water extract analysis |
| | Rughöft et al. 2016 | 28.9 mg/kg | 0.0 – 121.9 mg/kg | 2:5 water extract analysis |
| Soil Ammonium | Reed et al. | 1.38 mg/kg | 0.01 – 6.5 mg/kg | 1:2 water extract analysis |
| | Rughöft et al. 2016 | 11.3 mg/kg | 0.7 – 33.3 mg/kg | 2:5 water extract analysis |
| Soil Plant-Available P | Reed et al. | 2.20 mg/kg | 0.01 – 9.62 mg/kg | P Bray I |
| | Craine et al. 2008 | 38.52 mg/kg | 3.23 – 85.43 mg/kg | P Bray II |
| Leaf N:P Ratio | Reed et al. | 7.0 | 2.5 – 13.9 | |

| | | | | |
|---|---|---|---|---|
| | Craine et al. 2008 | 5.8 | 3.2 – 9.2 | |

*For Reed et al., we used values from the 15m distance, and for Craine et al. 2008, we used values from the control plots, as in both cases these best represent the background levels of soil nutrients.

Aranibar, J. N., Macko, S. A., Anderson, I. C., Potgieter, A. L. F., Sowry, R. & Shugart, H. H. Nutrient cycling responses to fire frequency in the Kruger National Park (South Africa) as indicated by stable isotope analysis. Isotopes Environ. Health Stud., 39, 141-158, 2003.

Rughöft, S., Hermann, M., Lazar, C. S., Cesarz, S., Levick, S. R., Trumbore, S. E. & Küsel, K. Community composition and abundance of bacterial, archaeal and nitrifying populations in savanna soils on contrasting bedrock material in Kruger National Park, South Africa. Frontiers in Microbiology, 7, 2016.

Craine et al. Nutrient concentration ratios and co-limitation in South African grasslands. New Phytologist, 179, 829-836, 2008.

Line 308 – the high available P and tissue N:P ratios (see below) indicate that there is no P limitation here. This might limit the likely influence of carcasses on foliar P, as found here (i.e. foliar P is not a strong indicator of the extent to which carcasses change P availability in general).

AUTHORS' RESPONSE: Thanks for bringing this up. We appreciate your comments here and below suggesting that P limitation might not be strong at our sites, and we have updated this paragraph in the discussion as follows:

"Elevated soil phosphate (Figure 2E) and plant-available P (Figure 2F) at the center of carcass sites were also consistent with expectations from the literature (Bump et al. 2009a; Parmenter & MacMahon, 2009). However, elevated P levels in soil did not translate to elevated P in grass leaves (Figure 4C), which could suggest a lag between trends in soil and plants that is longer for P than for N. This lag could occur because phosphate easily forms chemical bonds with other soil ions (e.g., iron and aluminum in acidic soils and calcium in basic soils). Nitrate does not form these bonds and therefore has greater water solubility and soil mobility (Wiersum, 1962; Arai & Sparks, 2007). However, it is also possible that P limitation in Kruger is not as strong as it is in some other African savanna systems (Pellegrini, 2016). The foliar N:P ratios measured in this experiment were consistently low, indicating that N limitation is more likely (Güsewell, 2004)."

We also compared our plant-available P and foliar N:P ratios with other studies in Kruger and found that our plant-available P results were actually lower, while our foliar N:P ratios were in about the same range.

| Metric | Source | Mean | Range | Method |
|---|---|---|---|---|
| Soil Plant-Available P | Reed et al. | 2.20 mg/kg | 0.01 – 9.62 mg/kg | P Bray I |
| | Craine et al. 2008 | 38.52 mg/kg | 3.23 – 85.43 mg/kg | P Bray II |

| | | | | |
|---|---|---|---|---|
| Leaf N:P Ratio | Reed et al. | 7.0 | 2.5 – 13.9 | |
| | Craine et al. 2008 | 5.8 | 3.2 – 9.2 | |

Craine et al. Nutrient concentration ratios and co-limitation in South African grasslands. New Phytologist, 179, 829-836, 2008.

Line 322 – water-extractable P represents a tiny proportion of the total soil P, so reliance on this procedure probably limits the possibility of detecting change in soil P.

AUTHORS' RESPONSE: We have updated the results to include mineralized P in soils in addition to phosphate and plant-available P, as described above in response to line 228.

Line 323- 330 – this text largely repeats results.

AUTHORS' RESPONSE: Thank you for this feedback. The goal of this section is to demonstrate the impacts of soil ammonium on grass productivity and succession. We have edited to these lines to remove some of the repetition of results and instead focus on the main point that *U. mosambicensis,* one of the only grass species found at the center of carcass sites, may have a higher degree of ammonium tolerance than some sympatric grass species but may still be limited by the extreme ammonium levels at the centers of very fresh carcass sites.

Line 340 - missing here is a discussion of the ecological consequences of the findings. What are the implications for plant and microbial ecology in savanna ecosystems?

AUTHORS' RESPONSE: Thanks for this feedback. We will update the discussion section to include a paragraph on the importance of these nutrient hotspots on plant and microbial diversity and functioning. We will include comparisons with well-known nutrient hotspots in Kruger (e.g., termite mounds; Davies, Baldeck, & Asner, 2016) as well as in other tropical/sub-tropical systems, such as those recommended by the reviewer.

Davies, A. B., Levick, S. R., Robertson, M. P., van Rensburg, B. J., Asner, G. P. & Parr, C. L. Termite mounds differ in their importance for herbivores across savanna types, seasons and spatial scales, 2016.

Figure 2B, C – these values should be presented in mg/kg soil.

AUTHORS' RESPONSE: We have updated the manuscript so that here and throughout, soil ion concentrations are given in mg/kg soil rather than mg/L. To do this conversion, we multiplied the

volume in mg/L by 2 based on the 1:2 soil to water extraction ratio, as the reviewer helpfully described below. The updated tables and figures are appended to the end of this document.

Figure 2B – these are extremely high nitrate concentrations, even out to 15 m. For example, 100 mg/L is equivalent to 200 mg/kg based on a 1:2 soil to water extraction ratio. Extractions done quickly after sampling and in 2 M KCl are in the range of 1-5 mg/kg. This seems to be a clear indication of storage effects.

AUTHORS' RESPONSE: We have found our N measurements to be consistent with other studies in Kruger. As we discuss in several places above, there is unlikely to be any significant storage effects on these measurements.

| **Metric** | **Source** | **Mean** | **Range** | **Method** |
|---|---|---|---|---|
| Soil %N | Reed et al. | 11.4% | 5 – 16% | Stable isotope analysis |
| | Aranibar et al. 2003 | | ~5 – 23% | Stable isotope analysis |
| Soil Nitrate | Reed et al. | 57.1 mg/kg | 11.1 – 95.7 mg/kg | 1:2 water extract analysis |
| | Rughöft et al. 2016 | 28.9 mg/kg | 0.0 – 121.9 mg/kg | 2:5 water extract analysis |
| Soil Ammonium | Reed et al. | 1.38 mg/kg | 0.01 – 6.5 mg/kg | 1:2 water extract analysis |
| | Rughöft et al. 2016 | 11.3 mg/kg | 0.7 – 33.3 mg/kg | 2:5 water extract analysis |
| Soil Plant-Available P | Reed et al. | 2.20 mg/kg | 0.01 – 9.62 mg/kg | P Bray I |
| | Craine et al. 2008 | 38.52 mg/kg | 3.23 – 85.43 mg/kg | P Bray II |
| Leaf N:P Ratio | Reed et al. | 7.0 | 2.5 – 13.9 | |
| | Craine et al. 2008 | 5.8 | 3.2 – 9.2 | |

*For Reed et al., we used values from the 15m distance, and for Craine et al. 2008, we used values from the control plots, as in both cases these best represent the background levels of soil nutrients.

Craine et al. Nutrient concentration ratios and co-limitation in South African grasslands. New Phytologist, 179, 829-836, 2008.

Aranibar, J. N., Macko, S. A., Anderson, I. C., Potgieter, A. L. F., Sowry, R. & Shugart, H. H. Nutrient cycling responses to fire frequency in the Kruger National Park (South Africa) as indicated by stable isotope analysis. Isotopes Environ. Health Stud., 39, 141-158, 2003.

Rughöft, S., Hermann, M., Lazar, C. S., Cesarz, S., Levick, S. R., Trumbore, S. E. & Küsel, K. Community composition and abundance of bacterial, archaeal and nitrifying populations in savanna soils on contrasting bedrock material in Kruger National Park, South Africa. Frontiers in Microbiology, 7, 2016.

Figure 2B, C – are these values as $NO_3$/$NH_4$ or on an N basis?

AUTHORS' RESPONSE: These values are on an N-basis (as are the values from Rughöft et al. 2016 in the tables above, and we have updated the figure legend to clarify.

Figure 2D – please express stable isotope ratios as $\delta^{15}N$. This may be how the results are presented, but this is not clear from the units.

AUTHORS' RESPONSE: We have changed the notation here and throughout the manuscript to $\delta^{15}N$.

Figure 2F – these are very high available P concentrations for a natural ecosystem, although there is no mention of the method used.

AUTHORS' RESPONSE: We have updated the methods section to include the method used to measure plant-available P as follows: "Plant available P, the proportion of water-soluble P in soil that is available for uptake by plants, was extracted from 4 g of soil and 30 ml extraction fluid (1:7.5 ratio) using an acid–fluoride solution (P Bray-1), measured colorimetrically using a Systea EasyChem200 analyser, and expressed as mg/kg. The detection limit was 0.5 mg/kg, and plant available P measurements <0.5 mg/kg were replaced with half the detection limit (0.25 mg/kg)."

In comparison with the literature, we found that our soil plant-available P values were actually lower than those from other studies performed in Kruger on the same granitic and basaltic soil types.

| **Metric** | **Source** | **Mean** | **Range** | **Method** |
|---|---|---|---|---|
| Soil Plant-Available P | Reed et al. | 2.20 mg/kg | 0.01 – 9.62 mg/kg | P Bray I |
| | Craine et al. 2008 | 38.52 mg/kg | 3.23 – 85.43 mg/kg | P Bray II |

*For Reed et al., we used values from the 15m distance, and for Craine et al. 2008, we used values from the control plots, as in both cases these best represent the background levels of soil nutrients.

Craine et al. Nutrient concentration ratios and co-limitation in South African grasslands. New Phytologist, 179, 829-836, 2008.

Figure 4 – it looks like foliar N:P ratios are around 4 (2% N, 0.5%P) – these very low values that suggest strong N limitation. This is incompatible with the very high nitrate values presented in Figure 2. This further indicates storage problems with N measurements.

AUTHORS' RESPONSE: The reviewer brings up an interesting point about the stoichiometry of the grass we sampled around the elephant carcasses. We now analyze foliar N:P ratios and found that they were overall higher in granitic soils compared to basaltic soils and decreased with

distance from the carcass center in both soil types. The median foliar N:P ratio was 9.38 at the 0m distance and 4.83 at the 15m distance, which may indicate that N limitation may be relatively stronger further from the carcass site, and P limitation may be relatively stronger closer to the center. These new results have been added to the manuscript (Figure 4D, Table S2) and are appended to the end of this document. Interestingly, some previous work in Kruger with this same species of grass, as well as other grasses, showed that N:P ratios in grasses responded similarly to the addition of N fertilizer (Crane et al. 2008). Under control nutrient conditions the grasses in their study had a N:P of 5.8 on average. But similar to our study, under N fertilization grasses had a N:P of 9.9 on average. These data argue against the reviewer's point that having high N availability in the soil necessarily increases the N:P to high levels that one would expect to indicate P limitation. This example, and the data we present above, suggest that the storage methods for the soils that we used are not impacting our dataset.

Craine et al. Nutrient concentration ratios and co-limitation in South African grasslands. New Phytologist, 179, 829-836, 2008.

Augustine, D. J. (2003). Long-term, livestock-mediated redistribution of nitrogen and phosphorus in an East African savanna. Journal of Applied Ecology, 40(1), 137-149.

Hudson et al. (2009). Temporal patterns of nutrient availability around nests of leaf-cutting ants (Atta*olombica*) in secondary moist tropical forest. Soil Biology and Biochemistry, 41(6), 1088-1093.

Turner, B. L., & Romero, T. E. (2009). Short-term changes in extractable inorganic nutrients during storage of tropical rain forest soils. Soil Science Society of America Journal, 73, 1972-1979.

AUTHORS' RESPONSE: Thank you for providing these references. We have found the Turner and Romero (2009) manuscript particularly instructive in interpretating the soil nitrogen data. We have updated the discussion to include comparisons with other types of savanna nutrient hotspots such as those found in the Augustine (2003) and Hudson et al. (2009) papers.

**Revised Methods**

**2.1 Study system and sample collection**

We performed this research in the southern part of the Kruger National Park (KNP), South Africa (24.996 S, 31.592 E, ~275m elevation). The two dominant soil types in KNP are granitic soils (inceptisols) and basaltic soils (versitols or andisols) (Khomo et al. 2017). The clay-rich basaltic soils have relatively large surface area, enabling them to retain larger quantities of water than granitic soils, which drain water more quickly and therefore are lower in water-soluble nutrients (Buitenweref, Kulmatiski, & Higgins, 2014; Rughöft et al. 2016). The landscape at KNP is a mix of savanna grasslands and broadleaf woodlands, with an overstory dominated by trees from the genus *Combretum* (red bushwillow, *C. apiculatum*; russet bushwillow, *C. hereroense*; leadwood, *C. imberbe*) and trees formerly known as acacias (knobthorn, *Senegalensis nigrescens*; umbrella thorn, *Vachellia tortillis*). The park hosts a full suite of African savanna animals, including ~30,000 elephants (*Loxodonta africana*) (Coetsee & Ferreira, 2023), with a mortality rate of ~2% (~600 elephants per year). The targeted region of KNP has a high density of scavengers and predators, including white-backed vultures (*Gyps africanus*), spotted hyenas (*Crocuta crocuta*), and lions (*Panthera leo*) (Owen-Smith & Mills, 2007).

During the wet season in March 2023, we identified ten elephant carcass sites (1-26 months post-death), five on relatively nutrient-rich basaltic soil and five on nutrient-poor granitic soil. KNP section rangers provided precise GPS locations of where elephant carcasses had been found. Most elephants died of old age, illness, injury, or, in the case of one young bull, territorial fighting. These sites were recognizable *in situ* by a persistent bonefield, undigested gut contents, and an absence of herbaceous vegetation. At each site, we hammered a rebar post into the center of the megacarcass disturbance and ran 15 m transects out from the post in each of the four

cardinal directions. Based on pilot data, we treat the 10-15m distances as controls, sine the high degree of landscape heterogeneity in the system (e.g., differences in hill slope, vegetation, water drainage, proximity to termite mounds) made random transects difficult for interpretation. 
[revised manuscript text omitted]

[Figure]

Relationship between carcass age and key metrics (soil ion concentrations and respiration potential). (A) Soil ammonium, (B) soil nitrate, (C) soil phosphate, and (D) soil respiration potential are all higher at fresher carcass sites. Point respresent values at the center of the carcass site (distance = 0-0.5m).

**Figure 5.** Relationship between carcass age and key soil metrics (soil ion concentrations and respiration potential). (A) Soil ammonium, (B) nitrate, (C) phosphate, and (D) respiration potential are all higher at fresher carcass sites. Points represent individual measurements taken at the center of the carcass site (distance = 0-0.5m).

**Revised Supplemental Tables**

**Table S1.** Generalized linear mixed model results for soil variables. The same five models were run for each response variable, including a null model, and each included site as a random effect to account for repeat measurements. AICc is Akaike's Information Criterion, and ΔAICc is the difference between a given model and the best fit model for that response variable. Cum.Wt stand for cumulative weight; it gives the sum of Akaike's weights and indicates the likelihood that the models up to that point are the best in the set. Models with a ΔAICc value of 2 are considered roughly equivalent in fit and are italicized. Marginal $R^2$ is the proportion of variance explained by both fixed and random effects in a model, and conditional $R^2$ is the proportion of variance explained by fixed effects. Coefficients (± standard error) are shown for each predictor and model and are in log units. Rows are organized in blocks by response variable. Within blocks, models are listed in order of increasing ΔAICc.

| **Model** | **Model Fit** | | | | | **Coefficients ± *SE*** | | |
|---|---|---|---|---|---|---|---|---|
| | **AICc** | **ΔAICc** | **Cum.Wt** | **Mar. $R^2$** | **Con. $R^2$** | **Soil** | **Distance** | **Soil × Distance** |
| **Nitrogen (%)** | | | | | | | | |
| *Soil × Distance* | *-227.32* | *0.00* | *0.99* | *0.54* | *0.74* | *-0.26 ± 0.22* | *0.00 ± 0.01* | *-0.03 ± 0.01* |
| Soil + Distance | -216.13 | 11.20 | 1.00 | 0.46 | 0.67 | -0.48 ± 0.21 | -0.01 ± 0.00 | |
| Distance | -214.95 | 12.37 | 1.00 | 0.04 | 0.52 | | -0.01 ± 0.00 | |
| Soil | -212.36 | 14.97 | 1.00 | 0.40 | 0.62 | -0.47 ± 0.21 | | |
| Null | -211.23 | 16.09 | 1.00 | | | | | |
| **δ15N** | | | | | | | | |
| *Soil × Distance* | *180.87* | *0.00* | *0.77* | *0.55* | *0.70* | *0.39 ± 0.16* | *-0.02 ± 0.01* | *-0.02 ± 0.01* |
| Soil + Distance | 184.66 | 3.79 | 0.88 | 0.50 | 0.66 | 0.26 ± 0.15 | -0.03 ± 0.00 | |
| Distance | 184.67 | 3.79 | 1.00 | 0.34 | 0.60 | | -0.03 ± 0.00 | |
| Soil | 219.35 | 38.47 | 1.00 | 0.20 | 0.34 | 0.28 ± 0.14 | | |
| Null | 219.96 | 39.09 | 1.00 | | | | | |
| **Nitrate (mg/kg)** | | | | | | | | |
| *Distance* | *624.84* | *0.00* | *0.70* | *0.48* | *0.52* | | *-0.14 ±* 0.02 | |

| | | | | | | | | |
|---|---|---|---|---|---|---|---|---|
| Soil + Distance | 627.06 | 2.23 | 0.93 | 0.48 | 0.52 | -0.14 ± 0.27 | -0.14 ± 0.02 | |
| Soil × Distance | 629.51 | 4.67 | 1.00 | 0.48 | 0.52 | -0.24 ± 0.39 | -0.14 ± 0.03 | 0.02 ± 0.04 |
| Null | 649.77 | 24.93 | 1.00 | | | | | |
| Soil | 651.82 | 26.99 | 1.00 | 0.01 | 0.04 | -0.18 ± 0.31 | | |
| **Ammonium (mg/kg)** | | | | | | | | |
| *Soil + Distance* | *219.52* | *0.00* | *0.65* | *0.58* | *0.77* | *2.49* ± 0.66 | *-0.18* ± 0.03 | |
| *Soil × Distance* | *220.94* | *1.43* | *0.97* | *0.60* | *0.77* | *2.91* ± 0.73 | *-0.15* ± 0.04 | *-0.07* ± 0.06 |
| Distance | 225.87 | 6.35 | 1.00 | 0.21 | 0.77 | | -0.18 ± 0.02 | |
| Soil | 244.57 | 25.05 | 1.00 | 0.34 | 0.70 | 2.51 ± 0.76 | | |
| Null | 249.38 | 29.86 | 1.00 | | | | | |
| **Phosphate (mg/kg)** | | | | | | | | |
| *Soil × Distance* | *167.99* | *0.00* | *0.98* | *0.52* | *0.79* | *2.20 ± 0.96* | *0.00 ± 0.05* | *-0.46 ± 0.08* |
| Soil + Distance | 178.68 | 10.69 | 1.00 | 0.18 | 0.18 | -0.38 ± 0.70 | -0.14 ± 0.06 | |
| Null | 180.65 | 12.66 | 1.00 | | | | | |
| Soil | Model did not converge | | | | | | | |
| Distance | Model did not converge | | | | | | | |
| **Plant Available Phosphorus (mg/kg)** | | | | | | | | |
| *Soil × Distance* | *447.18* | *0.00* | *0.94* | *0.34* | *0.63* | *0.16 ± 0.62* | *-0.04 ± 0.03* | *-0.13 ± 0.04* |
| Distance | 453.68 | 6.50 | 0.98 | 0.20 | 0.55 | | -0.10 ± 0.02 | |
| Soil + Distance | 454.80 | 7.62 | 1.00 | 0.26 | 0.55 | -0.66 ± 0.55 | -0.11 ± 0.02 | |
| Null | 467.35 | 20.17 | 1.00 | | | | | |
| Soil | 469.19 | 22.01 | 1.00 | 0.03 | 0.30 | -0.35 ± 0.47 | | |
| **Mineral Phosphorus (mg/kg)** | | | | | | | | |
| *Soil × Distance* | *537.77* | *0.00* | *1.00* | *0.86* | *0.95* | *-1.09 ± 0.32* | *0.00 ± 0.00* | *-0.04 ± 0.01* |
| Soil + Distance | 560.48 | 22.71 | 1.00 | 0.82 | 0.92 | -1.35 ± 0.31 | -0.02 ± 0.00 | |
| Distance | 566.38 | 28.61 | 1.00 | 0.04 | 0.76 | | -0.02 ± 0.00 | |
| Soil | 573.55 | 35.78 | 1.00 | 0.78 | 0.89 | -1.33 ± 0.31 | | |
| Null | 579.62 | 41.85 | 1.00 | | | | | |
| **Sodium (mg/kg)** | | | | | | | | |
| *Soil × Distance* | *438.56* | *0.00* | *0.73* | *0.29* | *0.59* | *0.22 ± 0.35* | *-0.03 ± 0.01* | *-0.04 ± 0.02* |
| Distance | 441.09 | 2.53 | 0.94 | 0.22 | 0.54 | | -0.05 ± 0.00 | |
| Soil + Distance | 443.53 | 4.97 | 1.00 | 0.22 | 0.54 | -0.06 ± 0.35 | -0.05 ± 0.01 | |
| Null | 464.02 | 25.45 | 1.00 | | | | | |

| | | | | | | | | |
|---|---|---|---|---|---|---|---|---|
| Soil | 466.38 | 27.82 | 1.00 | 0.00 | 0.34 | 0.00 ± 0.00 | | |
| **Potassium (mg/kg)** | | | | | | | | |
| *Soil × Distance* | *676.07* | *0.00* | *0.94* | *0.29* | *0.81* | *-0.23 ± 0.42* | *0.01 ± 0.00* | *-0.02 ± 0.01* |
| Null | 682.93 | 6.86 | 0.97 | | | | | |
| Soil | 684.55 | 8.48 | 0.99 | 0.25 | 0.78 | -0.37 ± 0.41 | | |
| Distance | 685.17 | 9.10 | 1.00 | 0.00 | 0.72 | | 0.00 ± 0.00 | |
| Soil + Distance | 686.89 | 10.82 | 1.00 | 0.26 | 0.78 | -0.37 ± 0.41 | 0.00 ± 0.00 | |
| **Calcium (mg/kg)** | | | | | | | | |
| *Soil* | *749.09* | *0.00* | *0.60* | *0.82* | *0.94* | *-1.45 ± 0.41* | | |
| *Soil + Distance* | *751.01* | *1.92* | *0.83* | *0.82* | *0.94* | *-1.45 ± 0.01* | *0.00 ± 0.00* | |
| Soil × Distance | 753.00 | 3.91 | 0.91 | 0.82 | 0.94 | -1.42 ± 0.41 | 0.00 ± 0.01 | -0.01 ± 0.01 |
| Null | 753.55 | 4.46 | 0.97 | | | | | |
| Distance | 755.37 | 6.27 | 1.00 | 0.00 | 0.81 | | 0.00 ± 0.00 | |
| **Iron (mg/kg)** | | | | | | | | |
| *Soil* | *914.44* | *0.00* | *0.67* | *0.88* | *0.96* | *-1.22 ± 0.28* | | |
| Soil + Distance | 916.83 | 2.39 | 0.87 | 0.88 | 0.96 | -1.22 ± 0.28 | 0.00 ± 0.00 | |
| Soil × Distance | 918.54 | 4.10 | 0.95 | 0.88 | 0.96 | -1.19 ± 0.28 | 0.00 ± 0.00 | 0.00 ± 0.01 |
| Null | 920.27 | 5.83 | 0.99 | | | | | |
| Distance | 922.55 | 8.11 | 1.00 | 0.00 | 0.82 | | 0.00 ± 0.00 | |
| **Magnesium (mg/kg)** | | | | | | | | |
| *Soil* | *700.88* | *0.00* | *0.63* | *0.87* | *0.96* | *-1.53 ± 0.37* | | |
| Soil + Distance | 703.33 | 2.45 | 0.81 | 0.87 | 0.96 | -1.53 ± 0.37 | 0.00 ± 0.00 | |
| Soil × Distance | 703.97 | 3.09 | 0.95 | 0.88 | 0.96 | -1.48 ± 0.37 | 0.00 ± 0.00 | -0.01 ± 0.01 |
| Null | 706.40 | 5.52 | 0.99 | | | | | |
| Distance | 708.75 | 7.87 | 1.00 | 0.00 | 0.84 | | 0.00 ± 0.00 | |
| **Water (mmol/mol)** | | | | | | | | |
| *Null* | *111.87* | *0.00* | *0.32* | | | | | |
| *Distance* | *112.09* | *0.22* | *0.61* | *0.03* | *0.38* | | *0.02 ± 0.01* | |
| *Soil* | *112.92* | *1.05* | *0.80* | *0.12* | *0.40* | *0.45 ± 0.38* | | |
| *Soil + Distance* | *113.27* | *1.40* | *0.96* | *0.14* | *0.42* | *0.45 ± 0.38* | *0.02 ± 0.01* | |
| Soil × Distance | 115.86 | 3.99 | 1.00 | 0.14 | 0.42 | 0.44 ± 0.42 | 0.02 ± 0.02 | 0.00 ± 0.03 |
| **pH** | | | | | | | | |
| *Soil × Distance* | *55.04* | *0.00* | *0.37* | *0.07* | *0.44* | *0.05 ± 0.07* | *0.00 ± 0.00* | *-0.01 ± 0.00* |
| *Null* | *55.26* | *0.22* | *0.71* | | | | | |

| | | | | | | | | |
|---|---|---|---|---|---|---|---|---|
| *Distance* | *56.94* | *1.90* | *0.86* | *0.01* | *0.38* | | *0.00 ± 0.00* | |
| Soil | 57.63 | 2.59 | 0.96 | 0.00 | 0.37 | 0.00 ± 0.07 | | |
| Soil + Distance | 59.41 | 4.37 | 1.00 | 0.01 | 0.38 | 0.00 ± 0.00 | 0.00 ± 0.00 | |

**Table S2.** Generalized linear mixed model results for leaf variables. The same five models were run for each response variable, including a null model, and each included site as a random effect to account for repeat measurements. AICc is Akaike's Information Criterion, and ΔAICc is the difference between a given model and the best fit model for that response variable. Cum.Wt stand for cumulative weight; it gives the sum of Akaike's weights and indicates the likelihood that the models up to that point are the best in the set. Models with a ΔAICc value of 2 are considered roughly equivalent in fit and are italicized. Marginal $R^2$ is the proportion of variance explained by both fixed and random effects in a model, and conditional $R^2$ is the proportion of variance explained by fixed effects. Coefficients (± standard error) are shown for each predictor and model and are in log units. Rows are organized in blocks by response variable. Within blocks, models are listed in order of increasing ΔAICc.

| **Model** | **Model Fit** | | | | | **Coefficients ± *SE*** | | |
|---|---|---|---|---|---|---|---|---|
| | **AICc** | **ΔAICc** | **Cum.Wt** | **Mar. $R^2$** | **Con. $R^2$** | **Soil** | **Distance** | **Soil × Distance** |
| **Nitrogen (%)** | | | | | | | | |
| *Distance* | *56.12* | *0.00* | *0.64* | *0.40* | *0.60* | | *-0.03 ±* 0.00 | |
| Soil + Distance | 57.79 | 1.67 | 0.92 | 0.43 | 0.61 | 0.13 ± 0.14 | -0.03 ± 0.00 | |
| Soil × Distance | 60.33 | 4.20 | 1.00 | 0.43 | 0.61 | 0.15 ± 0.15 | -0.03 ± 0.01 | 0.00 ± 0.01 |
| Null | 89.78 | 33.66 | 1.00 | | | | | |
| Soil | 91.66 | 35.53 | 1.00 | 0.03 | 0.21 | 0.10 ± 0.13 | | |
| **δ15N** | | | | | | | | |
| *Soil × Distance* | *229.95* | *0.00* | *0.95* | *0.51* | *0.77* | *-0.52 ± 0.43* | *-0.11 ± 0.01* | *0.06 ± 0.02* |
| Distance | 236.55 | 6.60 | 0.99 | 0.44 | 0.70 | | -0.08 ± 0.01 | |
| Soil + Distance | 238.97 | 9.02 | 1.00 | 0.45 | 0.70 | -0.12 ± 0.40 | -0.08 ± 0.01 | |
| Null | 282.45 | 52.50 | 1.00 | | | | | |
| Soil | 284.30 | 54.34 | 1.00 | 0.04 | 0.36 | -0.30 ± 0.41 | | |
| **Phosphorus (%)** | | | | | | | | |
| *Soil × Distance* | -87.04 | 0.00 | 0.99 | 0.47 | 0.75 | -0.24 ± 0.31 | 0.02 ± 0.01 | -0.04 ± 0.01 |
| Soil | -76.10 | 10.94 | 1.00 | 0.38 | 0.68 | -0.55 ± 0.31 | | |
| Null | -75.98 | 11.06 | 1.00 | | | | | |

| | | | | | | | | |
|---|---|---|---|---|---|---|---|---|
| Soil + Distance | -73.69 | 13.34 | 1.00 | 0.38 | 0.68 | -0.55 ± 0.31 | 0.00 ± 0.01 | |
| Distance | -73.68 | 13.36 | 1.00 | 0.00 | 0.56 | | 0.00 ± 0.01 | |
| **N:P Ratio** | | | | | | | | |
| *Soil × Distance* | 209.64 | 0.00 | 0.86 | 0.41 | 0.71 | 0.34 ± 0.38 | -0.05 ± 0.01 | 0.04 ± 0.01 |
| Distance | 214.60 | 4.96 | 0.94 | 0.09 | 0.59 | | -0.03 ± 0.01 | |
| Soil + Distance | 214.85 | 5.21 | 1.00 | 0.36 | 0.67 | 0.62 ± 0.01 | -0.03 ± 0.00 | |
| Null | 225.74 | 16.10 | 1.00 | | | | | |
| Soil | 226.21 | 16.57 | 1.00 | 0.23 | 0.57 | 0.55 ± 0.37 | | |
| **Sodium (mg/kg)** | | | | | | | | |
| *Soil + Distance* | *839.97* | *0.00* | *0.60* | *0.62* | *0.78* | *-0.99 ± 0.32* | *-0.03 ± 0.01* | |
| *Soil × Distance* | *841.56* | *1.59* | *0.88* | *0.62* | *0.79* | *-0.88 ± 0.34* | *-0.03 ± 0.01* | *-0.02 ± 0.01* |
| Distance | 843.18 | 3.21 | 1.00 | 0.09 | 0.64 | | -0.03 ± 0.01 | |
| Soil | 852.98 | 13.02 | 1.00 | 0.53 | 0.71 | -1.00 ± 0.32 | | |
| Null | 856.49 | 16.52 | 1.00 | | | | | |
| **Magnesium (mg/kg)** | | | | | | | | |
| *Soil × Distance* | *722.20* | *0.00* | *0.99* | *0.45* | *0.80* | *-0.20 ± 0.28* | *0.00 ± 0.00* | *-0.02 ± 0.01* |
| Distance | 731.74 | 9.54 | 0.99 | 0.07 | 0.66 | | -0.01 ± 0.00 | |
| Soil + Distance | 732.78 | 10.58 | 1.00 | 0.39 | 0.76 | -0.36 ± 0.28 | -0.01 ± 0.00 | |
| Null | 743.56 | 21.36 | 1.00 | | | | | |
| Soil | 744.46 | 22.26 | 1.00 | 0.31 | 0.69 | -0.37 ± 0.28 | | |
| **Potassium (mg/kg)** | | | | | | | | |
| *Distance* | *936.99* | *0.00* | *0.73* | *0.20* | *0.57* | | *-0.03 ± 0.00* | |
| Soil + Distance | 939.50 | 2.51 | 0.94 | 0.20 | 0.57 | 0.02 ± 0.25 | -0.03 ± 0.00 | |
| Soil × Distance | 941.96 | 4.97 | 1.00 | 0.20 | 0.57 | 0.05 ± 0.26 | -0.02 ± 0.01 | 0.00 ± 0.01 |
| Null | 956.55 | 19.57 | 1.00 | | | | | |
| Soil | 958.95 | 21.96 | 1.00 | 0.00 | 0.38 | 0.00 ± 0.24 | | |
| **Calcium (mg/kg)** | | | | | | | | |
| *Null* | *799.64* | *0.00* | *0.42* | | | | | |
| *Distance* | *800.68* | *1.04* | *0.67* | *0.01* | *0.50* | | *0.00 ± 0.00* | |
| *Soil* | *801.22* | *1.58* | *0.86* | *0.14* | *0.53* | *-0.20 ± 0.21* | | |
| Soil + Distance | 802.36 | 2.72 | 0.96 | 0.14 | 0.54 | -0.20 ± 0.21 | 0.00 ± 0.00 | |
| Soil × Distance | 804.45 | 4.81 | 1.00 | 0.15 | 0.54 | -0.16 ± 0.22 | 0.01 ± 0.01 | -0.01 ± 0.01 |
| **Iron (mg/kg)** | | | | | | | | |
| *Distance* | *591.87* | *0.00* | *0.69* | *0.21* | *0.57* | | *-0.08 ± 0.01* | |

| | | | | | | | | |
|---|---|---|---|---|---|---|---|---|
| Soil + Distance | 594.14 | 2.27 | 0.92 | 0.23 | 0.58 | -0.26 ± 0.50 | -0.08 ± 0.01 | |
| Soil × Distance | 596.15 | 4.27 | 1.00 | 0.23 | 0.59 | -0.09 ± 0.39 | -0.07 ± 0.00 | -0.02 ± 0.02 |
| Null | 616.95 | 25.08 | 1.00 | | | | | |
| Soil | 619.06 | 27.19 | 1.00 | 0.02 | 0.48 | -0.31 ± 0.00 | | |

**Table S3.** Generalized linear mixed model results testing for correlations between leaf and soil micronutrients. The same model was run for each of five micronutrients (Na, K, Ca, Mg, and Fe) with leaf micronutrient concentration as the response variable, soil micronutrient + distance as the main effects, and site as a random effect. Marginal $R^2$ is the proportion of variance explained by both fixed and random effects in a model, and conditional $R^2$ is the proportion of variance explained by fixed effects. Coefficients (± standard error) are shown for each predictor and model.

| **Leaf Micronutrient** | **Mar. $R^2$** | **Con. $R^2$** | **Soil Micronutrient Coefficient ± *SE*** | **Distance Coefficient ± *SE*** |
|---|---|---|---|---|
| Sodium | 0.08 | 0.82 | 11.56 ± 11.67 | -146.47 ± 43.04 |
| Potassium | 0.29 | 0.73 | 0.00 ± 0.00 | -0.06 ± 0.01 |
| Calcium | 0.12 | 0.58 | 0.00 ± 0.00 | 0.00 ± 0.00 |
| Magnesium | 0.17 | 0.79 | 0.00 ± 0.00 | 0.00 ± 0.00 |
| Iron | 0.11 | 0.32 | 0.00 ± 0.01 | -52.85 ± 20.57 |

**Revised Supplemental Figures**

A B

[Figure]

**Figure S1.** Representative photos of two elephant carcass sites of different ages and soil types. (A) The first site is 67 days post-death and is on granitic soil. (B) The second site is 811 days post-death and is on basaltic soil. In both images, there is a visible impact zone with reduced vegetation coverage. At the first site, elephant bones have all been dispersed, though some are still present at the second site. Photos taken by Deron Burkepile at time of sample collection in March 2023.

[Figure]

**Figure S2.** (A) Soil micronutrient composition did not differ significantly with distance from the carcass but (B) was distinct in different soil types.

[Figure]

**Figure S3.** Effects of elephant carcasses on soil micronutrients. (A) Soil sodium decreased significantly with distance from the carcass. (B) Potassium decreased with distance but only in granitic soils. (C) Iron, (D) magnesium, and (E) calcium were greater in basaltic soils. Distance appeared in the top model for calcium, but the effect size was minimal. Points represent individual measurements taken at 0, 2.5, 5, 10, and 15m and are offset to be visible when they would otherwise overlap. Lines show predictions calculated from the top model. Shading indicates the 95% confidence interval.

[Figure]

**Figure S4**. Neither (A) soil water nor (B) soil pH differed with distance or soil type. Points represent individual measurements taken at 0, 2.5, 5, 10, and 15m and are offset to be visible when they would otherwise overlap.

[Figure]

**Figure S5.** (A) Foliar micronutrient composition did not differ significantly with distance from the carcass but (B) was distinct in different soil types.

[Figure]

**Figure S6**. Effects of elephant carcasses on grass foliar micronutrients. (A) Foliar Na and (B) Mg were greatest in basaltic soil and decreased significantly with distance. (C) Foliar K and (D) Fe decreased with distance but did not differ with soil type. (E) Foliar Ca did not differ with distance or soil type. Points represent individual measurements taken at 0, 2.5, 5, 10, and 15m and are offset to be visible when they would otherwise overlap. Lines show predictions calculated from the top model. Shading indicates the 95% confidence interval.

---

## Author Response (AR1)

**Response to Feedback from Reviewer 1**

**Summary**

Animals impact elemental cycling in many direct and indirect ways. Evidence from several biomes demonstrates that even after death, animal carcasses can change the biogeochemistry of ecosystems and these impacts can be long lasting. Most studies of carcass impacts on ecosystems, however, are done on small to medium (1kg to 200kg) sized animals. In this contribution, the authors investigate the effects of elephant megacarcasses on the biogeochemistry of soils and plants. The authors report significant effects of elephant carcasses on components of soil and plant elemental cycling and they discuss how these effects may be important components of spatiotemporal heterogeneity in ecosystems.

**General comments**

1) Overall, I found the writing good. The authors have crafted a nice narrative that makes a compelling case that megacarcasses can be important parts of ecosystems and therefore we need to learn more about the impacts of these carcasses on ecosystems.

AUTHORS' RESPONSE: Thank you! We appreciate your kind words and thoughtful review.

2) I have a few questions about the analysis. The effective sample size is 10. Obviously, it is hard to find carcasses (I would have great difficulty in finding 10 fresh moose carcasses in my system!) but the authors are trying to squeeze a lot of information out of very few data points. I have the following specific questions about the analysis:

i) While I like the transect approach, the design may have been stronger if the authors had random transects (ie, transects with no known carcass) like Risch et al. work. This would strengthen inference.

AUTHORS' RESPONSE: Thanks for the suggestion, and we definitely appreciate the value of control/random sites. In fact, our original plan was to use random transects as controls (Risch et al. 2020), but during a pilot experiment we realized that high landscape heterogeneity (differences in hill slope, vegetation, water drainage, proximity to termite mounds, etc.), all of which have implications for nutrient distribution across the landscape (Venter et al. 2003; Holdo & McDowell, 2004), made the random transects challenging for interpretation as controls. Instead, we looked at our pilot data to see whether there was a consistent size of the impact site and found that soil nutrients were elevated until about 5-8m away from the center of the carcass site. Past this 5-8m radius, soil nutrients dropped to consistently lower levels, indicative of background concentrations. Thus, we designed the sampling scheme of 0.5m, 2.5m, 5m, 10m, and 15m distances away from the carcass site to capture both the impact of the elephant carcass and the background ("control") concentration of soil nutrients (at the 10m and 15m distance). There was never a significant difference in nutrient concentrations between the 10 and 15m distances, suggesting our sampling scheme successfully captured the transition from the influence of the elephant carcass through to the background level of nutrients in the matrix soils.

We have updated the methods section of the manuscript as follows (lines 146-150): "We treated the 10-15m distances as representative of background concentrations of nutrients based on pilot data showing that the effect of elephant carcasses on soil nutrient concentrations was undetectable at this distance away from the carcass site, similar to studies on the carcasses of other large vertebrates (e.g., Towne, 2000; Bump et al. 2009)."

Venter FJ, Scholes RJ, Eckhardt HC. Abiotic template and its associated vegetation pattern. In: JT Du Toit, KH Rogers, HC Biggs, eds. The Kruger experience: ecology and management of savanna heterogeneity. Washington, DC, USA: Island Press, 83–129, 2003.

Holdo, R. M. & McDowell, L. R. Termite mounds as nutrient-rich food patches for elephants. *Biotropica*, 36, 231-239, https://doi.org/10.1111/j.1744-7429.2004.tb00314.x, 2004.

ii) lines 180-183. The author's approach to checking for normality of response data does not seem sound to me. The assumption of normality (for linear models) is normality in light of the model, i.e., investigating the normality of residuals is a more common approach to this. Either way, it is often better to avoid transforming the data and generalized linear models do allow for a lot of flexibility to fit different error distributions. For example, the gamma family in glm is very flexible and can handle log-normal data sets. Did the authors try different families of error distributions before transforming their data?

AUTHORS' RESPONSE: Thanks so much for this suggestion. We have revised our analysis and implemented the gamma family (link = log) in for all of our models now instead of log-transforming and have updated the text in the methods accordingly. We have updated all results accordingly. Even with this change in the structure of the analyses, the major patterns across the different analyses did not change. In fact, these changes actually strengthen the major patterns in the results showing the importance of elephant carcasses in savanna nutrient dynamics.

We have updated the methods as follows (lines 253-256): "For each response variable, we ran five generalized linear mixed models using the gamma family (link = log) in the package *lme4* (Bates et al. 2015): (*i*) soil type + distance + soil type × distance interaction, (*ii*) soil type + distance, (*iii*) soil type, (*iv*) distance, and (*v*) a null model indicating no significant difference in slope or intercept after accounting for carcass site."

iii) lines 187-189. How many data points did the authors have per estimated parameter in the most complex model here?

AUTHORS' RESPONSE: For each of these models, we had 50 observations total (10 sites x 5 samples per site). In our most complicated model, that averages to ~17 observations per parameter, which is above the recommended 10 observations per parameter (Burnham & Anderson, 2002).

We have updated the methods as follows (lines 257-258): "Each model included 50 observations (10 sites x 5 distances per site)."

iv) line 194. This is fine but I think Burham & Anderson would say that any model within deltaAIC of 2 of the null model should not be considered to be supported. In several cases, the authors interpret top models that are ranked above the null but within deltaAIC of 2 of the null as supported (e.g., lines 217-218, 218-221).

AUTHORS' RESPONSE: With the updated model structure (see response to 2.iii), there are now only three response variables (soil water, soil pH, and foliar calcium) for which the null and another model fall within a ΔAICc value of 2 (Table S1). In all three cases, we will interpret this as the results not supporting a relationship between the response variable and soil type, distance from the carcass, or soil type by distance interaction.

v) what $R^2$ are the authors reporting? In the captions of Tables S1 and S2 (thank you for providing full AIC and coefficient tables), the authors state "$R^2$ is the proportion of variance explained by a model". This is unclear. These are mixed models, and the most common approach is to report the marginal $R^2$ and conditional $R^2$. Is the $R^2$ in these tables one of those or another pseudo $R^2$? This is critical for many reasons but most importantly, given the small sample size and large number of mixed-models, I would expect at least one of the models to not converge. There are many indicators when a mixed-model does not converge and one of the best is when the marginal $R^2$ = conditional $R^2$. Without having both of these pieces of information, the reader is unable to adequately assess the fit of the models. Other indicators of models not converging are coefficients estimates or errors that are very large or very small (i.e., 0 – see next comment).

AUTHORS' RESPONSE: We have updated supplemental tables to include both marginal and conditional $R^2$ values. The only place where we had issues with model non-convergence was soil phosphate; two of the five models failed to converge and are indicated in Table S1.

vi) I am confused by the magnitude of Table S2 sodium and iron coefficients and/or the scale of reported on the y-axis of Figure 5 for these. The iron coefficients in Table S2 seem small relative to the Figure 5c? Or am I misreading things?

AUTHORS' RESPONSE: The original models for soil and leaf micronutrients used log-transformed data, which meant that the coefficients and standard errors were in log units as well. When plotting, we back-transformed the data to make the axis scales easier to interpret, which is why the values in the table and the figures were different. The updated models (see response to 2.iii) still use a log link, so the model outputs in the updated tables are still in log units as well. Again in this version, we exponentiated when calculating the prediction lines so that we could plot them with the raw data, which we believe is visually more intuitive than figures with axes on the log scale. We have updated the table captions for clarification as follows: "Coefficients (± standard error) are shown for each predictor and model and are in log units."

3) the reporting of results could be improved. I recommend, the authors report: top ranked models (AIC + measure of independent fit like $R^2$). Then report effect size or relationships

(coefficients). I found key statistics to be missing throughout. Statements like "Phosphate concentrations were greater in granitic soils…" would be more informative if they included the coefficient + error in parenthesis. Coefficients can be reported for the top-ranked model or from model averaged results when there are several competing models.

AUTHORS' RESPONSE: We appreciate this feedback and have updated the tables accordingly, including AIC values, marginal and conditional $R^2$, and coefficients + standard error (see below). We considered including coefficients for the top-ranked models in the text of the results section. However, because the coefficients are in log units, we found that they were not biologically intuitive in a string of text, reducing the clarity and ease of reading. Thus, instead of including coefficients in the text, we refer readers to the appropriate tables for statistical results, where the coefficients are easier to interpret in their full context.

4) in section 3.2 I think the reader may be more interested in coefficients and confidence intervals around those relationships than p-values that are currently reported.

AUTHORS' RESPONSE: We have removed the p-values from this section and refer readers to the appropriate table, as described above in response to comment 3.

**Specific comments:**

5) I found the use of three different terms that mean similar things (nutrient flows, ecosystem processes, nutrient availability) in the introductory sentence confusing. I recommend the authors replace "nutrient availability" with "ecosystem processes" or "nutrient flows". Surely living animals (not just carcasses) influence nutrient availability (which is just a part of a continual nutrient cycle).

AUTHORS' RESPONSE: We have edited that line for consistency in phrasing (lines 51-53): "Living animals affect nutrient flows through ecosystems (Schmitz et al. 2018), but we have only recently acknowledged that the nutrients from animal carcasses could also influence ecosystem processes (Barton et al. 2013; Monk et al. 2024).

6) line 83. I believe there is no "e" at the end of the citation Risch et al.

AUTHORS' RESPONSE: We have corrected the citation. Thanks for catching it!

7) lines 96-111. How do these elephants die? As someone with no experience with megacarcasses, I would appreciate some insight on the causes of death. Most large herbivore deaths in my empirical systems are from predation which I assume is not the case for elephants.

AUTHORS' RESPONSE: We received GPS coordinates for carcasses from KNP rangers, who also keep record of the cause of death for each elephant. The reviewer is right that predation tends not to be a major issue for elephants, and none that we know of died from it. Most of the elephants in our dataset died of natural causes such as old age, illness, injury, or, in the case of one young bull, a territorial dispute that ended in his death.

We have updated the methods section as follows (lines 139-140): "Most elephants died of old age, illness, injury, or, in the case of one young bull, fighting over territory."

8) really excellent job with clear hypotheses and nice work carrying forward these hypotheses throughout the ms – really makes the job easier for the reader.

AUTHORS' RESPONSE: Thank you!

9) lines 132-133 Why 10cm deep core? Is that mineral soil only?

AUTHORS' RESPONSE: We used a 10cm core to ensure that we captured the soil surface horizon. It is a commonly used depth and is more conservative than shallower sampling. Prior work on the soil impacts of carcasses uses this depth (Bump, Peterson, & Vucetich, 2009; Monk et al. 2024). Moreover, previous work in the same system has shown that soil auger sampling depths of 7.5-10cm are sufficient for detecting differences in N, C, and soil micronutrients (Gray & Bond 2015, Holdo & Mack 2014). We have added these citations to the methods (lines 145-146).

Bump, J. K., Peterson, R. O., & Vucetich, J. A. Wolves modulate soil nutrient heterogeneity and foliar nitrogen by configuring the distribution of ungulate carcasses. *Ecology*, 90, 3159–3167, https://doi.org/10.1890/09-0292.1, 2009.

Monk, J. D., Donadio, E., Smith, J. A., Perrig, P. L., Middleton, A. D., & Schmitz, O. J. Predation and biophysical context control long-term carcass nutrient inputs in an Andean ecosystem. *Ecosystems*, 27, 346–359, https://doi.org/10.1007/s10021-023-00893-7, 2024.

Gray, E. F. & Bond, W. Soil nutrients in an African forest/savanna mosaic: Drivers or driven? *South African Journal of Botany*, 101, 66-72, https://doi.org/10.1016/j.sajb.2015.06.003, 2015.

Holdo, R. M. & Mack, M. C. Functional attributes of savanna soils: contrasting effects of tree canopies and herbivores on bulk density, nutrients and moisture dynamics. *Journal of Ecology*, 102, 1171-1182. https://doi.org/10.1111/1365-2745.12290, 2014.

10) the discussion is well done – concise and touches on all hypotheses.

AUTHORS' RESPONSE: Thank you!

11) Figure 1 is an outstanding visual!

AUTHORS' RESPONSE: Thank you!

12) in figures 2-5 I recommend the authors consider reminding the reader of the sampling resolution because the jitter of points makes it impossible to see what distances were measured below 5m.

AUTHORS' RESPONSE: We have updated the captions in figures 2-5 to include sampling resolution as follows: "Points represent individual measurements taken at 0, 2.5, 5, 10, and 15m and are offset to be visible when they would otherwise overlap."

**Response to Feedback from Reviewer 2**

Summary:

Reed and coauthors present a well-written study on the impacts of megacarcasses (elephants) to soil biogeochemistry after up to 2 years of decomposition. The authors examined 10 carcass hotspots with 5 carcasses each on two different soil types. They quantified soil major and trace element chemistry as well as plants associated with the hotspots to determine if carcasses influenced soil N and P chemistry and if those elements were subsequently enriched in vegetation. The current version of this manuscript does not adequately describe the methods in enough detail to make the work reproduceable. Additionally, the handling of the data for statistical analyses is strange and non-standard. The discussion needs to be re-written to better emphasize the importance of the work (as framed in the introduction). I think this work has potential to be an important contribution, but there needs to be some major revisions.

AUTHORS' RESPONSE: Thanks for all of your feedback. We have made substantial changes in response to your suggestions, including rewriting the methods section to include more details on the lab analyses. We have also updated our statistical analyses to use the gamma family of generalized linear mixed models, which allows us to run non-normally distributed data without the log transformation. These changes add to the methodological clarity and statistical robustness of this research, and they actually strengthen the major patterns in the results showing the importance of elephant carcasses in savanna nutrient dynamics. Finally, we have updated the text to ensure that the functional distinctiveness of megacarcass relative to smaller carcasses carries through from the introduction to the discussion.

General comments:

- The importance of this study in adding to our knowledge about nutrient transfer at carrion hotspots is not emphasized clearly in the discussion. The introduction frames how megacarcasses may be "functionally different than smaller carcasses" but never returns to this aspect in the discussion, which is really where this work could add to our knowledge. Adding more to the discussion would help address this issue and would make the impact of the work clearer.

AUTHORS' RESPONSE: Thanks for the suggestion. To more clearly link our results to the functional differences between megacarcasses and smaller carrion, we have updated the discussion as follows (lines 441-451): "The magnitude of nutrient inputs from megacarcasses, as well as the substantial size and duration of their impact zones, means their impacts on ecosystem processes may be functionally distinct from smaller carrion. Indeed, there is evidence that carcass size strongly impacts scavenger food web structure (Moleón et al. 2015; Morris et al. 2023). Moreover, the attraction of animals to carcasses via scavenging, predation, or mourning (Goldenberg & Wittemyer, 2020) could have positive feedbacks on nutrient cycling (Bump, Peterson, & Vucetich, 2009; Monk et al. 2024), which may be magnified by carcass size. Thus, the impacts of megacarcasses on savanna ecosystem processes may be dissimilar to the effects of small carrion and more similar to other more persistent contributors to savanna ecosystem

processes, such as termite mounds (Davies et al. 2016), cattle bomas (Augustine, 2003), and even mass animal mortality events (Subalusky et al. 2017, 2020)."

- Parts of the results belong in the discussion, and I've tried to highlight those below in specific comments.

AUTHORS' RESPONSE: Thanks for pointing this out. We have edited and/or removed those sentences from the results section and focused on them in the discussion, as described in response to specific comments below.

- The methods need significantly more specific details, highlighted in specific comments.

AUTHORS' RESPONSE: We have substantially updated the methods section in the manuscript (section 2.3) to include specific details on the lab analysis methods, and we address specific comments below as well.

- Additionally, there were no control soil or plant samples examined here. Please describe in the methods why there were no controls.

AUTHORS' RESPONSE: The reviewer raises an important point here, and we definitely appreciate the value of control/random sites that other studies of carrion have used. In fact, our original plan was to use random transects as controls (Risch et al. 2020), but during a pilot experiment we realized that high landscape heterogeneity (differences in hill slope, vegetation, water drainage, proximity to termite mounds, etc.), all of which have implications for nutrient distribution across the landscape (Venter et al. 2003; Holdo & McDowell, 2004), made the random transects challenging for interpretation as controls. Instead, we looked at our pilot data to see whether there was a consistent size of the impact site and found that soil nutrients were elevated until about 5-8m away from the center of the carcass site. Past this 5-8m radius, soil nutrients dropped to consistently lower levels, indicative of background concentrations. Thus, we designed the sampling scheme of 0.5m, 2.5m, 5m, 10m, and 15m distances away from the carcass site to capture both the impact of the elephant carcass and the background ("control") concentration of soil nutrients (at the 10m and 15m distance). There was never a significant difference in nutrient concentrations between the 10 and 15m distances, suggesting our sampling scheme successfully captured the transition from the influence of the elephant carcass through to the background level of nutrients in the matrix soils.

We have updated the methods section of the manuscript as follows (lines 146-150): "We treated the 10-15m distances as representative of background concentrations of nutrients based on pilot data showing that the effect of elephant carcasses on soil nutrient concentrations was undetectable at this distance away from the carcass site, similar to studies on the carcasses of other large vertebrates (e.g., Towne, 2000; Bump et al. 2009)."

Venter, F. J., Scholes, R. J. & Eckhardt, H. C. Abiotic template and its associated vegetation pattern. In: J. T. Du Toit, K. H. Rogers, H. C. Biggs, eds. The Kruger

experience: ecology and management of savanna heterogeneity. Washington, DC, USA: Island Press, 83–129, 2003.

Holdo, R. M. & McDowell, L. R. Termite mounds as nutrient-rich food patches for elephants. *Biotropica,* 36, 231-239, https://doi.org/10.1111/j.1744-7429.2004.tb00314.x, 2004.

The handling of the data for statistical analyses is non-standard and not clearly justified. If data were non-normally distributed (it seems like some datasets were and some were not), why not just use a non-parametric statistical test rather than log-transforming the data? It is a bit strange to log-transform some data but not all. The approach of adding 0.001 to zero values is also not correct (described below in specific comments).

AUTHORS' RESPONSE: We have updated our model selection procedure to use the gamma family with a log link for our generalized linear mixed models rather than transforming the data beforehand (see RC1, 2.ii).

We have re-run the analyses using 0.005 mg/L as the replacement value for any zeros in the soil ion concentration data, as that is half of the detection limit (0.10 mg/L) and have updated the results section, tables, and figures accordingly. This update did not result in any changes to statistical significance or model performance in the results. We have updated the methods as follows (lines 169-179):

"Plant-available P was extracted from 4 g of soil and 30 ml extraction fluid (1:7.5 ratio) using an acid–fluoride solution (P Bray-1), measured colorimetrically using a Systea EasyChem200 analyser, and expressed as mg/kg. The detection limit was 0.5 mg/kg, and plant available P measurements <0.5 mg/kg were replaced with half the detection limit (0.25 mg/kg) (Croghan & Egeghy, 2003; Keenan & Beeler, 2023). Water-soluble nitrate and phosphate anions were extracted from volume on volume 100 ml soil and 200 ml deionized water, analyzed by ion chromatography on a Metrohm 930 Compact Flex System, and measured as mg/L. Ammonium (also 1:2 water extract) was analyzed colorimetrically using a Systea EasyChem200 analyzer and measured as mg/L. Detection limits for soil ions were 0.01 mg/L, and soil ion concentrations measured as <0.01 mg/L were replaced with half the detection limit (0.005 mg/L)."

- The presentation of elemental data for soil and plant composition is non-standardized throughout. Some data (i.e., iron) are presented as mg/kg (is this soil dry weight?), while others are presented as % (Ca% of what?) in the same figure (figure 5 for example). Other data are presented as mg/L (figure 2). Part of this confusion is from the missing details in the methods that clearly explain how these data were generated. In several of the figures there is a statement about back-transformed data, which is also confusing.

AUTHORS' RESPONSE: We appreciate the attention to detail here from the reviewer. We have updated the manuscript so that soil ions, soil anions, and foliar micronutrients are all

in mg/kg. Soil and foliar nitrogen are given as the percentage of dry weight that is nitrogen, as this is the standard unit of measurement for the instrument used (IRMS). We appreciate you pointing this out and agree that including these methodological details aids greatly in interpretation.

With regards to the comment on back-transformed data, we originally were log-transforming the nutrient data prior to analysis. For aid in visual interpretation of the results, we had displayed the data in its original units. Now that we have updated the model structure and are no longer log-transforming before analysis, we have removed mentions of back-transformation from the manuscript.

- I can appreciate that finding carcasses that have decomposed for the same amount of time is challenging, but 1 month to 26 months is a huge range of time (at least from what we know from not megacarcasses). The biogeochemical processes occurring at a carcass decaying after 1 month postmortem is very different than a carcass that has been decaying for 26 months (from smaller carcasses). It would be useful to see some of the data, particularly ammonium, plotted as a function of postmortem interval (months) even if that is not a variable that could be included in statistical analyses because of the small sample size. It would also be helpful to see if the postmortem interval for the 10 carcasses is evenly distributed between the two soil types or if one has more fresh carcasses and the other has older carcasses, that could help with interpretation of the results.

AUTHORS' RESPONSE: Thanks for the suggestion! We have added analysis and figures to the manuscript testing for a relationship between key soil metrics and carcass age. We found that soil ammonium, phosphate, and respiration potential all decrease significantly with carcass age. In fact, the trends are so compelling that we have added this figure to the main text (Figure 5). This figure suggests the pattern of elevated soil nutrients that we found may be even stronger when considering younger carcasses given how quickly the nutrients decline with age.

We ran a t-test to test for a difference in mean carcass age across soil types and found no significant difference between the two groups ($P = 0.294$).

We have added these updates to the methods section as follows (lines 280-283): "Finally, to test the impact of carcass age on key soil metrics, we ran exponential decay functions for soil ammonium, nitrate, phosphate, and respiration verses carcass age for samples from the center of the carcass site (0.5m sampling location). We also performed a t-test to verify that there was no difference in mean carcass age across soil types."

And to the results section as follows (lines 343-347): "Soil ammonium, phosphate, and respiration potential all decreased significantly with carcass age (Figure 5A-C). The exponential decay model for nitrate failed to converge due to an outlier with extremely high soil nitrate (1454 mg/kg) at 258 days post-death (Figure 5D). We ran a t-test to test for a difference in mean carcass age across soil types and found no significant difference between the two groups ($P = 0.294$)."

• I think it may be useful to add some photos to supplemental information (or even the main text) showing what the carcasses/sites looked like (maybe representative images from a fresher carcass and one that is older).

AUTHORS' RESPONSE: Thanks for the suggestion! We have added a supplemental figure (Figure S1) that shows two carcass sites – one is fresh and on basaltic soil, and the other is older and on granitic soil.

Specific comments:

• Lines 99-100: There should be more details provided on the soil type and what makes the granitic soils "nutrient poor" compared to soils developed from a basalt protolith. Because soil type becomes an important part of this study, the details of the soil types need to be expanded in the introduction.

AUTHORS' RESPONSE: Thanks for the suggestion. Kruger National Park has two primary soil types – a clay-rich soil derived from basalt ("basaltic") and a sandy soil derived from granite ("granitic"). Basaltic soils have clay particles with relatively large surface area, thereby enabling them to retain larger quantities of water than granitic soils, which drain water more quickly and therefore are lower in water-soluble nutrients (Buitenweref, Kulmatiski, & Higgins, 2014).

We agree that these distinctions are important for understanding the impacts of carcass-derived nutrients on different soil types and have updated the methods section as follows (lines 122-126): "The two dominant soil types in KNP are granitic soils (inceptisols) and basaltic soils (versitols or andisols) (Khomo et al. 2017). The clay-rich basaltic soils have relatively large surface area, enabling them to retain larger quantities of water than granitic soils, which drain water more quickly and therefore are lower in water-soluble nutrients (Buitenweref, Kulmatiski, & Higgins, 2014; Rughöft et al. 2016)."

Khomo, L., Trumbore, S., Bern, C. R., & Chadwick, O. A. Timescales of carbon turnover in soils with mixed crystalline mineralogies. *SOIL*, 3, 17-30, https://doi.org/10.5194/soil-3-17-2017, 2017.

Buitenwerf, R., Kulmatiski, A. & Higgins, S. I. Soil water retention curves for the major soil types of the Kruger National Park. *Koedoe*, 56, a1228, http://dx.doi.org/10.4102/koedoe.v56i1.1228, 2014.

• Line 132: Include a citation or discuss why soil samples were collected to a depth of 10 cm rather than the upper 5 cm. For decomposition studies, typically the upper 5 cm is examined, not the upper 10 cm.

AUTHORS' RESPONSE: We used a 10cm core to ensure that we captured the soil surface horizon. It is a commonly used depth and is more conservative than shallower sampling. Prior work on the soil impacts of carcasses uses this depth (Bump, Peterson, & Vucetich, 2009; Monk et al. 2024). Moreover, previous work in the same system has shown that soil auger sampling depths of 7.5-10cm are sufficient for detecting differences in N, C, and soil micronutrients (Gray & Bond 2015, Holdo & Mack 2014). We have updated the text in the methods to include these references (lines 145-146).

Bump, J. K., Peterson, R. O., & Vucetich, J. A. Wolves modulate soil nutrient heterogeneity and foliar nitrogen by configuring the distribution of ungulate carcasses. *Ecol.,* 90, 3159–3167, https://doi.org/10.1890/09-0292.1, 2009.

Monk, J. D., Donadio, E., Smith, J. A., Perrig, P. L., Middleton, A. D., & Schmitz, O. J. Predation and biophysical context control long-term carcass nutrient inputs in an Andean ecosystem. *Ecosyst*., 27, 346–359, https://doi.org/10.1007/s10021-023-00893-7, 2024.

Gray, E. F. & Bond, W. Soil nutrients in an African forest/savanna mosaic: Drivers or driven? *S. Afr. J. Bot.,* 101, 66-72, https://doi.org/10.1016/j.sajb.2015.06.003, 2015.

Holdo, R. M. & Mack, M. C. Functional attributes of savanna soils: contrasting effects of tree canopies and herbivores on bulk density, nutrients and moisture dynamics. *J. Ecol.,* 102, 1171-1182, https://doi.org/10.1111/1365-2745.12290, 2014.

- Line 145: More details are needed beyond "measurements of soil ion concentrations". What instrumentation was used? What specific extraction protocol was followed? I'm assuming deionized water was used (1:2 soil to deionized water?), but those details are not provided. How long were samples mixed (shaking platform?), what speed, etc.

AUTHORS' RESPONSE: We have updated the relevant portion of the methods as follows (lines 166-179): "We sent 250 g of each soil sample to Eco-Analytica laboratory at the North-West University in Potchefstroom, South Africa for measurements of soil concentrations of ammonium $[NH_4]^+$, nitrate $[NO_3]^-$, phosphate $[PO_4]^{3-}$, and plant-available P. Samples were air-dried and sieved through < 2mm mesh prior to chemical analysis. Plant-available P was extracted from 4 g of soil and 30 ml extraction fluid (1:7.5 ratio) using an acid–fluoride solution (P Bray-1), measured colorimetrically using a Systea EasyChem200 analyser, and expressed as mg/kg. The detection limit was 0.5 mg/kg, and plant available P measurements <0.5 mg/kg were replaced with half the detection limit (0.25 mg/kg) (Croghan & Egeghy, 2003; Keenan & Beeler, 2023). Water-soluble nitrate and phosphate anions were extracted from volume on volume 100 ml soil and 200 ml deionized water, analyzed by ion chromatography on a Metrohm 930 Compact Flex System, and measured as mg/L. Ammonium (also 1:2 water extract) was analyzed colorimetrically using a Systea EasyChem200 analyzer and measured as mg/L. Detection limits for soil ions were 0.01 mg/L, and soil ion concentrations measured as <0.01 mg/L were replaced with half the detection limit (0.005 mg/L)."

- Line 148: "mass spectrometry"—elaborate on what this means with respect to instrumentation used to analyze cations. Here and throughout the methods, please also include what standards were used for the different analysis types.

AUTHORS' RESPONSE: We have updated the relevant section of the methods as follows (lines 182-197): "To determine whether soil anions were distinct and elevated at the center of carcass sites relative to soil further from the center, concentrations of sodium (Na), magnesium (Mg), iron (Fe), calcium (Ca), potassium (K), and phosphorus (P) cations were measured using microwave-assisted digestion. Air-dried and sieved (>2 mm) soil samples, weighed to 0.2 g, were microwaved in 9 ml 65% nitric acid ($HNO_3$) and 3 ml 32% hydrochloric acid (HCl) according to EPA 3051b in a Milestone, Ethos microwave digester with UP, Maxi 44 rotor. A period of 20 minutes allowed the system to reach 1800 MW at a temperature of 200 °C which was maintained for 15 minutes. After cooling, the samples were brought up to a final volume of 50 ml and analyzed on an Agilent 7500 CE ICP-MS fitted with CRC (Collision Reaction Cell) technology for interference removal. The instrument is optimized using a solution containing Li, Y, Ce, and Tl (1 ppb) for standard low-oxide/low interference levels ($\leq 1.5\%$) while maintaining high sensitivity across the mass range. The instrument was calibrated using ULTRASPEC® certified custom mixed multi-element stock standard solutions containing all the elements of interest (De Bruyn Spectroscopic Solutions, South Africa). Calibrations spanned the range of 0 – 30 ppm for the mineral elements Ca, Mg, Na, and K and 0 – 0.3 ppm for the rest of the trace elements. Elemental concentrations were expressed as mg/kg."

- Lines 146-150: Clarify if these analyses were conducted on the water extracts.

AUTHORS' RESPONSE: We have clarified as follows (lines 174-179): "Water-soluble nitrate and phosphate anions were extracted from volume on volume 100 ml soil and 200 ml deionized water, analyzed by ion chromatography on a Metrohm 930 Compact Flex System, and measured as mg/L. Ammonium (also 1:2 water extract) was analyzed colorimetrically using a Systea EasyChem200 analyzer and measured as mg/L. Detection limits for soil ions were 0.01 mg/L, and soil ion concentrations measured as <0.01 mg/L were replaced with half the detection limit (0.005 mg/L)."

- Line 152: Were stable isotope analyses conduct on oven-dried soil? 10 g is an exceptionally large amount of soil—how much was actually analyzed with EA-IRMS? Were samples powdered prior to combustion?

AUTHORS' RESPONSE: We have clarified as follows (lines 202-204): "Samples were oven-dried at 60°C for 48 hours and milled to a fine powder using a Retsch MM400 mill (Germany). The powdered samples were weighed (2 – 60 mg) prior to combustion at 950°C."

- Line 154 (and throughout with respect to stable nitrogen isotope results): The authors refer to "$^{15}$N" measurements, but surely this should be presented as the ratio of 15/14N and in delta notation? In the methods here there also needs to be more description of

the standard, the materials used for linearity, and the analytical precision of the instrument.

AUTHORS' RESPONSE: We have changed the notation throughout the manuscript to $\delta^{15}N$. We have updated the methods to include information on standards and precision as follows (lines 205-211): "A high organic carbon (HOC) soil standard (0.52 ± 0.02 %N), along with two international reference standards (USGS40 ($\delta^{15}N$ -4.52% AIR) and USGS41 ($\delta^{15}N$ +47.57% AIR)) were used for calibration. The N elemental content was expressed relative to atmospheric N as $N_2$ $\delta^{15}N_{AIR}$ (‰). The quantification limit for $\delta^{15}N$ on the IRMS is 1 nA (nanoAmp), and the quantification limit for %N is 0.06%. The precision for %N was 0.02% and for $\delta^{15}N$ is ±0.11%, determined using the HOC standard, which was run multiple times throughout the analysis."

- Line 175: More details on the ICP-MS are needed, including standards, detection limits, etc. Additionally, were these samples digested in nitric acid? Water? How long were they microwaved?

AUTHORS' RESPONSE: We have updated the relevant section of the methods as follows (lines 182-197): "To determine whether soil anions were distinct and elevated at the center of carcass sites relative to soil further from the center, concentrations of sodium (Na), magnesium (Mg), iron (Fe), calcium (Ca), potassium (K), and phosphorus (P) cations were measured using microwave-assisted digestion. Air-dried and sieved (>2 mm) soil samples, weighed to 0.2 g, were microwaved in 9 ml 65% nitric acid ($HNO_3$) and 3 ml 32% hydrochloric acid (HCl) according to EPA 3051b in a Milestone, Ethos microwave digester with UP, Maxi 44 rotor. A period of 20 minutes allowed the system to reach 1800 MW at a temperature of 200 °C which was maintained for 15 minutes. After cooling, the samples were brought up to a final volume of 50 ml and analyzed on an Agilent 7500 CE ICP-MS fitted with CRC (Collision Reaction Cell) technology for interference removal. The instrument is optimized using a solution containing Li, Y, Ce, and Tl (1 ppb) for standard low-oxide/low interference levels (≤ 1.5%) while maintaining high sensitivity across the mass range. The instrument was calibrated using ULTRASPEC® certified custom mixed multi-element stock standard solutions containing all the elements of interest (De Bruyn Spectroscopic Solutions, South Africa). Calibrations spanned the range of 0 – 30 ppm for the mineral elements Ca, Mg, Na, and K and 0 – 0.3 ppm for the rest of the trace elements. Elemental concentrations were expressed as mg/kg."

- Line 182: Adding some random number to each variable is not a standard way to handle data that are zero in your dataset (or if it is, there is no citation here and I am not familiar with that approach). Typically for geochemical data (like what was generated with ICP-MS), you can replace zero values with ½ the detection limit to remove non-zero data. There are other more technical ways to deal with zero values from a statistical standpoint, but the ½ the detection limit is the easiest and has the longest history of use. Please justify the use of your approach or re-run the analyses following a standard method for handling non-zero data in a geochemical dataset.

AUTHORS' RESPONSE: We have re-run the analyses using 0.005 mg/L as the replacement value for any zeros in the soil ion concentration data, as that is half of the detection limit (0.10 mg/L). We have updated the results accordingly, but this update did not result in any changes to statistical significance or model performance in the results. We have updated the methods as follows (lines 169-179): "Plant-available P was extracted from 4 g of soil and 30 ml extraction fluid (1:7.5 ratio) using an acid–fluoride solution (P Bray-1), measured colorimetrically using a Systea EasyChem200 analyser, and expressed as mg/kg. The detection limit was 0.5 mg/kg, and plant available P measurements <0.5 mg/kg were replaced with half the detection limit (0.25 mg/kg) (Croghan & Egeghy, 2003; Keenan & Beeler, 2023). Water-soluble nitrate and phosphate anions were extracted from volume on volume 100 ml soil and 200 ml deionized water, analyzed by ion chromatography on a Metrohm 930 Compact Flex System, and measured as mg/L. Ammonium (also 1:2 water extract) was analyzed colorimetrically using a Systea EasyChem200 analyzer and measured as mg/L. Detection limits for soil ions were 0.01 mg/L, and soil ion concentrations measured as <0.01 mg/L were replaced with half the detection limit (0.005 mg/L)."

- Line 255: The part of the sentence that reads "…we found evidence that N from carcasses had moved from soils into plants" does not belong in the results section. This is interpretation and should be moved to the discussion.

AUTHORS' RESPONSE: We have edited this sentence to read (lines 322-323): "Consistent with our third hypothesis, we found elevated foliar nutrient concentrations in *U. trichopus* at elephant carcass sites."

We include interpretation in the first paragraph of the discussion (lines 355-358): "Together, these results indicate that carcass-derived nutrients move into soil and subsequently get absorbed by plants over relatively short time scales, cycling essential nutrients such as N from carrion into the soil and then back into aboveground nutrient pools."

- Lines 256-258: Similar comment as above where the content of this sentence is interpretation and should be moved to the discussion.

AUTHORS' RESPONSE: We have deleted the following clause from that sentence: "….indicating that the high N content in leaves closer to the center of a megacarcass site likely had an animal origin."

- Lines 295-297: I'm not quite sure I understand the logic presented here. First, soil microbial biomass was not measured. The respiration potential (through production of $CO_2$) was measured, but heterotrophic activity (which is how respiration can be interpreted) consumes oxygen. I think the phrasing here needs to be re-worked to not imply that the soil respiration (and the communities producing $CO_2$) are not necessarily the same that are driving nitrification.

AUTHORS' RESPONSE: We have edited this sentence to focus on heterotrophic activity rather than nitrification. It now reads (lines 378-380): "Soil nitrate (Figure 2B) and soil respiration potential (Figure 3) were also elevated near the center of carcass sites, indicating higher rates of activity of heterotrophic microbes (Prosser, 2011).

- Lines 304-305: There are prior studies that demonstrate the impact of increased organic C during decomposition on soil microbial processes that should be cited here (see studies by DeBruyn and colleagues)

AUTHORS' RESPONSE: Thanks for the suggestion. We have added two relevant citations to these lines from DeBruyn and colleagues (lines 382-383).

Keenan, S. W., Schaeffer, S. M., Jin, V. L. & DeBruyn, J. M. Mortality hotspots: nitrogen cycling in forest soils during vertebrate decomposition. *Soil Biol. Biochem.,* 121, 165-176, https://doi.org/10.1016/j.soilbio.2018.03.005, 2018.

Keenan, S. W., Schaeffer, S. M., and DeBruyn, J. M. Spatial changes in soil stable isotopic composition in response to carrion decomposition, *Biogeosciences,* 16, 3929–3939, https://doi.org/10.5194/bg-16-3929-2019, 2019.

- Lines 310-311: I'm not sure that this is phrased correctly. Phosphorus (predominantly as phosphate) is considered immobile in soil partly because of low solubility because it is often sorbed with Ca, Fe, Al or organics, and the release of P is tightly controlled by soil (or fluid) pH. N does not face the same sorts of sorption immobilization constraints. I think if you rephrased it to clarify that P and N are held within different reservoirs within soils that make them behave differently (and add some citations), that would help.

AUTHORS' RESPONSE: Thanks for the feedback. We have expanded that section to better explain the differences in how N and P are held in soils and agree that this adds clarity. The lines now read (lines 398-401): "This lag could occur because phosphate easily forms chemical bonds with other soil ions (e.g., iron and aluminum in acidic soils and calcium in basic soils). Nitrate does not form these bonds and therefore has greater water solubility and mobility in soils and may be more readily taken up by plants (Wiersum, 1962; Arai & Sparks, 2007)."

We have also added a second citation:

Aria, Y. & Sparks, D. L. Phosphate reaction dynamics in soils and soil components: a multiscale approach. *Adv. Agron.*, 94, 135-179, https://doi.org/10.1016/S0065-2113(06)94003-6, 2007.

- Lines 324-328: As mentioned above, because the composition of the two soil types were not included, this part of the discussion is not supported by the results. It's unclear if the authors here are trying to say that the basaltic soils contain more nutrients after being impacted by decomposition or if the native state of the soils (background conditions) are more nutrient rich. I think if the introduction described the background

chemistry of the two soil types this would be better supported. Additionally, basalt and granite contain different types of minerals and therefore additional sources of elements like P. I don't know what the specific mineralogy is of these two rock types in KNP, but it might be worth exploring. In particular, the presence of apatite (Ca-P bearing mineral) in the granite might also be contributing to elevated P measured in the granite soils.

AUTHORS' RESPONSE: Thanks for bringing this up. We have updated the text to clarify that our results confirm a well-established pattern in the literature – that the background state of basaltic soils is more nutrient-rich than granitic soils. What we find interesting here is the significant interaction between soil type and distance with regards to ammonium and phosphate concentrations (Table S1; Figure 2). Ammonium and phosphate levels are elevated at the center of carcass sites in granitic soils relative to basaltic soils, but the difference between soil types disappears as distance from the carcass site increases. These results suggest that the impact of elephant carcasses on these soil ions is greater in the nutrient-poor granitic soils relative to basaltic soils.

We have updated the text in the discussion to better explain the importance of this soil type by distance interaction (lines 424-431): "Our results confirmed the previously-established trend that basaltic soils are overall more cation rich than granitic soils, with greater concentrations of P, K, Fe, Mg, and Ca (Figure 2G; Figure S3B-E; Gertenbach, 1983; Craine, Morrow, & Stock, 2008; Wigley et al. 2014). However, soil ammonium, $\delta^{15}$N, and phosphate were all higher in the granitic soils towards the center of carcass sites, decreasing steeply to be similar to basaltic soils about 10 m from the carcass center (Figure 2C-E). These results indicate that the impact of organic matter from megacarcasses may be stronger in relatively nutrient-poor and sandy granitic soil compared with nutrient-rich and clayey basaltic soil."

We have also updated the methods (lines 122-126) to include more information on the background differences between the two soil types, as described above.

- Figure 5 (and others): I'm a little bit confused by the figure caption. I think it would be better to present this as selected elements plotted as a function of distance. These are not the results of generalized linear mixed models, but the selection of how to present the data were informed by the models.

AUTHORS' RESPONSE: We have changed the text in the figure captions to clarify that we are plotting all the parameters that were included in the top model(s) for a given response variable (Tables S1-2), which is standard practice. We hope this change in language is clearer.

**Response to Feedback from Reviewer 3**

This manuscript presents data on the influence of elephant carcasses on nutrient availability in South African savanna soils. It would be a surprise if a decaying elephant did not increase nutrient concentrations in the proximity of the carcass, but there are some interesting differences among nutrients in terms of the distance over which the effects extend. There are parallels with other nutrient hotspots in tropical ecosystems, including glades in African savannas (e.g. Augustine 2003) and leafcutter ant nests in tropical forests (e.g. Hudson et al. 2009) - it would be worth introducing these into the discussion for comparison. I have several questions about methodology and results that should be addressed before this manuscript could be acceptable for publication.

AUTHORS' RESPONSE: Thank you for all of your feedback. We have made substantial changes in response to your suggestions, including rewriting the methods section to include more details on the soil lab analyses. We have updated the introduction to better explain the importance of cations and cation exchange in savanna soils. We have also updated the discussion to more thoroughly address the ecological significance of our results, including comparison with other nutrient hotspots.

The most significant critique throughout this review was concern that freezing the soil samples prior to ion analysis may have resulted in elevated ammonium and nitrate levels. We agree that this is an important concern, but we are reassured by the findings in the literature showing that nitrogen measurements (especially nitrate) are relatively robust to storage method. Indeed, in Turner and Romero (2009) (a paper the reviewer cites), freezing did not impact nitrate concentrations relative to fresh soils except in cases of high soil acidity, which were not present in our study. Other studies show that nitrate is unaffected by freezing treatment for at least seven weeks post collection, well within the time frame of analysis for our samples (Esala, 1995; Sollen-Norrlin & Rintoul-Hynes, 2024). Soil ammonium can increase when frozen, although the increase is often relatively small (<1 mg/kg per week frozen; Esala 1995, Fig. 1). It is true that freezing can have a large impact on ammonium in peaty soils, but freezing has only minimal effects on ammonium measurements in clay soils such as ours (Esala, 1995; Sollen-Norrlin & Rintoul-Hynes, 2024). Further, we have compared the soil % nitrogen, nitrate, and ammonium values from the 15m distance in our study (essentially representing the background levels of nutrients in the soils in Kruger) to those found in other soil analysis research in Kruger, and our values are consistent with that prior research (Aranibar et al. 2003; Rughöft et al. 2016; see table below). Our foliar N:P values were consistent with the Kruger literature as well. We found much higher soil nitrogen values at 0-5m distances from the carcass site than those found in these papers, but that is what we hypothesized would happen with nutrient inputs from elephant carcasses. Moreover, even if the absolute values of nitrogen in our study are elevated due to freezing, which there is little evidence to support, there is no reason that the effects of freezing would differ with distance from an elephant carcass, so we are confident that the overall trends in this manuscript are robust.

| Metric | Source | Mean | Range | Method |
|---|---|---|---|---|
| Soil N | Reed et al. | 11.4% | 5 – 16% | Stable isotope analysis |
| | Aranibar et al. 2003 | | ~5 – 23% | Stable isotope analysis |
| Soil Nitrate | Reed et al. | 57.1 mg/kg | 11.1 – 95.7 mg/kg | 1:2 water extract analysis |
| | Rughöft et al. 2016 | 28.9 mg/kg | 0.0 – 121.9 mg/kg | 2:5 water extract analysis |
| Soil Ammonium | Reed et al. | 1.38 mg/kg | 0.01 – 6.5 mg/kg | 1:2 water extract analysis |
| | Rughöft et al. 2016 | 11.3 mg/kg | 0.7 – 33.3 mg/kg | 2:5 water extract analysis |
| Soil Plant-Available P | Reed et al. | 2.20 mg/kg | 0.01 – 9.62 mg/kg | P Bray I |
| | Craine et al. 2008 | 38.52 mg/kg | 3.23 – 85.43 mg/kg | P Bray II |
| Leaf N:P Ratio | Reed et al. | 7.0 | 2.5 – 13.9 | |
| | Craine et al. 2008 | 5.8 | 3.2 – 9.2 | |

*For Reed et al., we used values from the 15m distance, and for Craine et al. 2008, we used values from the control plots, as in both cases these best represent the background levels of soil nutrients.

Aranibar, J. N., Macko, S. A., Anderson, I. C., Potgieter, A. L. F., Sowry, R. & Shugart, H. H. Nutrient cycling responses to fire frequency in the Kruger National Park (South Africa) as indicated by stable isotope analysis. *Isotopes Environ. Health Stud.,* 39, 141-158, https://doi.org/10.1080/1025601031000096736, 2003.

Rughöft, S., Hermann, M., Lazar, C. S., Cesarz, S., Levick, S. R., Trumbore, S. E. & Küsel, K. Community composition and abundance of bacterial, archaeal and nitrifying populations in savanna soils on contrasting bedrock material in Kruger National Park, South Africa. *Front. Microbiol.,* 7, https://doi.org/10.3389/fmicb.2016.01638, 2016.

Craine, J. M., Morrow, C., & Stock, W. D. Nutrient concentration ratios and co-limitation in South African grasslands. *New Phytol.,* 179, 829–836, https://doi.org/10.1111/j.1469-8137.2008.02513.x, 2008.

Esala, M. J. Changes in the extractable ammonium- and nitrate-nitrogen contents of soil samples during freezing and thawing. *Commun. Soil Sci. Plant Anal.*, 26, 61-68, https://doi.org/10.1080/00103629509369280, 1995.

Sollen-Norrlin, M. & Rintoul-Hynes, N. L. J. Soil sample storage conditions affect measurements of pH, potassium, and nitrogen. *SSSAJ*, 88, 930-941, https://doi.org/10.1002/saj2.20653, 2024.

Turner, B. L. & Romero, T. E. Short-term changes in extractable inorganic nutrients during storage of tropical rainforest soils. *SSSAJ*, 73, 1972-1979, https://doi.org/10.2136/sssaj2008.0407, 2009.

Line 38 – these are not graves. A grave is an excavation for burial.

AUTHORS' RESPONSE: We have changed the word gravesite to carcass site.

Line 77 – what about the amounts of cations in an elephant?

AUTHORS' RESPONSE: This is an interesting question. We estimated the N, P, and C quantities in an elephant based on the body size to macronutrient scaling rules described in Sterner & Elser (2002). Unfortunately, we are not aware of a well-established scaling rule for cations that would allow us to estimate cation concentrations in elephant tissue.

Sterner, R. W. & Elser, J. J. Ecological Stoichiometry: The Biology of Elements from Molecules to the Biosphere. Princeton University Press. 2002.

Line 89 – cations are not micronutrients. Is there direct evidence for widespread (or any) cation limitation of growth in savanna ecosystems?

AUTHORS' RESPONSE: Cation exchange is essential for soil fertility, and there is evidence of cation limitation (particularly $K^+$ and $Ca^{2+}$) on African savannas (Lathwell & Grove, 1986; Agbenin & Yakubu, 2006). We have edited the manuscript to refer to these elements as cations rather than micronutrients for clarity. We have also updated the introduction to include this information as follows (lines 91-95): "Aboveground, plant growth in African savannas is strongly limited by nutrient availability, most commonly N and P, but also by cations such as Ca, K, and magnesium (Mg) (Jobbágy & Jackson, 2004; Ries & Shugart, 2008; Pellegrini, 2016), and there is evidence of cation limitation of plants (particularly $K^+$ and $Ca^{2+}$) on African savannas (Lathwell & Grove, 1996; Agbenin & Yakubu, 2006)."

Agbenin, J. O. & Yakubu, S. Potassium-calcium and potatssium-magnesium exchange equilibria in an acid savanna soil from northern Nigeria. *Geoderma*, 136, 542-554, https://doi.org/10.1016/j.geoderma.2006.04.008, 2006.

Lathwell, D. J. & Grove, T. L. Soil-plant relationships in the tropics. *Annual Review of Ecology and Systematics*, 17, 1-16, https://www.jstor.org/stable/2096986, 1986.

Line 99 – please include classifications for the granitic and basaltic soils in one of the internationally recognized systems. In Soil Taxonomy these are presumably Inceptisols / Alfisols and Vertisols, respectively?

AUTHORS' RESPONSE: Thanks for the suggestion. The granitic soils are inceptisols, while the basaltic soils may be versitols or andisols (Khomo et al. 2017). We agree that these classifications are important for understanding the impacts of carcass-derived nutrients on different soil types and have updated the methods section as follows (lines 123-127): "The two dominant soil types in KNP are granitic soils (inceptisols) and basaltic soils (versitols or andisols) (Khomo et al. 2017). The clay-rich basaltic soils have relatively large surface area,

enabling them to retain larger quantities of water than granitic soils, which drain water more quickly and therefore are lower in water-soluble nutrients (Buitenweref, Kulmatiski, & Higgins, 2014; Rughöft et al. 2016)."

Khomo, L., Trumbore, S., Bern, C. R., & Chadwick, O. A. Timescales of carbon turnover in soils with mixed crystalline mineralogies. *SOIL*, 3, 17-30, https://doi.org/10.5194/soil-3-17-2017, 2017.

Buitenwerf, R., Kulmatiski, A. & Higgins, S. I. Soil water retention curves for the major soil types of the Kruger National Park. *Koedoe*, 56, a1228, http://dx.doi.org/10.4102/koedoe.v56i1.1228, 2014.

Line 138 – freezing soil has implications for subsequent measurements of extractable nutrients (e.g. Turner and Romero 2009). This should be mentioned here. Given the apparently very high values for some measurements (see below) I suspect that pretreatment had a major impact on results.

AUTHORS' RESPONSE: As detailed above, we acknowledge that freezing may have an impact on soil nitrogen measurements, though the literature is mixed on this topic. Generally, it seems that freezing more often impacts concentrations of ammonium but not nitrate (Esala, 1995; Turner & Romero, 2009; Sollen-Norrlin & Rintoul-Hynes, 2024). Indeed, in the Turner and Romero (2009) paper that the reviewer cites, freezing did not impact nitrate concentrations relative to fresh soils except in cases of high soil acidity, which were not present in our study. They write, "Frozen storage of the Pipeline Road and Fort Sherman soils preserved NO3 at similar concentrations to fresh samples (Table 2), but caused a marked decline for the acidic Albrook soil, indicating a possible effect of soil pH on NO3 stability. (Note: NO3 determined after 3 mo of storage in the Albrook soil)" (pg. 1974). The impacts of freezing on ammonium are also relatively minimal (<1 mg/kg per week of freezing according to Esala, 1995). Thus, based on what is known from the literature any impacts of freezing on the concentrations of nutrients in our samples were likely quite minimal. Further, we have compared the soil % nitrogen, nitrate, and ammonium values from the 10 and 15m distances in our study (those most similar to the background soils in Kruger) to those found in other soil analysis research in Kruger, and our values are consistent with that prior research (Aranibar et al. 2003; Rughöft et al. 2016).

| Metric | Source | Mean | Range | Method |
|---|---|---|---|---|
| Soil %N | Reed et al. | 11.4% | 5 – 16% | Stable isotope analysis |
| | Aranibar et al. 2003 | | ~5 – 23% | Stable isotope analysis |
| Soil Nitrate | Reed et al. | 57.1 mg/kg | 11.1 – 95.7 mg/kg | 1:2 water extract analysis |
| | Rughöft et al. 2016 | 28.9 mg/kg | 0.0 – 121.9 mg/kg | 2:5 water extract analysis |
| Soil Ammonium | Reed et al. | 1.38 mg/kg | 0.01 – 6.5 mg/kg | 1:2 water extract analysis |
| | Rughöft et al. 2016 | 11.3 mg/kg | 0.7 – 33.3 mg/kg | 2:5 water extract analysis |
| Soil Plant-Available P | Reed et al. | 2.20 mg/kg | 0.01 – 9.62 mg/kg | P Bray I |
| | Craine et al. 2008 | 38.52 mg/kg | 3.23 – 85.43 mg/kg | P Bray II |
| Leaf N:P Ratio | Reed et al. | 7.0 | 2.5 – 13.9 | |
| | Craine et al. 2008 | 5.8 | 3.2 – 9.2 | |

*For Reed et al., we used values from the 15m distance, and for Craine et al. 2008, we used values from the control plots, as in both cases these best represent the background levels of soil nutrients.

If freezing the samples did alter the measured soil ammonium concentrations, there is no reason to believe that the effect size would differ with distance from the carcass. Thus, even if absolute values are elevated by freezing, for which there is little evidence in the literature, the overall trends of decreasing ammonium and nitrate with carcass distance should still stand.

We have updated the methods section as follows (lines 158-161): "We chose to freeze samples rather than storing at room temperature based on literature demonstrating that the impacts of freezing on soil nitrate and ammonium concentrations are fairly minimal, except in specific cases of high soil acidity or peaty soils that were not present at our field site (Esala, 1995; Turner & Romero, 2009; Sollen-Norrlin & Rintoul-Hynes, 2024)."

Aranibar, J. N., Macko, S. A., Anderson, I. C., Potgieter, A. L. F., Sowry, R. & Shugart, H. H. Nutrient cycling responses to fire frequency in the Kruger National Park (South Africa) as indicated by stable isotope analysis. *Isotopes Environ. Health Stud.,* 39, 141-158, https://doi.org/10.1080/1025601031000096736, 2003.

Rughöft, S., Hermann, M., Lazar, C. S., Cesarz, S., Levick, S. R., Trumbore, S. E. & Küsel, K. Community composition and abundance of bacterial, archaeal and nitrifying populations in savanna soils on contrasting bedrock material in Kruger National Park, South Africa. *Front. Microbiol.,* 7, https://doi.org/10.3389/fmicb.2016.01638, 2016.

Craine, J. M., Morrow, C., & Stock, W. D. Nutrient concentration ratios and co-limitation in South African grasslands. *New Phytol.,* 179, 829–836, https://doi.org/10.1111/j.1469-8137.2008.02513.x, 2008.

Esala, M. J. Changes in the extractable ammonium- and nitrate-nitrogen contents of soil samples during freezing and thawing. *Commun. Soil Sci. Plant Anal.,* 26, 61-68, https://doi.org/10.1080/00103629509369280, 1995.

Sollen-Norrlin, M. & Rintoul-Hynes, N. L. J. Soil sample storage conditions affect measurements of pH, potassium, and nitrogen. *SSSAJ,* 88, 930-941, https://doi.org/10.1002/saj2.20653, 2024.

Turner, B. L. & Romero, T. E. Short-term changes in extractable inorganic nutrients during storage of tropical rainforest soils. *SSSAJ,* 73, 1972-1979, https://doi.org/10.2136/sssaj2008.0407, 2009.

Line 145/146 – please explain the difference between phosphate and plant-available P. As written, it appears they were both measured in the water extracts. Most plant-available P tests are not conducted in water (e.g. Olsen, Mehlich, Bray, etc).

AUTHORS' RESPONSE: We have updated the methods section as follows to include more details on soil nutrient analyses, including distinguishing between the methods used for phosphate and plant-available P (lines 166-179): "We sent 250 g of each soil sample to Eco-Analytica laboratory at the North-West University in Potchefstroom, South Africa for measurements of soil concentrations of ammonium $[NH_4]^+$, nitrate $[NO_3]^-$, phosphate $[PO_4]^{3-}$, and plant-available P. Samples were air-dried and sieved through < 2mm mesh prior to chemical analysis. Plant-available P was extracted from 4 g of soil and 30 ml extraction fluid (1:7.5 ratio) using an acid–fluoride solution (P Bray-1), measured colorimetrically using a Systea EasyChem200 analyser, and expressed as mg/kg. The detection limit was 0.5 mg/kg, and plant available P measurements <0.5 mg/kg were replaced with half the detection limit (0.25 mg/kg) (Croghan & Egeghy, 2003; Keenan & Beeler, 2023). Water-soluble nitrate and phosphate anions were extracted from volume on volume 100 ml soil and 200 ml deionized water, analyzed by ion chromatography on a Metrohm 930 Compact Flex System, and measured as mg/L. Ammonium (also 1:2 water extract) was analyzed colorimetrically using a Systea EasyChem200 analyzer and measured as mg/L. Detection limits for soil ions were 0.01 mg/L, and soil ion concentrations measured as <0.01 mg/L were replaced with half the detection limit (0.005 mg/L)."

Line 158 – was there any inorganic C in the samples? Savanna Vertisols developed in basalt can have considerable carbonate concentrations, albeit often in subsoil. Soil pH values would help indicate this possibility – how did carcasses affect soil pH?

AUTHORS' RESPONSE: We did not directly measure inorganic C, but we did measure soil pH and found no significant difference in soil pH with soil type or distance from the carcass center. We have added this result to the manuscript (Table S1, Figure S4B).

Line 161 – was moisture standardized prior to the incubations?

AUTHORS' RESPONSE: We did not standardize moisture prior to incubations, since the impact of carcasses on soil moisture was one of our questions (Figure S5 in original manuscript, Figure S4 below). However, we did account for soil moisture after the incubations following the methods process described in lines 227-230: "After soil respiration measurements, we determined sample dry weight by drying each sample at 60°C for 24-48 hours until stable mass was achieved. We subtracted dry weight from starting weight to obtain soil water content. Finally, we used the dry weights and the Ideal Gas Law to standardize all respiration measurements to $CO_2$ µg $h^{-1}g$ dry soil$^{-1}$." These methods are described in further detail in Lemoine et al. 2024, which is cited earlier in that paragraph.

Lemoine, N. P., Budny, M. L., Rose, E., Lucas, J., & Marshall, C. W. Seasonal soil moisture thresholds inhibit bacterial activity and decomposition during drought in a tallgrass prairie. Oikos, 2024, e10210, https://doi.org/10.1111/oik.10201, 2023.

Line 182 – an alternative is to set values to ½ detection limit.

AUTHORS' RESPONSE: We have re-run the analyses using 0.005 mg/L as the replacement value for any zeros in the soil ion concentration data, as that is half of the detection limit (0.10 mg/L). We updated the results accordingly, but this update did not result in any changes to statistical significance or model performance in the results. We have updated the methods as follows (lines 169-179): "Plant-available P was extracted from 4 g of soil and 30 ml extraction fluid (1:7.5 ratio) using an acid–fluoride solution (P Bray-1), measured colorimetrically using a Systea EasyChem200 analyser, and expressed as mg/kg. The detection limit was 0.5 mg/kg, and plant available P measurements <0.5 mg/kg were replaced with half the detection limit (0.25 mg/kg) (Croghan & Egeghy, 2003; Keenan & Beeler, 2023). Water-soluble nitrate and phosphate anions were extracted from volume on volume 100 ml soil and 200 ml deionized water, analyzed by ion chromatography on a Metrohm 930 Compact Flex System, and measured as mg/L. Ammonium (also 1:2 water extract) was analyzed colorimetrically using a Systea EasyChem200 analyzer and measured as mg/L. Detection limits for soil ions were 0.01 mg/L, and soil ion concentrations measured as <0.01 mg/L were replaced with half the detection limit (0.005 mg/L)."

Line 188 – I understand that there were insufficient carcasses to allow inclusion of carcass age in models. However, major differences would be expected between carcasses aged 1 month vs 2.5 years. Is there any way to provide an indication of the magnitude of the age effect? How would distance effects look if young carcasses were excluded, for example?

AUTHORS' RESPONSE: Thanks for the suggestion! We have added analysis and figures to the manuscript testing for a relationship between key soil metrics and carcass age. We found that soil ammonium, phosphate, and respiration potential all decrease significantly with carcass age. In fact, the trends are so compelling that we have added this figure to the main text (Figure 5). This figure suggests the pattern of elevated soil nutrients that we found may be even stronger when considering younger carcasses given how quickly the nutrients decline with age.

We have added this update to the methods section as follows (lines 280-282): "Finally, to test the impact of carcass age on key soil metrics, we ran exponential decay functions for soil ammonium, nitrate, phosphate, and respiration verses carcass age for samples from the center of the carcass site (0.5m sampling location)."

And to the results section as follows (lines 343-345): "Soil ammonium, phosphate, and respiration potential all decreased significantly with carcass age (Figure 5A-C). The exponential decay model for nitrate failed to converge due to an outlier with extremely high soil nitrate (1454 mg/kg) at 258 days post-death (Figure 5D)."

Line 228 – It is perhaps not surprising that P concentrations showed little variation with distance, given that P was measured in water extracts (i.e. the extraction is recovering a relatively small pool of soluble P).

AUTHORS' RESPONSE: This is an interesting point. The results included in this manuscript are soil phosphate (1:2 water extract) and plant-available P (P Bray-I), but we also measured mineral phosphorus (P31) in mg/kg using the same method we did for soil Na, Mg, Fe, Ca, and K— microwave-assisted digestion in a solution of 9 mL 65% nitric acid ($HNO_3$) and 3 mL 32% hydrochloric acid (HCl) (lines 182-197). We have added this result to the manuscript (Figure 2G; Table S1), thereby increasing the proportion of total soil P covered by our analyses. All three individual measurements of soil P (phosphate, plant-available P, and mineral P) indicate an interaction between distance and soil type in which P decreases with distance from the carcass center, but only in granitic soils.

Line 230 – how is plant-available P defined here?

AUTHORS' RESPONSE: Plant-available P refers to P measured via the P Bray I method. We have updated the methods as follows (lines 169-174): "Plant-available P was extracted from 4 g of soil and 30 ml extraction fluid (1:7.5 ratio) using an acid–fluoride solution (P Bray-1), measured colorimetrically using a Systea EasyChem200 analyser, and expressed as mg/kg. The detection limit was 0.5 mg/kg, and plant available P measurements <0.5 mg/kg were replaced with half the detection limit (0.25 mg/kg) (Croghan & Egeghy, 2003; Keenan & Beeler, 2023)."

Line 286 – Soil extractable nutrients should be expressed on the basis of dry soil, not volume.

AUTHORS' RESPONSE: We have updated the manuscript so that here and throughout, soil ion concentrations are given in mg/kg soil rather than mg/L.

Line 286 - these very high extractable nitrogen concentrations are presumably in part a consequence of soils being frozen prior to analysis. Another factor is time between sampling and freezing - or storage prior to freezing. Please provide a statement about sample treatment prior to analysis (time from sampling to freezing, storage conditions during this time, etc, as relevant).

AUTHORS' RESPONSE: We have updated the methods section to include more details on soil storage as follows (lines 154-158): "Soil samples were stored in a cooler during fieldwork. On the day they were collected, we used 5 g of each soil sample for soil respiration measurements (described below). The rest of each sample was stored in plastic bags in a -20°C freezer until nutrient analyses; they were stored in coolers with ice blocks during the transition from the freezer at the field site to the freezers at the labs."

As we discuss above, there were likely negligible, if any, impacts of freezing on nitrate concentrations given the data available in the literature. Similarly, the impacts on ammonium were likely also minor. Further, we have also compared our measured nitrogen concentrations to those from other studies performed in Kruger (% N, Aranibar et al. 2003; ammonium and nitrate,

Rughöft et al. 2016; see table below). We used the 15m distance in our dataset for comparison, since that is most representative of the baseline nitrogen concentrations. Our values were similar to those found by other researchers in Kruger, which increases our confidence that the measurements we took are accurate. The nitrogen concentrations that we found at the 0-5m distances are indeed much higher than those found elsewhere, but we attribute this to the presence of the elephant carcass.

| Metric | Source | Mean | Range | Method |
|---|---|---|---|---|
| Soil %N | Reed et al. | 11.4% | 5 – 16% | Stable isotope analysis |
| | Aranibar et al. 2003 | | ~5 – 23% | Stable isotope analysis |
| Soil Nitrate | Reed et al. | 57.1 mg/kg | 11.1 – 95.7 mg/kg | 1:2 water extract analysis |
| | Rughöft et al. 2016 | 28.9 mg/kg | 0.0 – 121.9 mg/kg | 2:5 water extract analysis |
| Soil Ammonium | Reed et al. | 1.38 mg/kg | 0.01 – 6.5 mg/kg | 1:2 water extract analysis |
| | Rughöft et al. 2016 | 11.3 mg/kg | 0.7 – 33.3 mg/kg | 2:5 water extract analysis |
| Soil Plant-Available P | Reed et al. | 2.20 mg/kg | 0.01 – 9.62 mg/kg | P Bray I |
| | Craine et al. 2008 | 38.52 mg/kg | 3.23 – 85.43 mg/kg | P Bray II |
| Leaf N:P Ratio | Reed et al. | 7.0 | 2.5 – 13.9 | |
| | Craine et al. 2008 | 5.8 | 3.2 – 9.2 | |

*For Reed et al., we used values from the 15m distance, and for Craine et al. 2008, we used values from the control plots, as in both cases these best represent the background levels of soil nutrients.

Aranibar, J. N., Macko, S. A., Anderson, I. C., Potgieter, A. L. F., Sowry, R. & Shugart, H. H. Nutrient cycling responses to fire frequency in the Kruger National Park (South Africa) as indicated by stable isotope analysis. *Isotopes Environ. Health Stud.,* 39, 141-158, https://doi.org/10.1080/1025601031000096736, 2003.

Rughöft, S., Hermann, M., Lazar, C. S., Cesarz, S., Levick, S. R., Trumbore, S. E. & Küsel, K. Community composition and abundance of bacterial, archaeal and nitrifying populations in savanna soils on contrasting bedrock material in Kruger National Park, South Africa. *Front. Microbiol.,* 7, https://doi.org/10.3389/fmicb.2016.01638, 2016.

Craine, J. M., Morrow, C., & Stock, W. D. Nutrient concentration ratios and co-limitation in South African grasslands. *New Phytol.,* 179, 829–836, https://doi.org/10.1111/j.1469-8137.2008.02513.x, 2008.

Line 308 – the high available P and tissue N:P ratios (see below) indicate that there is no P limitation here. This might limit the likely influence of carcasses on foliar P, as found here (i.e. foliar P is not a strong indicator of the extent to which carcasses change P availability in general).

AUTHORS' RESPONSE: Thanks for bringing this up. We appreciate your comments here and below suggesting that P limitation might not be strong at our sites, and we have updated this paragraph in the discussion as follows (lines 394-409):

"Elevated soil phosphate (Figure 2E) and plant-available P (Figure 2F) at the center of carcass sites were also consistent with expectations from the literature (Bump et al. 2009a; Parmenter & MacMahon, 2009). However, elevated P levels in soil did not translate to elevated P in grass leaves (Figure 4C), which could suggest a lag between trends in soil and plants that is longer for P than for N. This lag could occur because phosphate easily forms chemical bonds with other soil ions (e.g., iron and aluminum in acidic soils and calcium in basic soils). Nitrate does not form these bonds and therefore has greater water solubility and mobility in soils and may be more readily taken up by plants (Wiersum, 1962; Arai & Sparks, 2007). However, it is also possible that P limitation in Kruger is not as strong as it is in some other African savanna systems (Pellegrini, 2016). The foliar N:P ratios measured in this experiment were higher closer to the center of the carcass site (median 9.38 at 0 m and 4.83 at 15 m), indicating that N limitation may be relatively stronger further from the carcass site, and P limitation may be relatively stronger closer to the center (Figure 4D, Table S2). These relatively high foliar N:P ratios at the center of carcass sites are similar to those found in N fertilization studies in Kruger (Craine et al. 2008), further supporting the idea that the influx of N from megacarcasses may shift the soil from relatively more N limited to more P limited."

We also compared our plant-available P and foliar N:P ratios with other studies in Kruger and found that our plant-available P results were actually lower, while our foliar N:P ratios were in about the same range.

| Metric | Source | Mean | Range | Method |
|---|---|---|---|---|
| Soil Plant-Available P | Reed et al. | 2.20 mg/kg | 0.01 – 9.62 mg/kg | P Bray I |
| | Craine et al. 2008 | 38.52 mg/kg | 3.23 – 85.43 mg/kg | P Bray II |
| Leaf N:P Ratio | Reed et al. | 7.0 | 2.5 – 13.9 | |
| | Craine et al. 2008 | 5.8 | 3.2 – 9.2 | |

Craine, J. M., Morrow, C., & Stock, W. D. Nutrient concentration ratios and co-limitation in South African grasslands. *New Phytol.,* 179, 829–836, https://doi.org/10.1111/j.1469-8137.2008.02513.x, 2008.

Line 322 – water-extractable P represents a tiny proportion of the total soil P, so reliance on this procedure probably limits the possibility of detecting change in soil P.

AUTHORS' RESPONSE: We have updated the results to include mineralized P in soils in addition to phosphate and plant-available P, as described above in response to line 228.

Line 323- 330 – this text largely repeats results.

AUTHORS' RESPONSE: Thank you for this feedback. We have updated this section to better integrate statements of results with their interpretations, with the new text reading (lines 423-431): "The contributions of megacarcasses to soil macronutrient and cation pools were strongly associated with soil type. Our results confirmed the previously-established trend that basaltic

soils are overall more cation rich than granitic soils, with greater concentrations of P, K, Fe, Mg, and Ca (Figure 2G; Figure S3B-E; Gertenbach, 1983; Craine, Morrow, & Stock, 2008; Wigley et al. 2014). However, soil ammonium, $\delta^{15}N$, and phosphate were all higher in the granitic soils towards the center of carcass sites, decreasing steeply to be similar to basaltic soils about 10 m from the carcass center (Figure 2C-E). These results indicate that the impact of organic matter from megacarcasses may be stronger in relatively nutrient-poor and sandy granitic soil compared with nutrient-rich and clayey basaltic soil."

Line 340 - missing here is a discussion of the ecological consequences of the findings. What are the implications for plant and microbial ecology in savanna ecosystems?

AUTHORS' RESPONSE: Thanks for this feedback. We have updated the discussion section to include a paragraph on the importance of these nutrient hotspots for savanna ecosystem functioning (lines 441-451): "The magnitude of nutrient inputs from megacarcasses, as well as the substantial size and duration of their impact zones, means their impacts on ecosystem processes may be functionally distinct from smaller carrion. Indeed, there is evidence that carcass size strongly impacts scavenger food web structure (Moleón et al. 2015; Morris et al. 2023). Moreover, the attraction of animals to carcasses via scavenging, predation, or mourning (Goldenberg & Wittemyer, 2020) could have positive feedbacks on nutrient cycling (Bump, Peterson, & Vucetich, 2009; Monk et al. 2024), which may be magnified by carcass size. Thus, the impacts of megacarcasses on savanna ecosystem processes may be dissimilar to the effects of small carrion and more similar to other more persistent contributors to savanna ecosystem processes, such as termite mounds (Davies et al. 2016), cattle bomas (Augustine, 2003), and even mass animal mortality events (Subalusky et al. 2017, 2020)."

Figure 2B, C – these values should be presented in mg/kg soil.

AUTHORS' RESPONSE: We have updated the manuscript so that here and throughout, soil ion concentrations are given in mg/kg soil rather than mg/L. To do this conversion, we multiplied the volume in mg/L by 2 based on the 1:2 soil to water extraction ratio, as the reviewer helpfully described below.

Figure 2B – these are extremely high nitrate concentrations, even out to 15 m. For example, 100 mg/L is equivalent to 200 mg/kg based on a 1:2 soil to water extraction ratio. Extractions done quickly after sampling and in 2 M KCl are in the range of 1-5 mg/kg. This seems to be a clear indication of storage effects.

AUTHORS' RESPONSE: We have found our N measurements to be consistent with other studies in Kruger. As we discuss in several places above, there is unlikely to be any significant storage effects on these measurements.

| Metric | Source | Mean | Range | Method |
|--------|--------|------|-------|--------|
| Soil %N | Reed et al. | 11.4% | 5 – 16% | Stable isotope analysis |
| | Aranibar et al. 2003 | | ~5 – 23% | Stable isotope analysis |
| Soil Nitrate | Reed et al. | 57.1 mg/kg | 11.1 – 95.7 mg/kg | 1:2 water extract analysis |
| | Rughöft et al. 2016 | 28.9 mg/kg | 0.0 – 121.9 mg/kg | 2:5 water extract analysis |
| Soil Ammonium | Reed et al. | 1.38 mg/kg | 0.01 – 6.5 mg/kg | 1:2 water extract analysis |
| | Rughöft et al. 2016 | 11.3 mg/kg | 0.7 – 33.3 mg/kg | 2:5 water extract analysis |
| Soil Plant-Available P | Reed et al. | 2.20 mg/kg | 0.01 – 9.62 mg/kg | P Bray I |
| | Craine et al. 2008 | 38.52 mg/kg | 3.23 – 85.43 mg/kg | P Bray II |
| Leaf N:P Ratio | Reed et al. | 7.0 | 2.5 – 13.9 | |
| | Craine et al. 2008 | 5.8 | 3.2 – 9.2 | |

*For Reed et al., we used values from the 15m distance, and for Craine et al. 2008, we used values from the control plots, as in both cases these best represent the background levels of soil nutrients.

Aranibar, J. N., Macko, S. A., Anderson, I. C., Potgieter, A. L. F., Sowry, R. & Shugart, H. H. Nutrient cycling responses to fire frequency in the Kruger National Park (South Africa) as indicated by stable isotope analysis. *Isotopes Environ. Health Stud.,* 39, 141-158, https://doi.org/10.1080/1025601031000096736, 2003.

Rughöft, S., Hermann, M., Lazar, C. S., Cesarz, S., Levick, S. R., Trumbore, S. E. & Küsel, K. Community composition and abundance of bacterial, archaeal and nitrifying populations in savanna soils on contrasting bedrock material in Kruger National Park, South Africa. *Front. Microbiol.,* 7, https://doi.org/10.3389/fmicb.2016.01638, 2016.

Craine, J. M., Morrow, C., & Stock, W. D. Nutrient concentration ratios and co-limitation in South African grasslands. *New Phytol.,* 179, 829–836, https://doi.org/10.1111/j.1469-8137.2008.02513.x, 2008.

Figure 2B, C – are these values as NO3/NH4 or on an N basis?

AUTHORS' RESPONSE: These values are on an N-basis (as are the values from Rughöft et al. 2016 in the tables above), and we have updated the figure legend to clarify.

Figure 2D – please express stable isotope ratios as $\delta^{15}N$. This may be how the results are presented, but this is not clear from the units.

AUTHORS' RESPONSE: We have changed the notation here and throughout the manuscript to $\delta^{15}N$.

Figure 2F – these are very high available P concentrations for a natural ecosystem, although there is no mention of the method used.

AUTHORS' RESPONSE: We have updated the methods section to include the method used to measure plant-available P as follows (lines 169-174): "Plant-available P was extracted from 4 g of soil and 30 ml extraction fluid (1:7.5 ratio) using an acid–fluoride solution (P Bray-1), measured colorimetrically using a Systea EasyChem200 analyser, and expressed as mg/kg. The detection limit was 0.5 mg/kg, and plant available P measurements <0.5 mg/kg were replaced with half the detection limit (0.25 mg/kg) (Croghan & Egeghy, 2003; Keenan & Beeler, 2023)."

In comparison with the literature, we found that our soil plant-available P values were actually lower than those from other studies performed in Kruger on the same granitic and basaltic soil types.

| Metric | Source | Mean | Range | Method |
|---|---|---|---|---|
| Soil Plant-Available P | Reed et al. | 2.20 mg/kg | 0.01 – 9.62 mg/kg | P Bray I |
| | Craine et al. 2008 | 38.52 mg/kg | 3.23 – 85.43 mg/kg | P Bray II |

*For Reed et al., we used values from the 15m distance, and for Craine et al. 2008, we used values from the control plots, as in both cases these best represent the background levels of soil nutrients.

Craine, J. M., Morrow, C., & Stock, W. D. Nutrient concentration ratios and co-limitation in South African grasslands. *New Phytol.,* 179, 829–836, https://doi.org/10.1111/j.1469-8137.2008.02513.x, 2008.

Figure 4 – it looks like foliar N:P ratios are around 4 (2% N, 0.5%P) – these very low values that suggest strong N limitation. This is incompatible with the very high nitrate values presented in Figure 2. This further indicates storage problems with N measurements.

AUTHORS' RESPONSE: The reviewer brings up an interesting point about the stoichiometry of the grass we sampled around the elephant carcasses. We now analyze foliar N:P ratios and found that they were overall higher in granitic soils compared to basaltic soils and decreased with distance from the carcass center in both soil types. The median foliar N:P ratio was 9.38 at the 0m distance and 4.83 at the 15m distance, which may indicate that N limitation may be relatively stronger further from the carcass site, and P limitation may be relatively stronger closer to the center. We have added these new results to the manuscript (Figure 4D, Table S2).

Interestingly, some previous work in Kruger with this same species of grass, as well as other grasses, showed that N:P ratios in grasses responded similarly to the addition of N fertilizer (Crane et al. 2008). Under control nutrient conditions the grasses in their study had a N:P of 5.8 on average. But similar to our study, under N fertilization grasses had a N:P of 9.9 on average. These data argue against the reviewer's point that having high N availability in the soil necessarily increases the N:P to high levels that one would expect to indicate P limitation. This example, and the data we present above, suggest that the storage methods for the soils that we used are not impacting our dataset.

We have updated the discussion as follows (lines 406-409): "These relatively high foliar N:P ratios at the center of carcass sites are similar to those found in N fertilization studies in Kruger (Craine et al. 2008), further supporting the idea that the influx of N from megacarcasses may shift the soil from relatively more N limited to more P limited."

Craine, J. M., Morrow, C., & Stock, W. D. Nutrient concentration ratios and co-limitation in South African grasslands. *New Phytol.*, 179, 829–836, https://doi.org/10.1111/j.1469-8137.2008.02513.x, 2008.

Augustine, D. J. (2003). Long-term, livestock-mediated redistribution of nitrogen and phosphorus in an East African savanna. Journal of Applied Ecology, 40(1), 137-149.

Hudson et al. (2009). Temporal patterns of nutrient availability around nests of leaf-cutting ants (Atta*olombica*) in secondary moist tropical forest. Soil Biology and Biochemistry, 41(6), 1088-1093.

Turner, B. L., & Romero, T. E. (2009). Short-term changes in extractable inorganic nutrients during storage of tropical rain forest soils. Soil Science Society of America Journal, 73, 1972-1979.

AUTHORS' RESPONSE: Thank you for providing these references. We have found the Turner and Romero (2009) manuscript particularly instructive in interpretating the soil nitrogen data. We have updated the discussion to include comparisons with other types of savanna nutrient hotspots (lines 441-451): "The magnitude of nutrient inputs from megacarcasses, as well as the substantial size and duration of their impact zones, means their impacts on ecosystem processes may be functionally distinct from smaller carrion. Indeed, there is evidence that carcass size strongly impacts scavenger food web structure (Moleón et al. 2015; Morris et al. 2023). Moreover, the attraction of animals to carcasses via scavenging, predation, or mourning (Goldenberg & Wittemyer, 2020) could have positive feedbacks on nutrient cycling (Bump, Peterson, & Vucetich, 2009; Monk et al. 2024), which may be magnified by carcass size. Thus, the impacts of megacarcasses on savanna ecosystem processes may be dissimilar to the effects of small carrion and more similar to other more persistent contributors to savanna ecosystem processes, such as termite mounds (Davies et al. 2016), cattle bomas (Augustine, 2003), and even mass animal mortality events (Subalusky et al. 2017, 2020)."

---

## Author Response (AR2)

**Response to Feedback from Associate Editor**

Your manuscript has been reviewed by the three original reviewers. While they all reckon that you dealt with most of the issues that had previously been raised, there remain still some concerns that you need to address before the manuscript can be accepted for publication.

AUTHORS' RESPONSE: Thank you for considering a revised resubmission. We appreciate your constructive feedback and the opportunity to address reviewer comments.

Specifically, you need to provide (1) details on sampling storage and discuss the implication of storage time between sampling and freezing on available nitrogen concentrations;

AUTHORS' RESPONSE: We have updated the manuscript to provide additional details on sample storage time and conditions (lines 153-162). We do not have specific times that each sample was frozen, and we would be surprised if anyone had such detailed freeze-thaw dates on most of their samples as some studies do not even report how their soil samples were handled. It would be highly unlikely that we had frozen and thawed the samples in a way that created a bias in the data that would create the clear patterns with distance that we show (as the one reviewer notes). We have provided comparisons with sample storage in other studies in our response to Reviewer 3.

(2) compelling explanation on the contrast between strong nitrogen limitation based on foliar data and high nitrogen availability based on KCl extractions;

AUTHORS' RESPONSE: We have provided a thorough response to Reviewer 2's concern about the high soil nitrate values relative to foliar N, including demonstrating papers showing similar results using KCl extractions. One such study (Seymour et al. 2014) used KCl extractions to demonstrate that termite mounds, a well-studied hotspot of nutrient heterogeneity on southern African savanna landscapes, have soil nitrate concentrations of up to 974 mg/kg, a concentration that is comparable with our study. That same study found only weakly elevated foliar N on termite mounds relative to control sites. The uptake of nutrients by plants is controlled by many different mechanisms, which are beyond the scope of our study. While one could expect higher levels of N in the grasses given the concentration of nutrients in the soil, the response we show is not uncommon. But given that our paper is not about the physiological mechanisms with which plants take up nutrients, we do not discuss this discrepancy in detail. We have updated the manuscript discussion to include comparison between our results and this study (lines 453-458).

(3) evidence for cation limitation of plant growth in natural communities on African savannas;

AUTHORS' RESPONSE: We have rephrased that portion of the introduction (lines 91-94) to focus on the importance of micronutrients more generally (both cations and anions), as this is what we actually measured in the study. We also updated the citations to ones that experimentally demonstrated increased plant productivity with the addition of micronutrients (sodium and potassium). However, we limited this section to just one sentence in the introduction, as we do not actually measure or report plant productivity in the study, and we found that attempts to expand this section were distracting from the main focus of the manuscript. Micronutrient limitation clearly exists in plants (Epron et al. 2012; Chen et al. 2024), and we show that elephant carcasses are significant sources of micronutrients. We simply draw this parallel to suggest the potential importance of these carcasses. Any further discussion would be conjecture on our part, and we chose not to do that.

Epron, D., Laclau, J-P., Almeida, J. C. R., Gonçalves, J. L. M., Ponton, S., Sette, Jr., C. R., Delgado-Rojas, J. S., Bouillet, J-P. & Nouvellon, Y. Do changes in carbon allocation account for the growth response to potassium and sodium applications in tropical Eucalyptus plantations? *Tree Physiology*, 32, 667-679, https://doi.org/10.1093/treephys/tpr107, 2012.

Chen, B., Fang, J., Piao, S., Ciais, P., Black, T. A., Wang, F., Niu, S., Zeng, Z. & Luo, Y. A meta-analysis highlights globally widespread potassium limitation in terrestrial ecosystems. *New Phytologist,* 241, 154-165, https://doi.org/10.1111/nph.19294, 2024.

(4) detailed description on how geochemical data were obtained and convert all soil chemistry values to mg/kg (soil dry weight) to guarantee comparability of your results with other studies.

AUTHORS' RESPONSE: As noted above, we updated the methods to include further methodological details, including sample storage information. We also provided a detailed explanation for Reviewer 2 on soil conversions, providing the stoichiometry for our calculations and a Sigma-Aldrich protocol confirming them. All of our soil chemistry values are in mg/kg.

Finally, revise your Results and Discussion based on the data obtained after conversion soil chemistry values.

AUTHORS' RESPONSE: We have updated our discussion based on reviewer feedback, particularly the paragraphs discussing the impacts of elephant carcasses on soil ammonium, nitrate, and phosphate concentrations. We also expanded the discussion to include better comparisons with the literature on smaller carrion (lines 446-467), as recommended by Reviewer 3. We believe that these changes greatly strengthen the quality of the discussion, and we appreciate the feedback.

**Response to Feedback from Reviewer 1**

The authors have completed a comprehensive revision which includes new and revised analyses, greater details in the methods, and improvements to the narrative. The ms is now greatly improved - well done!

AUTHORS' RESPONSE: Thank you!

**Response to Feedback from Reviewer 2**

General comments/major concerns:

The authors addressed some of the concerns raised in the initial review, but there are still areas that need some significant clarification and correction related to the conversion of concentrations measured in soil chemistry with the conversion to mg/kg (dry weight?). The authors describe (line 175) that water soluble nitrate and phosphate were extracted using 100 ml soil and 200 ml deionized water. I am unfamiliar with measuring soil in ml rather than mass, and I am concerned that this is not a standardized approach and does not provide a meaningful measure of those ions. This extends to the ammonium extraction—if this was also conducted as a 1:2 ratio (100 ml soil to 200 ml water), that is also a non-standardized approach. Is this a typo and this should be 100 mg of soil? This confusion extends further to the calculations for converting mg/L, which as described, that math would end up with the units of mL/mL of soil, which is not a unit I've seen before with soil chemistry. The concentrations should be based on a gram dry weight basis, not volume of soil. There should be some correction for dry mass of the soil.

AUTHORS' RESPONSE: This is a typo; thanks for catching it! We have fixed this line to read "100 mg soil and 200 mL deionized water". The units for soil ions (ammonium, nitrate, and phosphate) were mg/L, which we then converted to mg/kg by multiplying by 2 based on the 1:2 soil:water extraction ratio. The stoichiometry behind this conversion is as follows:

mg/L (1 L / 1000 ml)  (200 ml / 100 mg) (1000 mg / 1 kg) = mg/L x 2

Multiplying by 2 is also the conversion recommended by Sigma-Aldrich:
https://www.sigmaaldrich.com/US/en/technical-documents/protocol/environmental-testing-and-industrial-hygiene/soil-solid-waste-and-groundwater-testing/ammonium-in-soils#calculation

Further, despite describing this conversion process to present ammonium concentrations in soil in a standardized way (mg/some gram dry weight basis), the authors later (lines 370-372) describe ammonium in soil as mg/L. The figures of ammonium show data as mg/kg. The values presented (assuming there was a typo and the data are actually mg/kg gram dry weight basis) are also significantly smaller than other studies with smaller carcasses (e.g., Quagiotto et al., 2019),

which is the opposite of the discussed results and conclusions stated. Here, the maximum concentration of ammonium reported (based on your figure 2C) is ~150 mg/kg. The max concentration of ammonium in Quagiotto et al. (2019) is ~800 mg/kg. The opposite is true of nitrate, which your results suggest is 500-1500 mg/kg whereas in Quagiotto et al. it is ~2 mg/kg. This is exceptionally high nitrate, even compared to some of the prior data generated from Kruger and included in the author's responses to reviewer #3. In sum, it is really difficult to figure out how the data were generated and then handled to get to the results presented for the soil geochemistry.

There is something strange going on with the soil nitrogen data (excluding the %N and nitrogen isotopes) that does not make sense from the perspective of what should be expected (high ammonium, low nitrate), what is typical of ecosystems (mentioned by reviewer #3 in the first review), and compared to other studies at Kruger.

AUTHORS' RESPONSE: In lines 368-372, we gave ammonium concentrations in mg/L to allow easy comparison with Britto & Kronzucker (2002) (cited in line 369), showing how soil ammonium levels compare with those typically considered toxic to plants. We understand that this was confusing and have updated the paragraph as follows to present all units in mg/kg:

"The mean ammonium level at the center of carcass sites (34.8 mg/kg) was higher than the level generally considered toxic to plants (Britto & Kronzucker, 2002). Yet, we found living grass— typically *U. trichopus*—in the center of the carcass site at seven out of ten of our sites (ammonium range 10-172 mg/kg) and at the 2.5m distance for all sites (ammonium range 0-72 mg/kg). The three sites without vegetation in the center had the highest ammonium levels (70-144 mg/kg), suggesting that *U. trichopus* has a higher degree of ammonium tolerance than some sympatric grass species but may still be limited by the high ammonium levels at the centers of these three relatively fresh carcass sites."

We believe that the difference between our results and the results in Quaggiato et al. (2019) is a matter of time scale. The Quaggiato paper measured soil ammonium and nitrate under rabbit carcasses for 20 days postmortem, finding that soil ammonium at carcass sites was 20x higher than controls at day 4, while nitrate did not significantly differ between carcass and control sites across the 20-day period. However, studies that continue measurements for longer time periods consistently exhibit the same trend that we did – soil ammonium concentrations peak earlier in the decay period than nitrate, with a transition from high soil ammonium to high soil nitrate over time (Yong et al. 2019; Keenan et al. 2018). Moreover, elephant body size may be coming into play with these results. Parmenter and MacMahon (2009) compared the impacts of carrion body size on soil nutrient concentrations and found that soil nitrogen peaks occurred earlier for smaller carrion (e.g. deer mice, chipmunks) relative to larger carrion (e.g., dog, mule deer) (Parmenter &

MacMahon, 2009, Table 5). Thus, we would expect that peak soil nitrogen concentrations at elephant carcass sites would take longer than the rabbits in Quaggiato et al. (2019).

As shown in previous responses to reviewers, the nitrate concentrations at our control 15m distances are consistent with those found in the Kruger literature. The contribution that our research makes to the literature is demonstrating the impacts of elephant carcasses on soil nitrate, something that has previously not been tested and therefore cannot be directly compared with other studies. However, we are not the first researchers to record these high nitrate concentrations in southern African savanna soils. Soil nitrate concentrations of up to 974 mg/kg have been recorded on termite mounds (using the KCl extraction method), another major nutrient hotspot on savanna landscapes (Seymour et al. 2014).

Finally, with regards to ammonium, our freshest elephant carcass was 24 days postmortem, and it had the highest soil ammonium concentration. However, the mean site age was older – 350 days postmortem (range 24-811 days). Because of this age distribution, our data may be capturing the high soil nitrate concentrations at later stages of decomposition after nitrification has occurred but may not be capturing the full impact of ammonium in those early stages. Indeed, this trend of conversion from ammonium to nitrate is visible in Figure 5, where we see that soil ammonium (Fig. 5A) and respiration potential (Fig. 5C) both spike early then decrease rapidly, while soil nitrate (Fig. 5D) does not follow that pattern.

Seymour, C. L., Milewski, A. V., Mills, A. J., Joseph, G. S., Cumming, G. S., Cumming, D. H. M. & Mahlangu, Z. Do the large termite mounds of *Macrotermes* concentrate micronutrients in addition to macronutrients in nutrient-poor African savannas? *Soil Biology and Biogeochemistry*, 68, 95-105, https://doi.org/10.1016/j.soilbio.2013.09.022, 2014.

Additionally, the phosphate seems really low, which is also maybe a conversion issue?

AUTHORS' RESPONSE: Our maximum soil phosphate concentration was 58 mg/kg, which is indeed lower than Quaggiatto et al. (2019), which recorded a max of ~275 mg/kg soil phosphate at rabbit carcass sites. However, our results are consistent with Parmenter & MacMahon (2009), which found that mule deer carcasses increase soil P by only 13g after 2 years and 41g after 5 years postmortem. Our phosphorus results are also greater than Mlambo et al. 2007, which found that intercanopy P in southern African savanna soils was only 8-16 mg/kg. These differing results across studies highlight that the impacts of carcasses on soil nutrient dynamics are complex and differ with factors such as species, scavenger densities, soil type, precipitation, etc. Indeed, one reason why phosphate levels at carcass sites are lower than expected may be due to bone dispersal by scavengers, which we address in lines 414-426:

"The elevated plant-available P at the center of carcass sites likely came primarily from phosphate released from decomposing tissue (Yong et al. 2019). Bone decomposition, which is

also likely a major source of P from animal carcasses (Subalusky et al. 2020), occurs over long time scales (Coe, 1978; Subalusky et al. 2020) and therefore should result in the slow release of P and a gradual decrease in the N:P ratio (Parmenter & MacMahon, 2009; Quaggiotto et al. 2019). Indeed, initial inorganic N influxes to the Mara River in Kenya from mass wildebeest die-offs are 10-fold greater than concurrent increases in P, which instead releases slowly over about seven years of bone decomposition (Subalusky et al. 2017). Research following megacarcasses over longer timeframes postmortem is needed to clarify when P from enriched soil is absorbed by plants and at what stage megacarcass bones begin contributing to soil P dynamics. It is also possible that bone dispersal by scavengers may result in less P leaching from bones close to where the elephant died and more P being distributed across the landscape at distances far from the carcass site."

The concerns related to the soil geochemical data are significant and have impacts on the discussion and conclusions presented. Until these data issues are resolved, aspects of the discussion are not supported by the current results.

AUTHORS' RESPONSE: We have updated and expanded the discussion in response to specific comments below, particularly the sections on soil nitrate and ammonium.

Additionally, one suggested change is to replace the words "post-death" with either "post-mortem" or "postmortem." While "post-death" is not technically incorrect, the convention and terminology used more broadly is "postmortem."

AUTHORS' RESPONSE: We have changed all instances of the word "post-death" to "postmortem".

Specific comments:
- Lines 172-174: It may be better to cite an EPA or USGS protocol rather than these papers.

AUTHORS' RESPONSE: We have added citations for specific protocols for P Bray-1 and the 1:2 soil:water extraction analyses. We have kept the Croghan & Egeghy (2003) citation because it refers to the use of half the detection value for measurements under the detection limit.

- Line 280: I think it's great to evaluate change in soil chemistry as a function of carcass time since death, but could the authors explain why an exponential decay would be expected?

AUTHORS' RESPONSE: Thanks! We chose to use the exponential decay function because it models a non-constant rate of decrease that asymptotes without reaching zero, which is what we could expect for soil nutrients.

- Line 182: The authors state, "To determine whether soil anions were distinct…" but then list cations. At the end of this same sentence, you don't measure cations or anions "using microwave assisted digestion."

AUTHORS' RESPONSE: We have edited this sentence as follows for clarity (lines 183-185): "To determine whether soil micronutrients were distinct and elevated at the center of carcass sites relative to soil further from the center, we measured concentrations of sodium (Na), magnesium (Mg), iron (Fe), calcium (Ca), potassium (K), and phosphorus (P)."

- Line 331: Can you clarify what is meant by the term "variance." Is this change in mean?

AUTHORS' RESPONSE: We mean variance in the standard statistical definition – a measure indicating the average squared deviation of each point from the mean. A higher variance means the points are spread out further from the mean, while a lower variance means they are clustered closer to the mean.

- Line 366: The added text, "but much greater in magnitude in these much larger carcasses" is not supported by the data presented

AUTHORS' RESPONSE: We have updated this discussion paragraph as follows to more fully explore the relationships amongst carcass age, ammonium inputs, and plant responses (lines 366-381):

"The initial influx of ammonium from elephant carcasses may have time-dependent impacts on plant abundance at elephant carcass sites. The mean ammonium level at the center of carcass sites (34.8 mg/kg) was higher than the level generally considered toxic to plants (Britto & Kronzucker, 2002). Yet, we found living grass—typically *U. trichopus*—in the center of the carcass site at seven out of ten of our sites (ammonium range 10-172 mg/kg) and at the 2.5m distance for all sites (ammonium range 0-72 mg/kg). The three sites without vegetation in the center had the highest ammonium levels (70-144 mg/kg), suggesting that *U. trichopus* has a higher degree of ammonium tolerance than some sympatric grass species but may still be limited by the high ammonium levels at the centers of these three relatively fresh carcass sites. However, the recentness of the disturbance from the carcass likely also plays a role in determining plant abundance near the center of the carcass. Because of the elephant carcass site age distribution, (mean 350 days postmortem; range 24-811 days), this study may not have captured the full impact of ammonium release from carcasses during the early stages of decomposition. Soil ammonium spiked early and decreased rapidly (Figure 5A), and future research on carcasses within the first few weeks postmortem would enhance our understanding of these early nutrient dynamics."

- Lines 367-372: As discussed above, why is mg/L being used here. This part of the discussion is not supported by the results presented.

AUTHORS' RESPONSE: As described above, in lines 368-372, we gave ammonium concentrations in mg/L to allow easy comparison with Britto & Kronzucker (2002) (cited in line 369), showing how soil ammonium levels compare with those typically considered toxic to plants. We see now that this was unclear and have updated the paragraph to show all units in mg/kg:

"The mean ammonium level at the center of carcass sites (34.8 mg/kg) was higher than the level generally considered toxic to plants (Britto & Kronzucker, 2002). Yet, we found living grass— typically *U. trichopus*—in the center of the carcass site at seven out of ten of our sites (ammonium range 10-172 mg/kg) and at the 2.5m distance for all sites (ammonium range 0-72 mg/kg). The three sites without vegetation in the center had the highest ammonium levels (70-144 mg/kg), suggesting that *U. trichopus* has a higher degree of ammonium tolerance than some sympatric grass species but may still be limited by the high ammonium levels at the centers of these three relatively fresh carcass sites."

- Line 379: The way this sentence reads suggests that nitrate is sourced from heterotrophic microbes. Consider revising this sentence to clarify that the carbon story relates to heterotrophic microbes, but the nitrate does not necessarily inform heterotrophic microbial activity.

AUTHORS' RESPONSE: We have revised this sentence as follows (lines 382-384): "Soil nitrate (Figure 2B) and soil respiration potential (Figure 3) were also elevated near the center of carcass sites, indicating higher activity rates for nitrifying bacteria and heterotrophic microbes (Prosser, 2011)."

- Line 386-387: The units here for respiration rates are not presented on a gram dry weight basis—is this just a typo?

AUTHORS' RESPONSE: We have updated the units to "µg $CO_2$ per hour per gram of dry soil".

- Lines 442-444: Because of some of the confusing regarding the data as presented, this sentence is not supported clearly by the data. The discussion should have some more specific comparisons to smaller carcasses and their impacts on ecosystems to support this sentence (and whole paragraph).

AUTHORS' RESPONSE: We have updated this paragraph to include comparisons with smaller carrion and other nutrient sources (e.g., termite mounds) (lines 446-467):

"The magnitude of nitrogen inputs from megacarcasses, as well as the substantial size and duration of their impact zones, means their impacts on ecosystem processes may be functionally distinct from smaller carrion. Soil nitrate concentrations at elephant carcass sites are orders of magnitude higher than at carcass sites of smaller carrion (e.g., rabbits, white-tailed deer, kangaroo) (Quaggiato et al. 2019; Bump et al. 2009; Barton et al. 2016). Even for large ungulates such as moose, total soil inorganic nitrogen (ammonium + nitrate) at carcass sites is a mean 300 mg/kg (Bump, Peterson, & Vucetich, 2009), substantially lower than the mean total soil inorganic nitrogen at elephant carcass sites (2.5m distance; 473 mg/kg). Termite mounds, another long-lasting source of savanna nutrient heterogeneity, have mean soil nitrate concentrations (197 mg/kg) lower than elephant carcass sites, but maximum nitrate concentrations that are on par with them (974 mg/kg) (Seymour et al. 2014), again indicating that elephant carcasses are one of the strongest known individual contributors of soil nitrogen in African savanna ecosystems, which may have important implications for savanna ecology. Indeed, there is evidence that carcass size strongly impacts scavenger food web structure (Moleón et al. 2015; Morris et al. 2023). Moreover, the attraction of animals to carcasses via scavenging, predation, or mourning (Goldenberg & Wittemyer, 2020) could have positive feedbacks on nutrient cycling (Bump, Peterson, & Vucetich, 2009; Monk et al. 2024), which may be magnified by carcass size. Thus, the impacts of megacarcasses on savanna ecosystem processes may be dissimilar to the effects of small carrion and more similar to other more persistent contributors to savanna ecosystem processes, such as termite mounds (Davies et al. 2016), cattle bomas (Augustine, 2003), and even mass animal mortality events (Subalusky et al. 2017, 2020)."

- Line 457: I would be careful with listing "nitrate" here as being sourced from the carcasses. Ammonium is sourced from the carcasses, but nitrate forms as microbes transform the ammonium into nitrate through nitrification. The increase in nitrate only occurs later in decomposition once the system returns to being oxygenated (not examined here). The carcass itself is not releasing/introducing nitrate as the nitrogen pool. I would delete the reference to nitrate here.

AUTHORS' RESPONSE: We have changed the phrase "pulses of ammonium, nitrate, and phosphate" to "pulses of nitrogen and phosphate".

- Line 462: While I agree that this type of work could help us understand the impacts of megaherbivore declines on ecosystems, I would remove the term "Anthropocene." This is not an officially recognized term (it was rejected this year by the geological community). I would just replace that word with "at present" or "in modern ecosystems" or something similar.

AUTHORS' RESPONSE: We have removed the word Anthropocene and rephrased this as "in modern ecosystems".

- In all figures, use appropriate superscript or subscript on axis labels

AUTHORS' RESPONSE: We have corrected figures so that all axis labels show appropriate superscripts and/or subscripts.

**Response to Feedback from Reviewer 3**

The authors provide careful revisions to my comments, which are very much appreciated. I have a couple of remaining comments.

AUTHORS' RESPONSE: Thanks for your constructive feedback.

On the available N, the authors make a convincing case that freezing didn't affect the results greatly. However, this leaves the potential influence of storage time between sampling and freezing, which can lead to considerable changes in nitrate and ammonium concentrations. How long between sampling soil and freezing it? This is not stated in the revised description. Comparison with literature values is only helpful if those data were collected with minimal storage time, but I expect that almost all studies mentioned involve some storage time before extraction, freezing, etc. Often samples are stored for many days (fridge etc) before extraction, leading to 100+ pp concentrations of available N. This is a well-known artifact that is still not widely recognized in ecological studies. Studies that extract immediately - e.g. using in-field KCl extraction or rapidly upon return to the laboratory - tend to find very low extractable N concentrations. Anyway, as the authors mention, the samples were all treated in the same way, so the patterns with distance are presumably realistic. However, the stark contrast between the apparent strong N limitation based on foliar data and extremely high N availability based on KCl extractions is a conundrum. If someone can do in-field KCl extractions sometime at Kruger (especially seasonally?) it would be very helpful in interpreting the nutrient situation there. If such studies have already been done, it would be useful to compare them to the results presented here.

AUTHORS' RESPONSE: We stored our soil samples in commercial freezers (-20°C) the same day they were collected. Soil freezing times varied depending on the date the soils were collected, with the earliest samples frozen for ~1 month longer than the latest samples. We have updated the methods as follows (lines 156-158): "The rest of each sample was placed in a plastic bag on the day of collection and stored in a -20°C freezer for up to 1 month; samples were stored in coolers with ice blocks during the transition from the freezer at the field site to the lab for analysis."

Unfortunately, many savanna nutrient studies do not provide sample storage times or conditions. We performed a search on Google Scholar for peer-reviewed manuscripts from 2000 onward using the search time "soil nitrate African savanna". Of the first 20 papers listed, 8 performed lab analyses measuring soil nitrate on untreated African savanna soils. Five of the eight studies used KCl extractions, while the other three used soil:water extractions. Three of the eight relevant manuscripts stored samples in the freezer at -20°C, two stored them in refrigerators, and two did not state how samples were stored. None of the studies provided information on sample storage time. We agree with the reviewer that this systemic problem in reporting methods makes cross-study comparisons difficult, and we appreciate the suggestion that we add our own sample storage times to the manuscript, which we have done.

Seymour et al. (2014) measured soil nitrate concentrations in Zimbabwe using KCl extractions and found that mean soil nitrate in control sites was 3.72 mg/kg (range 2-7 mg/kg) but was substantially higher (mean 197 mg/kg, range 3-974 mg/kg) in soil taken from termite mounds. However, they found only a weak difference in foliar nitrogen concentrations on vs. off termite mounds. These results are consistent with what we found – extremely high nitrate concentrations at the center of a hypothesized nutrient hotspot, with a much smaller impact on foliar nitrate.

Seymour, C. L., Milewski, A. V., Mills, A. J., Joseph, G. S., Cumming, G. S., Cumming, D. H. M. & Mahlangu, Z. Do the large termite mounds of *Macrotermes* concentrate micronutrients in addition to macronutrients in nutrient-poor African savannas? Soil Biology and Biogeochemistry, 68, 95-105, https://doi.org/10.1016/j.soilbio.2013.09.022, 2014.

In response to my question about evidence for Ca limitation in savannas the authors state that evidence does exist. They provide citations and they have amended their manuscript to include this. However, one citation is a review from 1986 that provides no evidence for cation limitation in natural plant communities anywhere (in fact, they state that "There is virtually no direct evidence that native terrestrial ecosystems are nutrient limited"), another citation is a classic paper on uplift of nutrients by plants but does not provide direct evidence of cation limitation. A third citation concludes that productivity in woodland savanna is co-limited by N and P. Another review deals with N and P and also does not provide any evidence of cation limitation. The final paper deals with potassium dynamics in cropping systems where K is an issue because N and P are supplied as fertilizer. So, I ask again - is there any evidence for cation limitation of plant growth in natural communities on African savannas? Appropriate evidence would be an increase in productivity following addition of calcium or potassium to a savanna plant community.

AUTHORS' RESPONSE: This manuscript tests the impacts of elephant carcasses on soil and plant nutrient concentrations, with potential implications for herbivore foraging activity. Thus, whether or not African savanna plants are cation limited is outside the scope of what we are

testing. We have updated this section of the introduction to better reflect the scope and focus of our manuscript as follows (lines 91-96):

"Aboveground, plant growth in African savannas is limited by nutrient availability, most commonly N and P (Ries & Shugart, 2008; Pellegrini, 2016), and micronutrients such as sodium (Na) potassium (K) may co-limit plant productivity as well (Epron et al. 2012; Chen et al. 2024). Thus, the large influx of nutrients released from megacarcasses might increase the mobilization of nutrients by plants, potentially increasing nutrient accessibility for vertebrate and invertebrate herbivores (Yang, 2008; Grant & Scholes, 2006; Anderson et al. 2010; Joern et al. 2012)."

Throughout the manuscript, we have gone back to our original use of the word "micronutrient" to refer to soil cations and anions, as we include both in our analyses.

Please note that orders in Soil Taxonomy are capitalized (Alfisols, Andisols, etc).

AUTHORS' RESPONSE: Thanks for catching this. We have capitalized Inceptisols, Versitols, and Andisols (line 124).